# Unraveling the developmental dynamic of visual exploration of social interactions in autism

**Nada Kojovic[1]\*, Sezen Cekic[2], Santiago Herce Castañón[3], Martina Franchini[4], Holger Franz Sperdin[1], Corrado Sandini[1], Reem Kais Jan[5], Daniela Zöller[6], Lylia Ben Hadid[1], Daphné Bavelier[2], Marie Schaer[1]\***

[1]Psychiatry Department, Faculty of Medicine, University of Geneva, Geneva, Switzerland; [2]Faculte de Psychologie et Science de l'Education, University of Geneva, Geneva, Switzerland; [3]Instituto de Investigaciones en Matemáticas Aplicadas y en Sistemas, Universidad Nacional Autónoma de México, Ciudad Universitaria, Mexico City, Mexico; [4]Fondation Pôle Autisme, Geneva, Switzerland; [5]College of Medicine, Mohammed Bin Rashid University of Medicine and Health Sciences, Dubai, United Arab Emirates; [6]Bosch Sensortec GmbH, Reutlingen, Germany

**Abstract** Atypical deployment of social gaze is present early on in toddlers with autism spectrum disorders (ASDs). Yet, studies characterizing the developmental dynamic behind it are scarce. Here, we used a data-driven method to delineate the developmental change in visual exploration of social interaction over childhood years in autism. Longitudinal eye-tracking data were acquired as children with ASD and their typically developing (TD) peers freely explored a short cartoon movie. We found divergent moment-to-moment gaze patterns in children with ASD compared to their TD peers. This divergence was particularly evident in sequences that displayed social interactions between characters and even more so in children with lower developmental and functional levels. The basic visual properties of the animated scene did not account for the enhanced divergence. Over childhood years, these differences dramatically increased to become more idiosyncratic. These findings suggest that social attention should be targeted early in clinical treatments.

**\*For correspondence:**
nada.kojovic@unige.ch (NK);
marie.schaer@unige.ch (MS)

**Competing interest:** The authors declare that no competing interests exist.

## Editor's evaluation

This is an important study investigating a rare longitudinal dataset of eye-tracking to a cartoon video, measured in a group of children with autism and a control group that is typically developing. The core finding is a divergence in exploratory gaze onto the video stimulus in the children with ASD, compared to typically developing children, this finding is supported by convincing evidence. In addition, the effect appeared to be parametric: those autistic children with the least divergence also had the best adaptive functioning and communication skills. Additional strengths of the study are a relatively large sample size for this type of work and analyses that aim at generalizability. This study will be interesting for autism specialists, but also for a wider community interested in social cognitive, affective neuroscience, and developmental disorders.

## Introduction

Newborns orient to social cues from the first hours of life. They show privileged attention to faces (*Simion et al., 2001*), face-like stimuli (*Goren et al., 1975*; *Johnson et al., 1991*; *Valenza et al., 1996*), and orient preferentially to biological motion (*Simion et al., 2008*). This automatic and

preferential orientation to social cues early in life is highly adaptive as it provides grounds for developing experience-dependent competencies critical for an individual's adequate functioning. Social visual engagement is one of the first means of exploration and interaction with the world, preceding and determining more advanced levels of social interaction and autonomy (*Klin et al., 2015*). Impairments in this elemental skill are one of the core characteristics of ASD, a highly heterogeneous lifelong neurodevelopmental condition (*American Psychiatric Association, 2013*). Broad impairments in social communication and interaction, along with repetitive behaviors and circumscribed interests, have been suggested to lead to a spectrum of functional disabilities in ASD (*Klin et al., 2007*). In this regard, atypical social attention strategies may at least partially contribute to the emergence of the ASD phenotype. Many studies using eye-tracking have explored the atypicalities in attentional processes and their contribution to core symptoms in ASD (*Chawarska and Shic, 2009*; *Klin et al., 2003*; *Falck-Ytter et al., 2013a*). Recent meta-analyses concluded that, besides generally reduced social attention (*Chita-Tegmark, 2016b*), autism is also characterized by atypical attention deployment during the exploration of social stimuli (*Chita-Tegmark, 2016a*). Indeed, aside from a generally diminished interest in social stimuli, when individuals with ASD do attend to social information, they spend less time exploring key features, such as eyes while showing an increased interest in less relevant cues, such as bodies (*Chita-Tegmark, 2016b*). These atypicalities are observed as early as two months of age (*Jones and Klin, 2013*) and thus can exert a tremendous impact on downstream developmental processes that critically depend on experience. The exact biological mechanisms that govern the emergence of these aberrant social attention patterns and their course of evolution are currently unknown.

In typical development, following the initial social preference, social attention deployment shows dynamic changes during infancy and early childhood. During their first year of life, infants progressively increase the time spent looking at faces compared to other elements of their environment (*Frank et al., 2009*). The increasing ability to attend to faces in complex environments has been related to developmental changes in visual attention (*Frank et al., 2014*). Indeed, during the first year of life, we observe the development of more endogenous, cortically controlled attention (*Colombo, 2001*), which allows more flexible and controlled displacement of gaze (*Hunnius and Geuze, 2004*; *Hendry et al., 2018*; *Frank et al., 2014*; *Helo et al., 2016*). Developmental improvement in attentional abilities thus promotes engagement with social targets. Furthermore, the increase in capacity to attend to highly relevant social elements is followed by increased similarity in fixation targets between TD children (*Frank et al., 2014*). With increasing age, the TD children show more coherence in their visual behavior, as they increasingly focus on similar elements of the scene (*Franchak et al., 2016*; *Frank et al., 2009*; *Shic et al., 2008*). A trend toward progressively more coherent gaze patterns continues into adulthood (*Kirkorian et al., 2012*; *Rider et al., 2018*). In other words, despite the impressive complexity of our social environment and the diversity of each individual's experiences, social visual engagement takes a convergent path across TD individuals, who are increasingly looking at similar elements of the social environment. However, the current understanding of the dynamic of this progressive tuning of gaze patterns is limited by the scarcity of studies using longitudinal designs. Indeed, most studies used cross-sectional designs when inferring developmental patterns, which can be biased by interindividual differences.

In regards to autism, understanding the typical development of social visual exploration is of utmost importance, as the social difficulties associated with ASD result from the cascading effect of a reduced social interest during the child's development (*Dawson et al., 1998*; *Dawson et al., 2005*; *Chevallier et al., 2012*). Studies focusing on the developmental changes in visual exploration in autism are still rather scant but point to altered maturational changes in orienting to social cues. Attention deployment begins to differ from the age of 2 months in babies who later develop autism, suggesting that divergent trajectories of social visual exploration may start in the first months of life (*Jones and Klin, 2013*). A study by *Shic et al., 2008* highlighted the absence of typical maturational change in face scanning strategies in children with ASD between 2 and 4 years of age. Longitudinal studies focusing on typical and atypical development are thus crucially needed to highlight the underlying developmental mechanisms of atypical attention deployment in ASD. Longitudinal follow-up design would allow the identification of periods of critical changes in visual behavior that can be targeted by early interventions. In addition to the parsing of the developmental patterns, a comprehensive characterization of factors that influence visual behavior

in the social context is necessary to understand the mechanisms of atypical attention deployment in autism.

Gaze deployment is mediated by numerous factors acting simultaneously, including bottom-up and top-down processes. Bottom-up mechanisms direct attention to visually prominent elements as a function of their basic properties (such as orientation, intensity, color, and motion) (*Itti and Koch, 2000*; *Itti et al., 2001*; *Koch and Ullman, 1985*) while top-down factors (*Itti et al., 2001*) are more endogenous in nature and depend on previous experience, motivation, specific task demands, etc. (*Yarbus, 1967*). The complex interplay between these two processes orchestrates our attention deployment during everyday tasks. We can hypothesize that the imbalance, such as enhanced focus on bottom-up properties of visual content, maybe at the origin of atypical social attention in autism, driving it away from conventional social targets. Indeed, it has been shown that in the context of naturalistic static scenes, children and adults with ASD tend to focus more on basic, pixel-level properties than on semantic categories, compared to their TD peers (*Amso et al., 2014*; *Wang et al., 2015*). However, less is known of the contribution of these basic properties to a real-time visual exploration of dynamic content, as static stimuli only allow limited inference to the real-world dynamic deployment of attention. Studies using dynamic social content are rare and point to somewhat contrasting results compared to the ones using static stimuli. For example, it has been shown that in the context of dynamic social content, preschoolers with ASD tend to focus less on the motion properties of the scene and more on luminance intensity compared to age-matched TD children (*Shic et al., 2007*). However, there is currently no consensus in the literature on the relative predominance between bottom-up and top-down properties in generating aberrant visual exploration. These two processes were mostly analyzed separately, and studies using ecological dynamic stimuli are scarce. Hence, another important element is the content type, as it dramatically influences the attentional processes summoned. For instance, non-social content is prone to elicit more heterogeneous patterns of exploration (*Wang et al., 2018*). On the other hand, the social content of higher complexity induces more divergence in gaze deployment in TD (*Wang et al., 2018*) while giving rise to atypicalities in visual attention deployment in ASD (*Chawarska et al., 2012*; *Chita-Tegmark, 2016b*).

Measures of gaze deployment (e.g. time spent on the face or eyes) provided valuable insight into the specificity of social attention patterns in autism (*Klin et al., 2002*). These measures reflect the 'macrostructure' (*Guillon et al., 2014*) of the gaze deployment by quantifying the overall time spent exploring a predefined scene region. However, complementary to the 'what' of gaze, the 'when' of it is of equal importance as the demands in the real world come online and require a timely response. We attend to only a limited amount of elements from a breadth of possibilities, and what finds the way to our perception will dramatically influence the meaning we attribute to the social situation. Recent studies have provided important advances in our understanding of the mechanisms that control what we select to gaze upon on a moment-to-moment basis (*Constantino et al., 2017*; *Kennedy et al., 2017*). Quite strikingly, while viewing social scenes, toddler and school-age twins showed a high concordance not solely in the direction but also in the timing of their gaze movements (*Constantino et al., 2017*; *Kennedy et al., 2017*). Thus, subtle variations in the visual exploration of social scenes are strongly influenced by genetic factors that favor the selection of crucial social information (*Constantino et al., 2017*). The continuous active selection of pertinent elements from the abundance of possibilities is critical for the interactive specialization of our brain (*Johnson, 2001*) and significantly affects how our internal world is shaped. Only a few studies tackled the question of the moment-to-moment gaze deployment in ASD compared to TD. Indeed, while on this microstructural level, TD children and adults show coherence in fixation targets, the fine-grained gaze dynamic in their peers with ASD is highly idiosyncratic and heterogeneous (*Nakano et al., 2010*; *Falck-Ytter and von Hofsten, 2011*; *Wang et al., 2018*; *Avni et al., 2020*). Atypicalities in the fine-grained extraction of social information may have important consequences on learning opportunities and social functioning (*Schultz, 2005*). Overall, these findings urge for a better characterization of the underlying mechanisms and factors that contribute to coherence in visual patterns in typical development at different timescales, over months and years but also at the microstructural level (moment-to-moment) as a gateway for understanding the emergence of atypical gaze patterns in autism.

In the current study, we opted for a comprehensive approach to characterize atypical visual exploration in a large sample of 166 children with ASD (1.7–6.9 years old) compared to their age-matched TD peers (1.7–6.8 years old) by considering both bottom-up and top-down processes. We first measured

the divergence from referent gaze patterns (obtained from the TD children) in autism on a micro-structural level (moment-to-moment) and over larger temporal scales, measuring the developmental change during early childhood. We quantified the divergence between gaze patterns among the two groups of children while watching a cartoon depicting social interaction using a custom data-driven approach used in our previous studies (*Sperdin et al., 2018*; *Jan et al., 2019*; *Kojovic et al., 2019*). We estimated the relative contribution of basic visual properties of the scene to the visual exploration of this dynamic social scene in both groups. Finally, we measured the contribution of the different features of the video content (visual and social complexity, directedness of speech) to the divergence from the referent gazing patterns in the ASD group. We further measured the developmental change in visual exploration in young children with ASD and their TD peers using the yearly follow-up eye-tracking recordings.

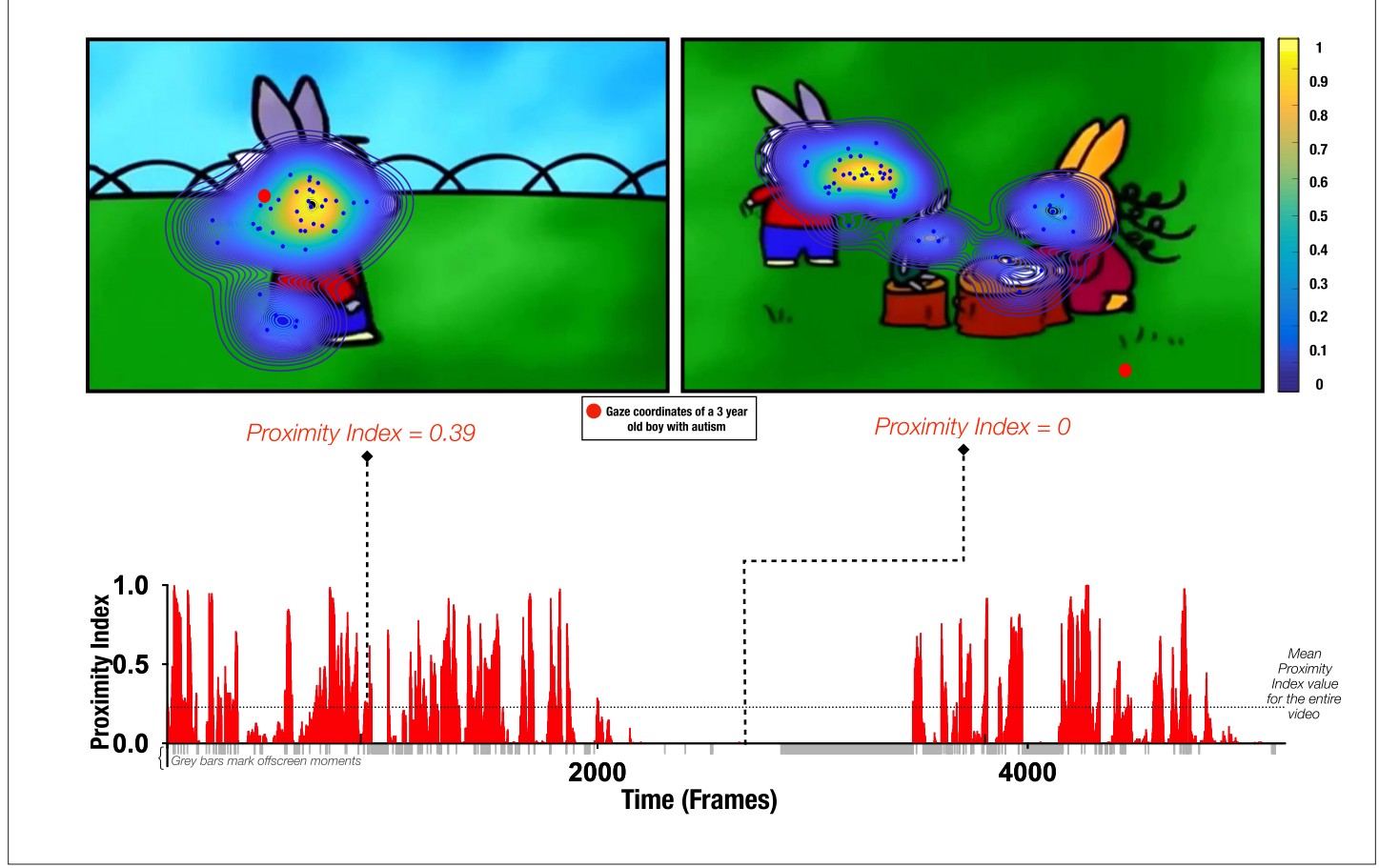

**Figure 1.** Proximity Index method illustration. Referent gaze data distribution ('reference') was created using gaze coordinates from 51 typically developing (TD) males (aged 3.48±1.29 years old). Upper row: two example frames with gaze coordinates of TD children (blue dots) used to define the 'reference' (delimited by contours) and gaze data from a three-year-old male with autism spectrum disorder (ASD) (whose gaze coordinates are depicted as a red circle). Hotter contour color indicates the area of higher density of distribution of gaze in the TD group, meaning that a particular area was more appealing for a higher number of TD preschoolers for the given frame; the Proximity Index value for the 3-year-old male with ASD for the frame on the left had a value of 0.39 and for the frame on the right a value of 0. Lower row: Proximity Index values for the visual exploration of the 3-year-old boy with ASD over the entire video with the mean Proximity Index value indicated by the dashed red lines.

## Results

### Divergence from the typical gazing patterns, its relation to clinical phenotype and movie properties

#### Moment-by-moment divergence from the referent gazing patterns

Gaze data from 166 males with ASD (3.37 ±1.16 years) were recorded while children watched a 3 min episode of the French cartoon *Trotro* (**Lezoray, 2013**). The cartoon depicts social interaction between the three donkey characters at a relatively slow pace. We were interested in capturing the difference in moment-to-moment gaze deployment in ASD children compared to the TD group while watching this animated social scene. For this, we compared the gaze allocation of each child with ASD to the referent gaze patterns obtained from 51 age-matched TD males (3.48 ±1.29 years) who watched the same social scene. Referent gaze patterns ('reference') were obtained by applying the probability density estimation function (**Botev et al., 2010**) on gaze data from the TD group on each frame. Hence, for each child with ASD, we obtained a measure indicating the closeness to the reference that we denote *Proximity Index-PI,* (see **Figure 1** and Methods section for detailed explanation). Lower PI values indicate a higher divergence from the reference for the given frame. As the obtained measure

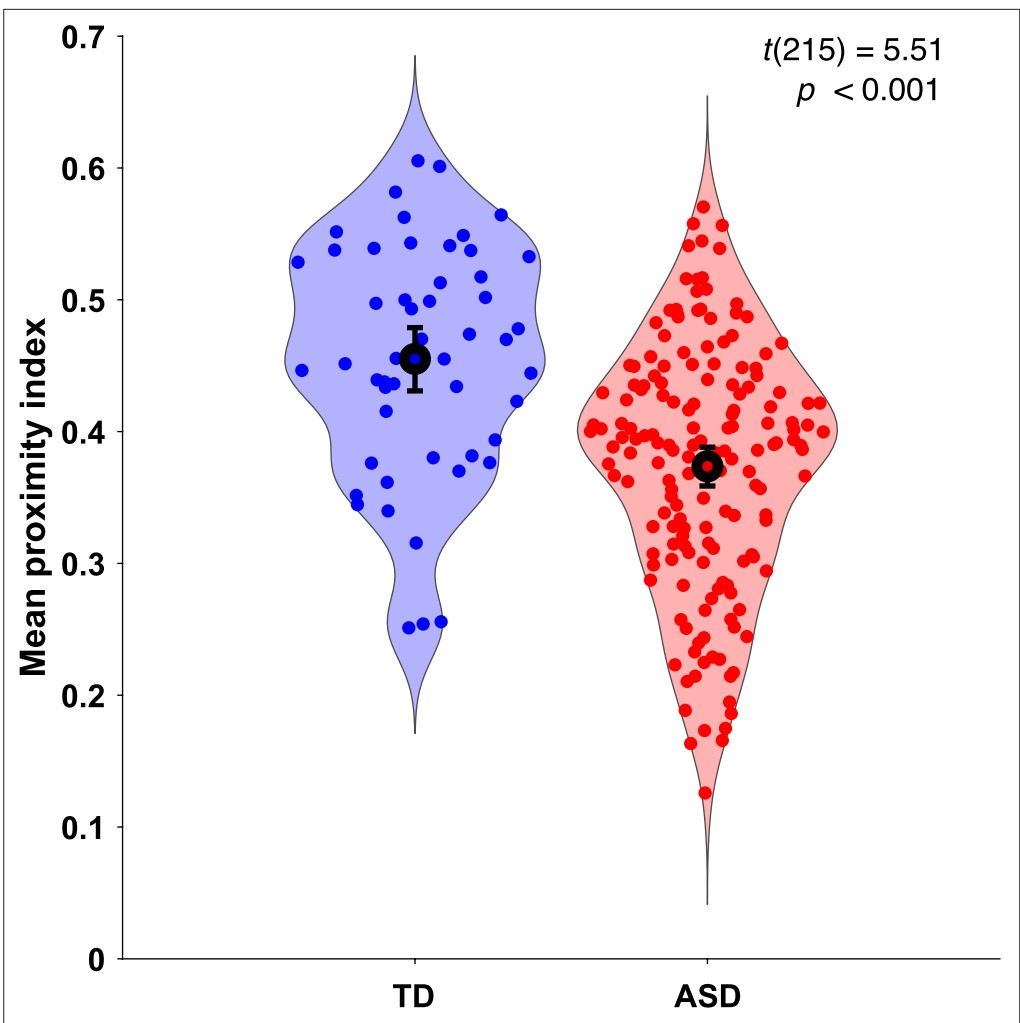

**Figure 2.** Mean proximity index (PI) comparison between groups. Violin plots illustrate the distribution of Proximity Index (PI) values for two groups: typically developing (TD) in blue (n = 51) and autism spectrum disorder (ASD) in red (n = 166). The error bars on each plot represent the 95% confidence intervals around the means. Statistical significance of the differences between means was assessed using a two-sample t-test. The PI values for the TD group were derived using a leave-one-out approach, where the PI for each ASD child was calculated based on the referent gaze data from the 51 TD children in the original sample.

dynamically determines the proximity to the referent gaze distribution, there is no need to define areas of interest based on the theoretical priors. Moreover, as it will be further detailed, this method allowed flexibly redefining the referent gaze distribution by constraining the reference sample to a specific age range or group.

As the reference TD group was a convenience sample, we ran a bootstrap analysis to ensure that the obtained referent distribution was not affected by sample size (see Appendix 1 for more details). According to our stability analyses, the sample size of 51 TD children allows us to define the reference with enough stability, considering it is more than two times bigger than the estimated smallest stable sample size of 18.

As the gaze data of the TD group were used as a reference, we wanted to understand how their individual gazing patterns would behave compared to a fixed average. To this end, we employed the leave-one-out method to obtain the PI value for each of the 51 TD children. In this manner, the gazing pattern of each TD child was compared to the reference created by the gaze data of 50 other TD children. The difference in average PI values between the two groups was found significant, $t(215)=5.51$, p<0.001 (*Figure 2*).

## Less divergence in visual exploration is associated with better overall functioning in children with ASD

To explore how the gaze patterns, specifically divergence in the way children with ASD attended to the social content, related to the child's functioning, we conducted a multivariate analysis. We opted for this approach to obtain a holistic vision of the relationship between visual exploration, as measured by PI, and different features of the complex behavioral phenotype in ASD. Behavioral phenotype included the measure of autistic symptoms and the developmental and functional status of the children with ASD. Individuals with ASD often present lower levels of adaptive functioning (*Bal et al., 2015*; *Franchini et al., 2018*) and this despite cognitive potential (*Klin et al., 2007*). Understanding factors that contribute to better adaptive functioning in very young children is of utmost importance (*Franchini et al., 2018*) given the important predictive value of adaptive functioning on later quality of life. The association between behavioral phenotype and PI was examined using the PLS-C analysis (*Krishnan et al., 2011*; *McIntosh and Lobaugh, 2004*). This method extracts commonalities between two data sets by deriving latent variables representing the optimal linear combinations of the variables of the compared data sets. We built the cross-correlation matrix using the PI on the left (A) and 12 behavioral phenotype variables on the right (B) side (see Methods section for more details on the analysis).

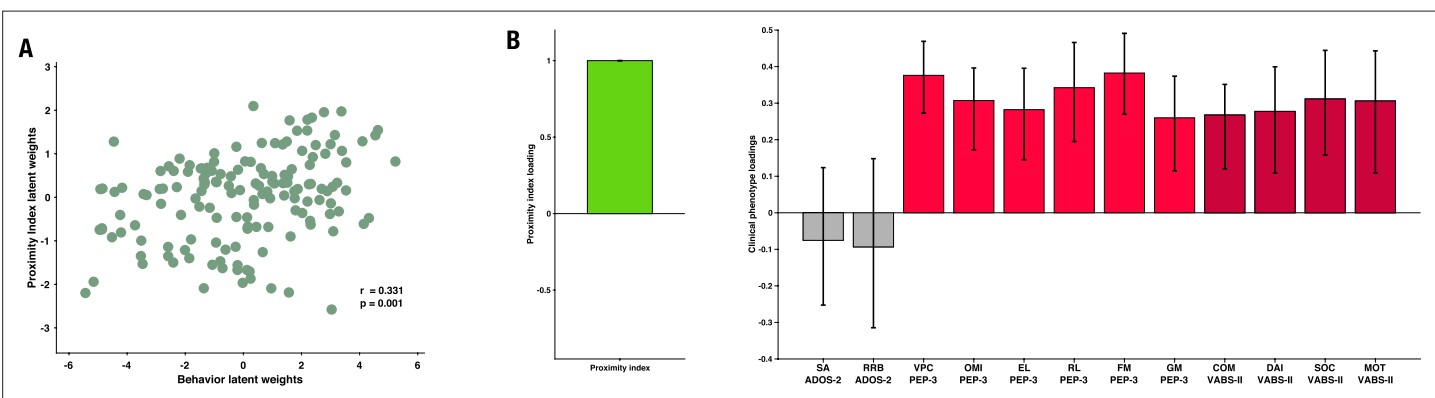

**Figure 3.** Proximity Index and its relation to behavioral phenotype in children with autism spectrum disorder (ASD). Loadings on the latent component were derived using partial least squares correlation analysis in the sample of 166 children with ASD. The cross-correlation matrix consisted of the Proximity Index on the imaging (**A**) side and 12 variables on the behavior (**B**) side. The behavioral matrix encompassed two domains of autistic symptoms assessed by ADOS-2: Social Affect (SA) and Repetitive and Restricted Behaviors (RRB); six subscales of the PEP-3: Verbal and Preverbal Cognition (VPC), Expressive Language (EL), Receptive Language (RL), Fine Motor Skills (FM), Gross Motor Skills (GM), and Oculomotor Imitation (OMI); and four domains from VABS-II: Communication (COM), Daily Living Skills (DAI), Socialization (SOC), and Motor Skills (MOT). Age was controlled for by regressing it out from both sides (**A** and **B**) of the cross-correlation matrix. There was a positive correlation between the Proximity Index and all measures of developmental (PEP-3) and adaptive functioning (VABS-II). Error bars represent the bootstrapping 5th to 95th percentiles. Results that were not robust are indicated by a gray boxplot color.

In our cohort, child autistic symptoms were assessed using the ADOS (*Lord et al., 2000*; *Lord et al., 2012*), child developmental functioning using the PEP-3 scale (*Schopler, 2005*) and child adaptive behavior using the Vineland Adaptive Behavior Scales, Second Edition, (*Sparrow et al., 2005*). Thus the final behavior matrix included two domains of autistic symptoms from the ADOS: social affect (SA) and repetitive and restricted behaviors (RRB); six subscales of the PEP-3: verbal and preverbal cognition (VPC), expressive language (EL), receptive language (RL), fine motor skills (FM), gross motor skills (GM), oculomotor imitation (OMI) and four domains from VABS-II: communication (COM), daily living skills (DAI), socialization (SOC), and motor skills (MOT). Age was regressed from both sets of the imputed data.

The PLS-C yielded one significant latent component ($r=0.331$, p=0.001), best explaining the cross-correlation pattern between the PI and the behavioral phenotype in the ASD group. The significance of the latent component was tested using 1000 permutations, and the stability of the obtained loadings was tested using 1000 bootstrap resamples. Behavioral characteristics that showed stable contributions to the pattern reflected in the latent component are shown in red *Figure 3*. Higher values of the PI were found in children with better developmental functioning across all six assessed domains and better adaptive functioning across all four assessed domains. Autistic symptoms did not produce a stable enough contribution to the pattern (loadings showed in gray bars on the *Figure 3*). Still, numerically, a more TD-like gazing pattern (high PI) was seen in the presence of fewer ASD symptoms (negative loading of both SA and RRB scales of the ADOS-2). Despite the lack of stability of this pattern, the loading directionality of ASD symptoms is in line with the previous literature (*Wen et al., 2022*; *Avni et al., 2020*), showing a negative relationship between visual behavior and social impairment. Among the developmental scales, the biggest loading was found on verbal and preverbal cognition, followed by fine motor skills. While the involvement of verbal and nonverbal cognition in the PI, an index of visual exploration of these complex social scenes is no surprise, the role of fine motor skills might be harder to grasp. Interestingly, in addition to measuring the control of hand and wrist small muscle groups, the fine motor scale also reflects the capacity of the child to stay focused on the activity while performing controlled actions. Thus, besides the measure of movement control, relevant as scene viewing implies control of eye movement, the attentional component measured by this scale might explain the high involvement of the fine motor scale in the latent construct pattern we obtain.

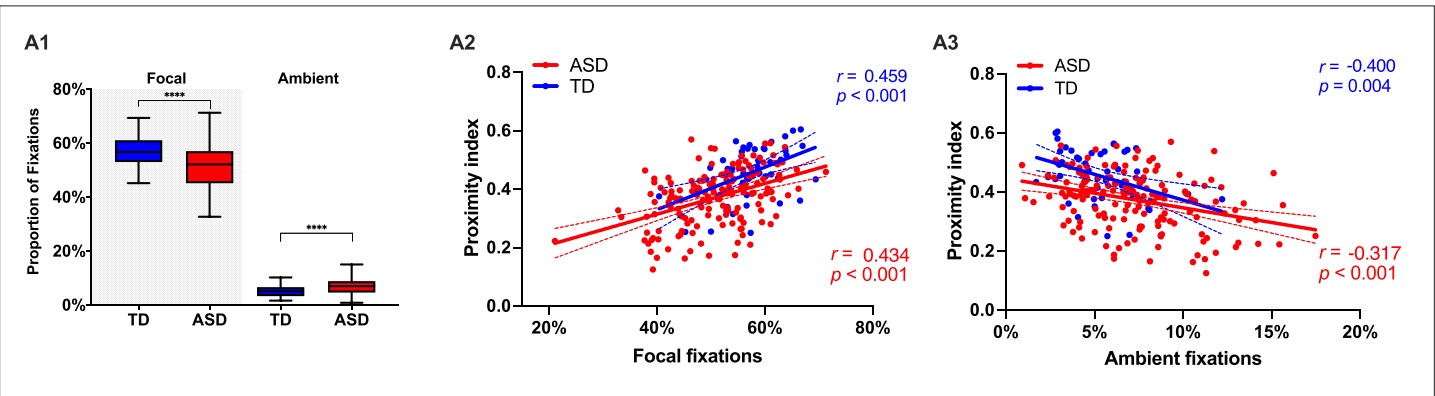

**Figure 4.** Focal and ambient fixation modes, between-group comparison, and their relation to the Proximity Index (PI) across ASD and TD groups. (**A1**) Relative proportion of focal and ambient fixations in a sample of 51 TD children and 166 ASD children. Box-and-whisker plots illustrate the distribution of fixation proportions. The interquartile range (IQR) is represented by each box, with the median shown as a horizontal line. Whiskers extend to the most extreme data points within 1.5 IQR from the box, as per Tukey's method. Differences between groups were statistically assessed using the Mann-Whitney U test, with asterisks (****) indicating p-values less than 0.0001. (**A2 & A3**): Scatter plots show the correlation between the proportion of focal (**A2**) or ambient (**A3**) fixations and PI. Red points represent ASD individuals and blue points represent TD individuals. Spearman's correlation was used for analysis. Each group's data is fitted with its own linear regression line and includes 95% confidence bands.

## More ambient and less focal fixations in children with ASD compared to the TD group

Next, we wanted to complement our analysis using standard measures of visual behavior. In our cross-sectional sample of 166 males with ASD (3.37 ±1.16 years) and 51 TD males (3.48 ±1.29 years), we did not find any significant difference between groups with regard to the overall number of fixations, saccades, median saccade duration, or saccade amplitude for the duration of the cartoon (p>0.05). However, there was a tendency in median fixation duration to be slightly higher in TD children compared to the ASD group ($t$(215) = 1.85, p=0.06), suggesting a more focused attentional style in the TD group. To characterize the predominant attention exploration mode while watching the cartoon, we defined two types of fixations based on their duration and the length of the preceding saccade. Thus using thresholds as in *Unema et al., 2005*, a fixation was considered as 'focal' if longer than 180 ms and preceded by a saccade of an amplitude smaller than 5° of visual angle. Shorter fixations <180 ms preceded by a longer saccade >5° were classified as 'ambient.' We then obtained the proportion of these two fixation types normalized for the overall fixation number. In the ASD group, we observed significantly more ambient fixations (Mann-Whitney test: $U$=2530, p<0.001) compared to the TD group. The TD group showed more focal fixations ($U$=2345, p<0.001) in comparison to the ASD group. In both groups, focal fixations were more frequent than ambient (p<0.001) (see *Figure 4A1*). Higher presence of focal fixations was positively correlated to higher values of Proximity Index in both groups ($r_{TD}$=0.459, $r_{ASD}$ = 0.434, p<0.001) while the opposite relationship was evidenced between Proximity index and proportion of ambient fixations ($r_{TD}$=–0.400, $r_{ASD}$ = –0.31, p=0.002) (see *Figure 4* Panels A2 & 3). Compared to the ASD group, the TD group stays less in the 'shallow' exploration mode reflected by the ambient fixations. This exploration mode is deployed first to quickly extract the gist of a scene before a more in-depth scene analysis is carried out through focal fixations. Thus our findings suggest that, while in the TD group, the gist of the scene is rapidly extracted, the children in the ASD group spends significantly more time in the exploration mode, wondering where to place more deep attention compared to the TD group. Subsequently, they stay less in the focused mode of attention compared to the TD group.

## The relative contribution of the basic visual properties of the animated scene to gaze allocation in ASD and TD children

We next measured the group difference in the relative contribution of basic visual properties of the scene to visual exploration. Previous studies in adults with ASD have shown that these basic properties play an important role in directing gaze in ASD individuals while viewing naturalistic images (*Amso et al., 2014*; *Wang et al., 2015*). Less is known about the contribution of the basic scene properties to gaze allocation while viewing dynamic content. Moreover, besides using static stimuli, most studies focused on the adult population, while the early developmental dynamics of these mechanisms remain elusive. Therefore, we extracted the values of five salience features (intensity, orientation, color, flicker, motion) for each frame of the video using the variant of the biologically inspired salience model, namely graph-based visual saliency (GBVS) (*Harel et al., 2006*) as explained in details in the Methods section. We calculated salience measures for our cross-sectional sample with 166 males with ASD and age-matched 51 TD males individually for each frame. For each channel (intensity, orientation, color, flicker, and motion) as well as the full model (linear combination of all five channels), we calculated the area under a receiver operating characteristic curve (ROC) (*Green and Swets, 1966*). The mean ROC value was then used to compare the two groups.

Contrarily to our hypothesis, for all channels taken individually as well as for the full model, the salience model better-predicted gaze allocation in the TD group compared to the ASD group (Wilcoxon t-test returned with the value of p<0.001, *Figure 5*). The effect sizes ($r = Z/\sqrt{N}$, *Rosenthal, 1991*) of this difference were most pronounced for the flicker channel $r$=0.182, followed by the orientation channel $r$=0.149, full model $r$=0.132, intensity $r$=0.099, color $r$=0.083, and lastly motion $r$=0.066, Appendix 2. The finding that the salient model predicted better gaze location in TD groups compared to the ASD was not expected based on the previous literature. Still, most studies used static stimuli and the processes implicated in the process of the dynamic content are very different. The salience model itself was validated on the adult vision system. It might be that the gaze in TD better approximates the adult, mature gaze behavior than the gaze behavior in the ASD group.

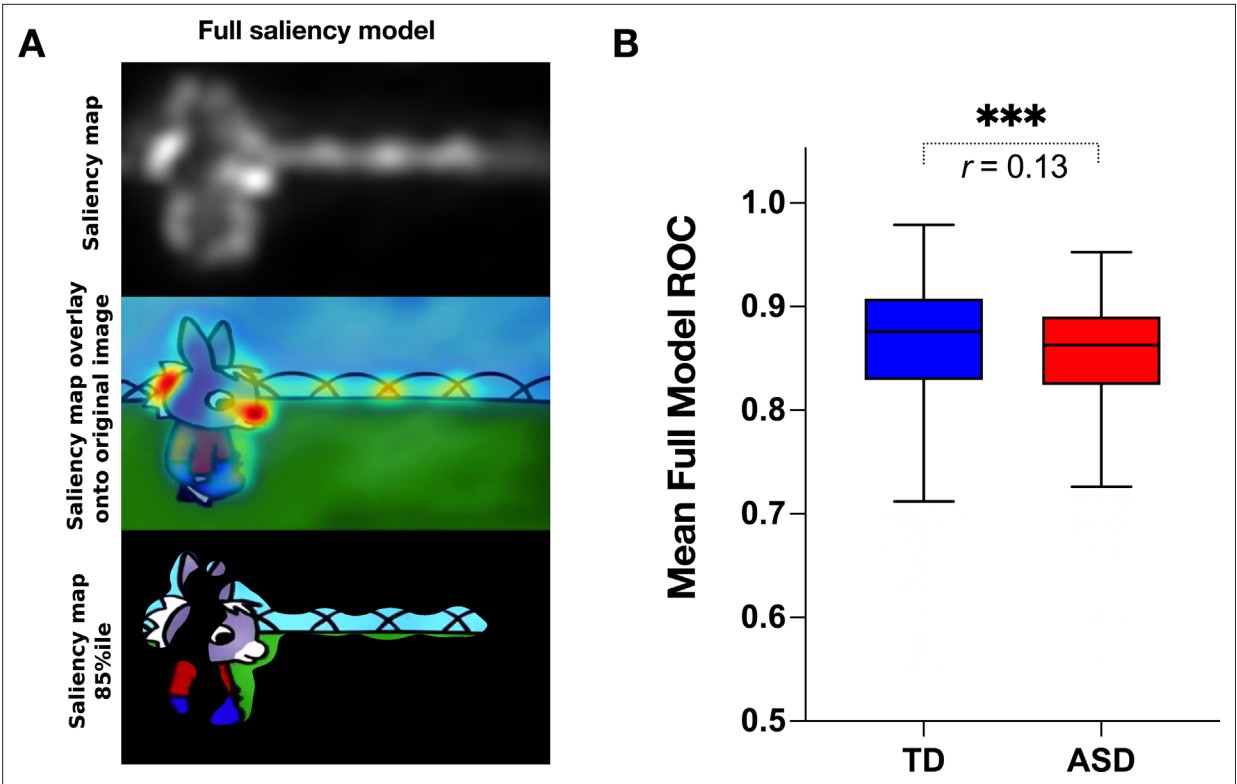

**Figure 5.** Visual salience group differences. (**A**) Illustration of the graph-based visual saliency (GBVS) salience model (Full model combining five channels: I-Intensity, O-Orientation, C-Color, F-Flicker, M-Motion). From top to bottom: Saliency map extracted for a given frame, Saliency map overlay on the original image, Original image with 15% most salient parts shown. (**B**) Box plot depicting mean receiver operating characteristic (ROC) values, derived framewise from full salience maps and fixation coordinates (x,y), for a sample of 51 TD (Typically Developing) and 166 ASD (Autism Spectrum Disorder) children. Boxes indicate the interquartile range (IQR) and medians are shown as horizontal lines within the boxes. Whiskers extend to the farthest data points not exceeding 1.5 times the IQR from the box edges, in line with Tukey's method. Framewise statistical between group differences were evaluated using the Wilcoxon paired test, with asterisks (***) indicating p-values less than 0.001. Effect size is calculated using formula $r = Z/\sqrt{N}$, (**Rosenthal, 1991**).

## The association of movie content with divergence in visual exploration in ASD group

Taking into account previous findings of enhanced difficulties in processing more complex social information (**Frank et al., 2012**; **Chita-Tegmark, 2016b**; **Parish-Morris et al., 2019**) in individuals with ASD, we tested how the intensity of social content influenced visual exploration of the given social scene. As detailed in the Methods section, social complexity was defined as the total number of characters for a given frame and ranged from 1 to 3. Frames with no characters represented a substantial minority (0.02% of total video duration) and were excluded from the analysis. We also analyzed the influence of the overall visual complexity of the scene on this divergent visual exploration in the ASD group. The total length of edges defining details on the images was employed as a proxy for visual complexity (see Methods section for more details). Additionally, we identified the moments of vocalization (monologues versus directed speech) and more global characteristics of the scene (frame cuts and sliding background) to understand better how these elements might have influenced gaze allocation. Finally, as an additional measure, we considered how well the gaze of ASD children was predicted by the GBVS salience model or the average ROC scores we derived in the previous section *Figure 6*, panel A.

To explore the relationship between the PI and different measures of the movie content as previously, we used a PLS-C analysis that is more suitable than the GLM in case of strong collinearity of the regressors this is particularly the case of the visual and social complexity (r=0.763, p<0.001), as well as social complexity and vocalization (r=0.223, p<0.001), as can be appreciated on the *Figure 6*, panel B. The PLS-C produced one significant latent component (r=0.331, p<0.001). The latent component

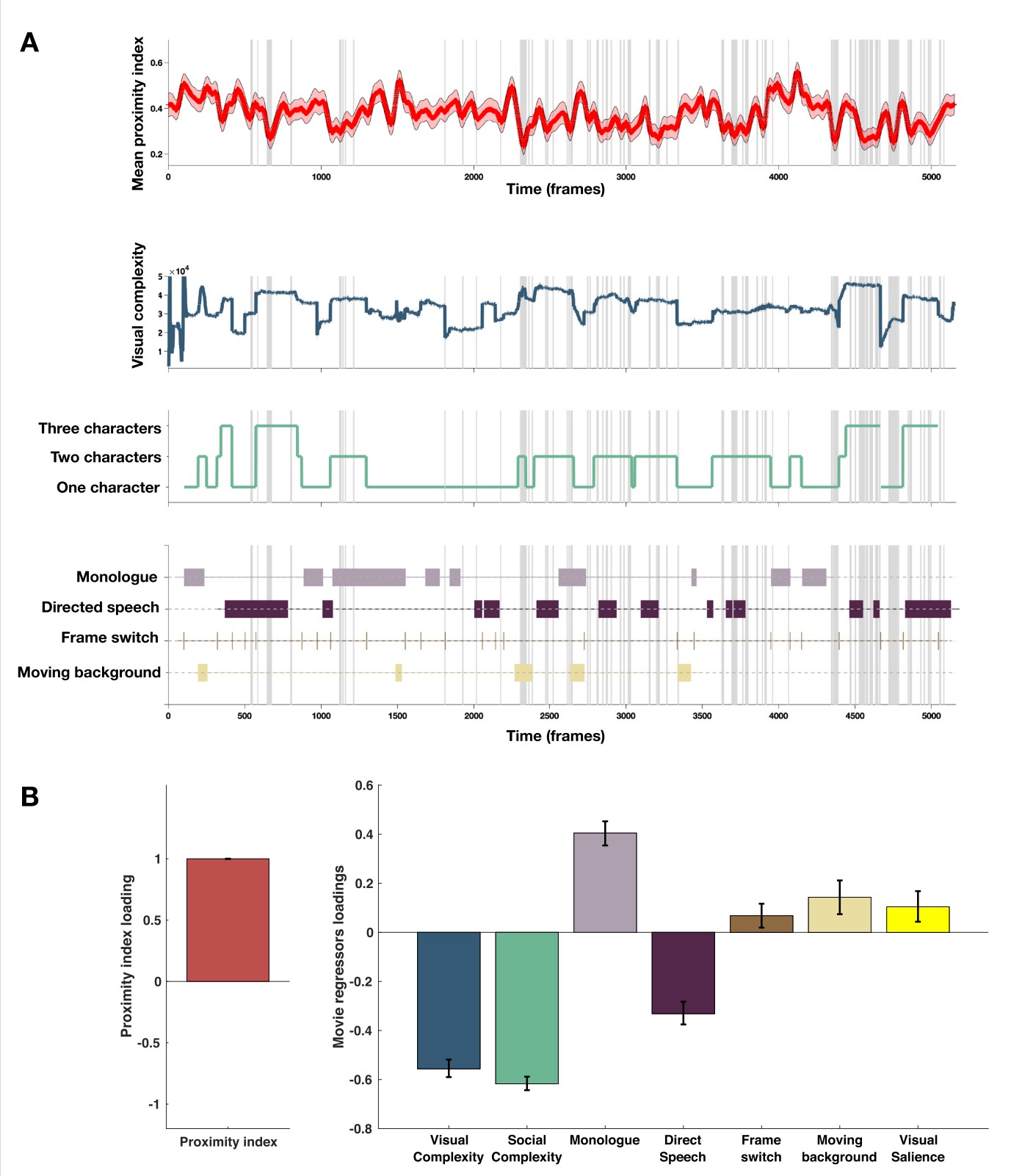

**Figure 6.** Proximity Index and its relation to movie content. (**A**) From top to down: In red, the average proximity index (PI) from 166 children with autism spectrum disorder (ASD) over time frames. Red-shaded regions denote a 95% confidence interval of the mean, gray-shaded regions mark the moments of the significant drop in mean values of the PI (below 2.5 SD compared to the theoretical mean of 1); Dark blue: Visual complexity over time frames; Green: Social complexity over time frames; the last panel denotes moments of the movie with the monologue, directed speech, frame switching, or

*Figure 6 continued on next page*

*Figure 6 continued*

moments involving moving background. (**B**) PLS-C illustration with PI on the A side and on the B side: Visual complexity, Social Complexity, Monologue, Directed Speech, Frame switch, Moving background and graph-based visual saliency (GBVS), the salience model derived receiver operating characteristic curve (ROC) scores for children with ASD (average ROC framewise). Positive correlation between the Proximity Index and was found between the Proximity Index and monologue, frame switch, moving background and also visual salience. PI negatively correlated with the social and visual complexity, as well as directed speech. Error bars represent the bootstrapping 5th to 95th percentiles.

pattern was such that lower PI was related to higher social complexity, followed by higher visual complexity and the presence of directed speech. In addition, moments including characters engaged in monologue, moments of frame change, and background sliding increased the PI in the group of ASD children. The monologue scenes also coincide with the moments of lowest social complexity that produces higher PI values. For the frame switch and the sliding background, the TD reference appears more dispersed in these moments as children may recalibrate their attention onto the new or changing scene, making the referent gaze distribution more variable in these moments and thus giving ASD more chance to fall into the reference space as it is larger. Finally, visual salience also positively contributed to the PI loading, which is in line with our previous finding of the salience model being more successful in predicting TD gaze than ASD gaze.

## Developmental patterns of visual exploration

### More divergence in visual exploration is associated with unfolding autistic symptomatology a year later

To capture the developmental change in the PI and its relation to clinical phenotype we conducted the multivariate analysis considering only the subjects that had valid eye-tracking recordings at two time points one year apart. Out of 94 eligible children (having two valid eye-tracking recordings a year apart), 81 had a complete set of phenotype measures. All 94 children had an ADOS, but ten children were missing PEP-3 (nine were assessed using Mullen Scales of Early Learning [*Mullen, 1995*], one child was not testable at the initial visit), and three children were missing VABS-II as the parents were not available for the interview at a given visit. The proximity index in this smaller paired longitudinal sample was defined using the age-matched reference composed of 29 TD children spanning the age (1.66–5.56) who also had a valid eye-tracking recording a year later. As the current subsample was smaller than the initial one, we limited our analyses to more global measures, such as domain scales (not the test subscales as in our bigger cross-sectional sample). Thus, for the measure of autistic symptoms, we used the total severity score of ADOS. Cognition was measured using the Verbal and preverbal cognition scale of PEP-3 (as the PEP-3 does not provide a more global measure of development *Schopler, 2005*) and adaptive functioning using the Adaptive behavior Composite score of Vineland (*Sparrow et al., 2005*). To test how the PI relates within and across time points, we built three cross-covariance matrices (T1-PI to T1-symptoms; T1-PI to T2-symptoms; T2-PI to T2-symptoms) with the PI on one side (A) and the measure of autistic symptoms, cognition, and adaptation on the other side (B). As previously, the significance of the patterns was tested using 1000 permutations, and the stability of the significant latent components using 1000 bootstrap samples.

The PLS-C conducted on simultaneous PI and phenotype measures at the first time point (T1-PI - T1 symptoms) essentially replicated the pattern we observed on a bigger cross-sectional sample. One significant LC ($r=0.306$ and $p=0.011$) showed higher PI co-occurring with higher cognitive and adaptive measures (see Appendix 4). The cross-covariance matrix using a PI at T1 to relate to the phenotype at the T2 also yielded one significant latent component ($r=0.287$ and $p=0.033$). Interestingly, the pattern reflected by this LC showed higher loading on the PI co-occurring with lower loading on autistic symptoms. Children who presented lower PI values at T1 were the ones with higher symptom severity at T2. The gaze pattern at T1 was not related to cognition nor adaptation at T2 (see *Figure 7*, panel A). Finally, the simultaneous PLS-C done at T2 yielded one significant LC where higher loading of the PI coexisted with negative loading on autistic symptoms and higher positive loading on the adaptation score ($r=0.322$ and $p=0.014$) *Figure 7*, panel B. The level of typicality of gaze related to the symptoms of autism at T2 (mean age of 4.05±0.929) but not at a younger age (mean age of 3.01±0.885). This finding warrants further investigation. Indeed, on the one hand, the way children with TD comprehend the world changes tremendously during the preschool years, and this directly influences how the typicality of gaze is estimated. Also, on the other hand, the symptoms of autism

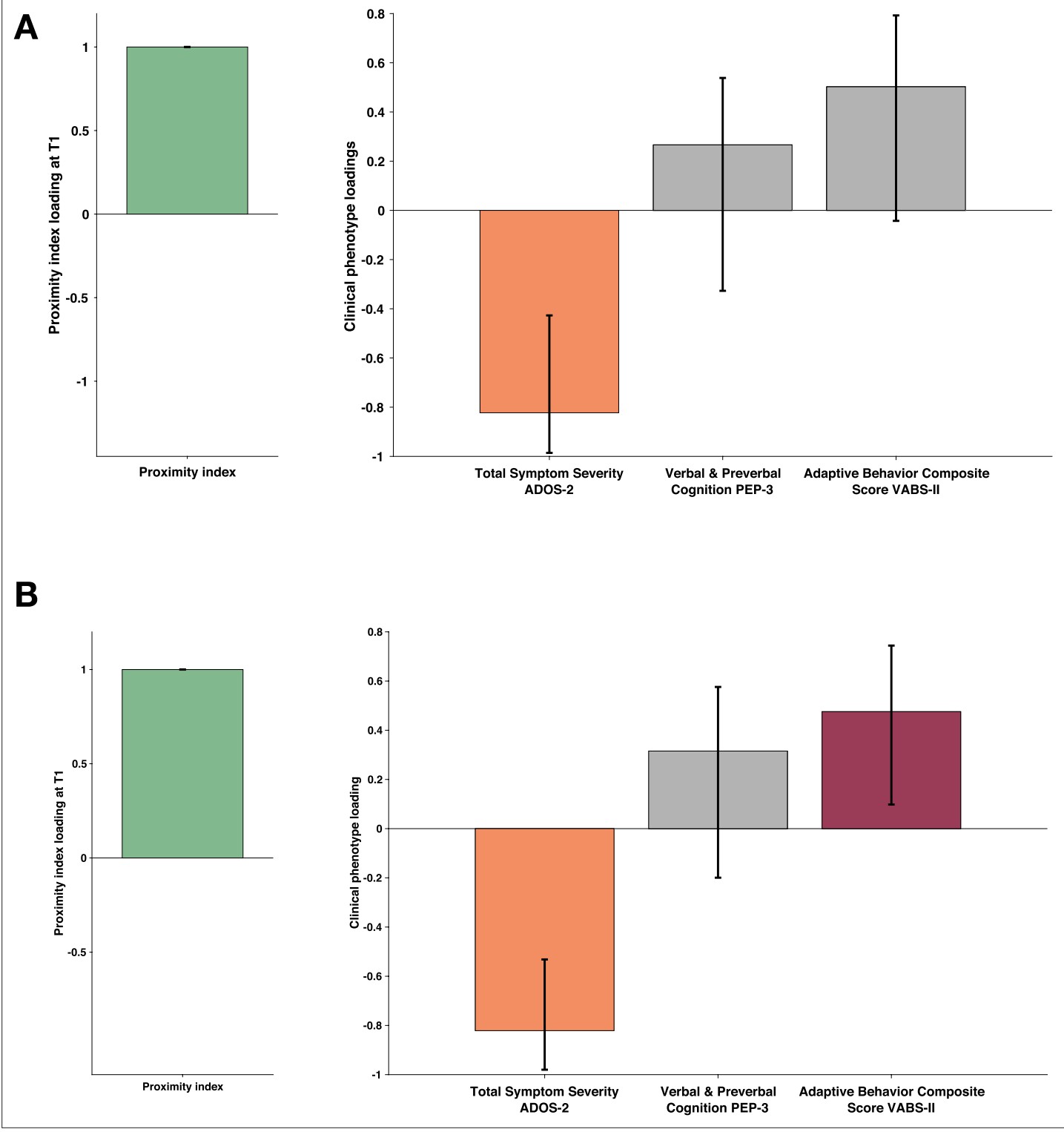

**Figure 7.** Proximity Index and its relation to behavioral phenotype in children with autism spectrum disorder (ASD) seen two times a year apart. Sample comprised 81 children with ASD who had valid eye-tracking recording and a complete set of behavioral phenotype measures a year after the baseline (T2). The PI for this paired longitudinal cohort was established using an age-matched reference group of 29 Typically Developing (TD) children. Loadings on the latent component were derived using PLS correlation analysis. The cross-correlation matrix included the Proximity Index (PI) on the imaging side A and three behavioral variables on the B side. The behavioral matrix accounted for two domains of autistic symptoms as assessed by ADOS-2, Verbal and Preverbal Cognition (VPC) from the PEP-3, and the Adaptive Behavior Composite Score from the VABS-II. Error bars represent the bootstrapping 5th to 95th percentiles. Results that were not robust are indicated by a gray boxplot color. (**A**) Proximity index (PI) obtained at T1

*Figure 7 continued on next page*

*Figure 7 continued*

and phenotype measures obtained a year later (T2). PI at T1 positively correlated with reduced symptoms at T2 (**B**) Simultaneous PLS-C: both PI and phenotype measures were obtained at T2. PI at T2 positively correlated with symptoms at T2 and positively with adaptive behavior. Loading on the latent component was obtained using the partial least squares correlation analysis. The cross-correlation matrix was composed of the proximity index-PI on the imaging **A** and three variables on the behavior **B** side. The behavior matrix included two domains of autistic symptoms assessed by ADOS-2, Verbal and preverbal cognition (VPC) of PEP-3, and the Adaptive Behavior Composite Score of VABS-II.

naturally change over the preschool years, and all these elements can be responsible for the effect we observe.

## Divergent developmental trajectories of visual exploration in children with ASD

After exploring the PI association with various aspects of the behavioral phenotype in ASD children, we were also interested in the developmental pathway of visual exploration in this complex social scene for both groups of children. Previous studies using cross-sectional designs have demonstrated important changes in how children attend to social stimuli depending on their age (*Frank et al., 2012*; *Helo et al., 2014*). As our initial sample spanned a relatively large age range (1.7–6.9 years), we wanted to obtain a more fine-grained insight into the developmental dynamic of visual exploration during the given period. To that end, when study-specific inclusion criteria were satisfied, we included longitudinal data from our participants who had a one-year and/or a two years follow-up visit (see Methods section). With the available 306 recordings for the ASD group and 105 for the TD group, we applied a sliding window approach (*Sandini et al., 2018*) (see Methods section). Our goal was to discern critical periods of change in the visual exploration of complex social scenes in ASD compared to the TD group. We opted for a sliding window approach considering its flexibility to derive a continuous trajectory of visual exploration and thereby capture such non-linear periods. The sliding window approach yielded a total of 59 age-matched partially overlapping windows for both groups covering the age range between 1.88–4.28 years (mean age of the window) (*Figure 8*, panel A illustrates the sliding window method).

We then estimated gaze dispersion on a group level across all 59 windows. Dispersion on a single frame was conceptualized as the mean pairwise distance between all gaze coordinates present on a given frame (*Figure 8*, panel B). Gaze dispersion was computed separately for ASD and TD. The measure of dispersion indicated an increasingly discordant pattern of visual exploration between groups during early childhood years. The significance of the difference in the gaze dispersion between two groups across age windows was tested using the permutation testing (see Methods section). The statistically significant difference (at the level of 0.05) in a window was indicated using color-filled circles and as can be appreciated from the *Figure 8*, panel C was observed in 46 consecutive windows out of 59 starting at the age of 2.5–4.3 (average age of the window). While the TD children showed more convergent visual exploration patterns as they got older, as revealed by progressively smaller values of dispersion (narrowing of focus), the opposite pattern was characterized by gaze deployment in children with ASD. From the age of 2 years up to the age of 4.3 years, this group showed a progressively discordant pattern of visual exploration (see *Figure 8*, panel C).

To ensure the robustness and validity of our findings, we addressed several potential confounding factors. These included differences in sample size TD (TD sample included 51 and ASD sample 166 children), the heterogeneity of ASD behavioral phenotypes, and the use of developmental age rather than chronological age in our sliding window approach. We adopted a sequential approach, first examining the impact of unequal sample sizes and then considering both sample size and phenotypic heterogeneity together. Additionally, we implemented a sliding window methodology using developmental age as the primary matching parameter (for a detailed description, see Appendix 5). Our results consistently reaffirmed our initial findings obtained when using chronologically age-matched samples. Specifically, when matched for both sample size and developmental age, children with ASD consistently demonstrated a greater degree of interindividual disparity across childhood years compared to TD children (Appendix 5, Panels D1-D2).

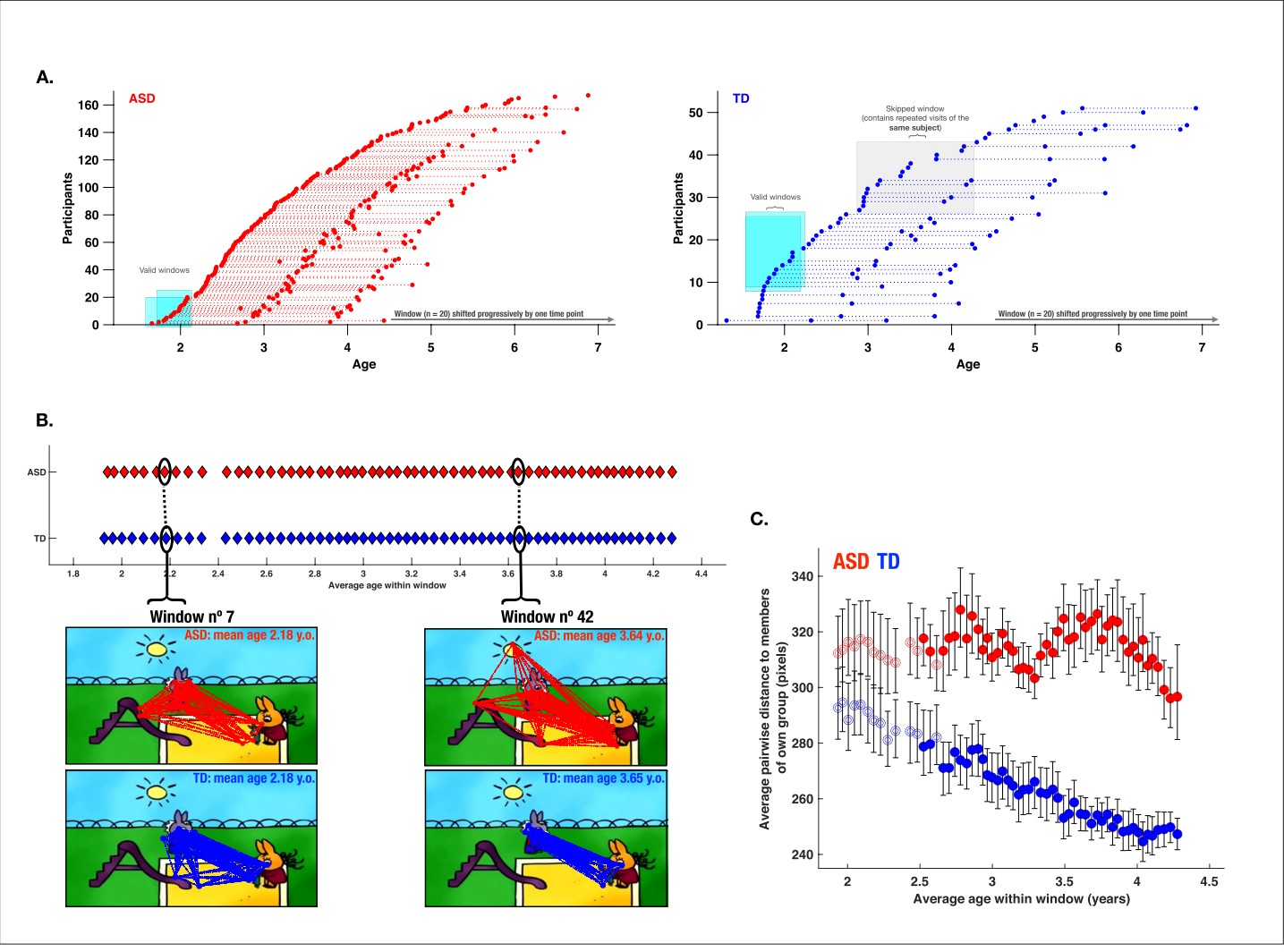

**Figure 8.** Characterization of the evolution of visual exploration patterns in young children with autism spectrum disorder (ASD) and the typically developing (TD) group using a sliding window approach. Panel **A**: The sliding window approach applied to the available recordings in our ASD group (red) and our TD group (blue); Panel **B**: gaze dispersion in two groups for the sliding windows n°7 and n°42 (mean age of windows 2.18 and 3.64 years, respectively); each circle represents a window encompassing 20 recordings; Panel **C**: Comparison of the gaze dispersion between two groups using Mean pairwise distance of gaze coordinates on each frame. The dispersion was calculated across 59 sliding windows spanning 1.88–4.28 years of age on average (here again, every circle represents a window encompassing 20 recordings). The windows with filled circles are those where a statistically significant difference between the two groups was shown using permutation testing. Error bars indicate a 95% confidence interval of the mean. As can be seen on panel **C**, dispersion values diminished in the TD group with advancing age, while the opposite pattern is observed in the ASD group showing a progressively more dispersed gaze behavior in the ASD group during childhood years.

## Discussion

In the present study, we used a data-driven method to quantify differences in spatio-temporal gaze patterns between children with ASD and their TD peers while watching an animated movie. Children with ASD who showed less moment-to-moment divergence in the exploration of a 3 min cartoon compared to referent gaze distribution of age-matched TD children had better adaptive functioning and better communication and motor skills. Visual exploration in the group of children with ASD was not better predicted by the low-level salience of the visual scene compared to their TD peers. Among various features of the video that children saw, the intensity of social content had the most important impact on divergence from the TD gaze patterns; children with ASD showed a more divergent deployment of attention on scene sequences with more than one character suggesting difficulties in processing social cues in the context of social interaction. On a larger temporal scale, across childhood years, the TD children showed a progressive tuning in the focus of their attention, reflected

by a narrowing of the group focus while the ASD group showed no such narrowing. Instead, their gaze patterns showed increasing dispersion over the same period. Of note, the children with ASD showing lower levels of divergence in gaze deployment compared to the age-matched TD group tended to have fewer symptoms of autism a year later.

Our results corroborate and extend the findings of a body of studies that have explored microstructural gaze dynamics in autism (*Avni et al., 2020*; *Nakano et al., 2010*; *Falck-Ytter et al., 2013b*; *Wang et al., 2018*) and have demonstrated divergent moment-to-moment gaze deployment in children with ASD compared to their age-matched TD peers. These processes are very important as any slight but systematic divergence in gaze deployment can have a tremendous influence on the experience-dependent brain specialization (*Johnson, 2001*; *Klin et al., 2009*). These subtle but relevant patterns might not be detected by methods focusing on macrostructural gaze structure measuring overall attention allocation on distinct visual features (e.g. faces, eyes, etc.) based on predefined areas of interest (AOI). Here, we extend the existing findings by first using a different data-driven methodology and, second, by including a developmental aspect to the spatiotemporal gaze deployment in autism and typical development. In our study, to define the referent gaze behavior, we present a novel index – the proximity index - that accounts for the entire scene, whether multiple socially relevant targets are present or just a few objects, and in doing so, provides a more subtle estimation of ASD gaze deployment in comparison to TD (see *Figure 1*). Furthermore, in this study, we used a cartoon, and thus a dynamic stream that is also more ecological in its representation of social interactions and has the advantage of being very appealing to young children. Previous research (*Riby and Hancock, 2009*) has shown that children with ASD attend more dynamic cartoon stimuli representing social interaction than when shown natural movies of people interacting. Despite animated movies being a simplified version of social interaction with reduced social complexity, the movie we analyzed provided us with ample insight into the atypicality of gaze behavior in children with ASD.

We showed that the level of divergence in gaze exploration of this 3 min video was correlated with ASD children's developmental level in children with ASD and their overall level of autonomy in various domains of everyday life. This finding stresses the importance of studying the subtlety of gaze deployment with respect to its downstream contribution to more divergent global behavioral patterns later in development (*Schultz, 2005*; *Young et al., 2009*; *Klin et al., 2015*; *Jones and Klin, 2013*). Gaze movements in a rich environment, as the cartoon used here, inform not only immediate perception but also future behavior as experience-dependent perception now is likely to alter the ongoing developmental trajectory. In accordance with this view, the level of typicality of visual exploration in ASD children at T1 was related to the level of autistic symptoms at T2 but not at T1. One possible interpretation of the lack of stable association at T1 might be due to the lower stability of symptoms early on. Indeed, while diagnoses of ASD show stability with age, still a certain percentage of children might show fluctuation. The study by Lord and collaborators *Lord et al., 2006* following 172 2-year-olds up to the age of 9 years old showed that diagnosis fluctuations are more likely in children with lesser symptoms compared to children with more severe symptoms. Still, as our study included all ASD severities, it is subject to such fluctuations. Another possible interpretation comes from the maturation of the gaze patterns in the TD group, against which we define the typicality of gaze in the ASD group. As can be seen in our results, children with TD show a tremendous synchronization of their gaze during the age range considered, resulting in a tighter gaze distribution at T2 and thus, a more sensitive evaluation of ASD gaze at that time point. The possibility that TD shows more similar gaze allocation with age, while ASD's gaze becomes increasingly idio-syncretic with age, highlights the value of addressing the mechanisms underlying the developmental trajectories of gaze allocation in future studies.

With regards to the exploration style, while watching the cartoon, compared to their TD peers, children with ASD presented more ambient, exploratory fixations, indicative of rapid acquisition of low-frequency information (*Eisenberg and Zacks, 2016*). On the other hand, they showed significantly fewer focal fixations that are known to operate with more fine-grained high-frequency information. This suggests that children with ASD spent more time than the TD group in an ambient mode trying to grasp the global scene configuration (*Ito et al., 2017*) and less in a detail-sensitive focused mode. These two modes of exploration are supported by distinct and yet functionally related systems of dorsal attention (ambient mode-related processing of spatial relations) and ventral attention (dealing with behaviorally salient object representation through the involvement of focused mode) (*Helo et al.,*

*2014*). Our finding of differential recruitment of these two modes during the viewing of social stimuli might suggest differential recruitment of these two attentional networks during the processing of these complex social scenes. In our previous work on a smaller sample for which we also acquired EEG recording during the time that children watched the Trotro cartoon, we found that the divergence in gaze deployment was related to the vast abnormalities in neural activation, including reduced activation of frontal and cingulate regions and increased activation of inferior parietal, temporal, and cerebellar regions (*Jan et al., 2019*). In a similar EEG-eye-tracking study using videos involving biological motion (children doing yoga in nature) (*Sperdin et al., 2018*), we found increased contribution from regions such as the median cingulate cortex and the paracentral lobule in the toddlers and preschoolers with ASD who had a more similar visual exploration pattern to their TD peers (higher PI). Thus, the children who showed less divergence from referent gaze patterns (TD-like viewing patterns) more actively engaged the median cingulate cortex and the paracentral regions suggesting potential compensatory strategies to account for the divergent brain development over time. Longitudinal studies combining eye-tracking and neuroimaging techniques are necessary to confirm the hypothesis of such compensatory hyperactivation.

In an effort to parse the complexity in gaze deployment evidenced in our ASD group across childhood years, we measured the contribution of basic visual properties of the scene to the gaze deployment in this group as compared to the TD group. We found that the basic visual properties played a less important role in directing gaze in our group of young children with ASD as compared to their TD peers. This was observed across all separate channels, namely, intensity, orientation, color, motion, and flicker, as well as the full salience mode with all channels combined. Previous research has shown that bottom-up features are responsible for directing attention in very young infants, but from 9 months of age, top-down processes take predominance in directing gaze (*Frank et al., 2009*). Less is known about the relative contribution of these processes while watching complex dynamic stimuli over the developmental span. Using a cross-sectional sample of TD children and adults *Rider et al., 2018* showed that gaze deployment in both children and adults was better predicted by the presence of a face in the scene (summoning top-down processing mechanisms) than by low-level visual properties of the scene. However, the two salience models they used (I&K and GBVS, the latter being the same as the one used in our study) were better at predicting gaze data in adults than in children suggesting that these dynamic salience models might be more adapted to the mature visual system. Indeed our sample is relatively young, and it is possible that the lesser success of the salience models to predict gaze allocation in ASD children might be influenced by the visual and motor abnormalities characterizing this age range (*Rider et al., 2018*; *Farber and Beteleva, 2005*).

Contrary to the bottom-up visual properties of the scenes, social intensity was an important element in governing the gaze divergence in children with ASD. The finding of a more divergent pattern in frames comprising the interaction between characters corroborates previous findings of atypical face (*Hanley et al., 2013*) and dynamic social stimuli (*Speer et al., 2007*) processing, particularly in the context of interaction (*Parish-Morris et al., 2019*). Social interaction processing depends strongly on the top-down inputs, as the choice of what is to be attended relies on prior expectations, attributed meaning, and global language and scene understanding. Here, our data show that ASD children most at risk on these skills also show lower, less TD-like PI.

The sliding window approach yielded a fine grained-measure of change in gaze deployment in both groups of children during early childhood. With advancing age, TD children showed increasingly coherent gaze patterns, corroborating previous findings of increased consistency in TD gaze behavior over time (*Frank et al., 2009*; *Shic et al., 2007*; *Franchak et al., 2016*; *Rider et al., 2018*; *Kirkorian et al., 2012*). On the other hand, children with ASD showed increasingly heterogeneous patterns during the same period. A similar contrasting pattern with gaze in TD individuals getting more stereotyped from childhood to adulthood and gaze in ASD groups showing more variability was brought forward in a study by *Nakano et al., 2010*. While this study used a cross-sectional design to study the developmental change in a group of children and adults, to our knowledge, our study is the first to extend the findings on both TD and those with ASD using a longitudinal design and focusing on a moment-to-moment gaze deployment. This higher consistency in gaze in the TD group with increasing age was put in relation to more systematic involvement of top-down processes (*Kirkorian et al., 2012*; *Franchak et al., 2016*; *Helo et al., 2017*). During typical development through the phylogenetically (*Rosa Salva et al., 2011*) favored mechanism of preferential orientation to social stimuli, children show

increasing experience with and subsequently increasing understanding of social cues setting them on the typical path of social development (*Klin et al., 2009*; *Jones and Klin, 2013*). On the other hand, strikingly divergent patterns in children with ASD might be seen as a product of the accumulation of atypical experiences triggered by social attention deployment diverging early on in their development (*Jones and Klin, 2013*). Behaviorally, in children with ASD during the preschool years, we observe the emergence of circumscribed interests alongside the tendency of more rigid patterns of behaviors (insistence on sameness) (*Richler et al., 2010*). These emerging patterns of interests might contribute to the divergence in gaze as attention is rather attracted to elements related to the circumscribed interests (*Sasson et al., 2008*; *Sasson et al., 2011*), thus amplifying the derailment from the referent social engagement path (*Klin et al., 2015*). Ultimately, interests that are, indeed, idiosyncratic in nature might limit group-level coherence; however, a discernible amount of within-subject stability in gaze patterns over shorter time scales may be expected. While the present study does not address the latter, our results highlight the loss of group cohesion in gaze as ASD children age in line with emerging findings of marked gaze in-consistency across individuals with ASD (*Nakano et al., 2010*; *Wang et al., 2018*). Whether, as shown by *Avni et al., 2020*, within-individuals consistency also decreases when the same video is seen twice is an important topic for future studies to address. Overall, our results are consistent with the presence of growing idiosyncrasy in the selection and processing of information, particularly in the context of social interaction in ASD. An increased idiosyncrasy on the neural level while watching dynamic social scenes has been put forward by a number of studies (*Hasson et al., 2009*; *Byrge et al., 2015*; *Bolton et al., 2018*; *Bolton et al., 2020*) and was related to lower scene understanding (*Byrge et al., 2015*) and higher presence of autistic symptoms (*Bolton et al., 2020*). The mechanisms of efficient selection of relevant social elements are genetically controlled (*Constantino et al., 2017*; *Kennedy et al., 2017*), and a disturbance we observe in ASD is most likely a downstream product of the gene-environment correlation (*Klin et al., 2015*). According to this view, the initial vulnerability (*Jones and Klin, 2013*; *Constantino et al., 2017*) characterizing autism would lead to a lifetime of atypical experiences with the social world, which in turn could result in atypical brain specialization and more idiosyncratic behavioral patterns.

The finding of progressive divergence in gaze patterns in children with ASD during the childhood years urges for early detection and early intensive intervention to prevent further derailment from the typical social engagement path (*Dawson et al., 2010*). The present study is one of the first to tackle microstructural atypicalities in gaze deployment in young children with ASD taking into account developmental change. Our longitudinal findings of the initial gazing divergence informativeness of the later autistic symptomatology reflect the potential of the present method as a promising tool for understanding the mechanisms of developmental change in ASD. This work stresses the need to better characterize the link between behavioral phenotypes and the underlying neurobiological substrates to adapt early intervention strategies to the neurodevelopmental mechanisms involved.

The current study comes with a number of limitations. The lack of a control group of comparable size to the ASD group was a severely limiting factor. The study protocol inside which the present work was realized, is rather dense, and longitudinal visits are spaced 6 months from each other, which asks for an important investment from families who would otherwise not need this highly precise assessment of the developmental functioning of their child. From the developmental perspective, a bigger TD sample would allow more precision in measurements of the developmental change with age. It would allow defining the referent groups that are tightly matched with regard to age and allow pure longitudinal measures. We tried our best to account for this by using a sliding window approach with partially overlapping windows in order to infer developmental dynamics in both groups over childhood years, but an ideal design would be purely longitudinal. A bigger TD sample would also allow more sophisticated analysis, such as unsupervised clustering to test the potential of the Proximity Index method for data-driven classification. Moreover, an important question to address is the development of gaze dynamics in girls with ASD. In the current study, we focused only on males, as the number of eligible females with ASD was much smaller. Finally, another important element that was out of the scope of the present study but that would warrant an in-deep investigation in this early post-diagnosis period is the role of the behavioral treatment children received after the diagnosis was established. Early intensive behavioral intervention greatly improves the symptoms and the functioning profile of the individuals on the spectrum. It would be important to learn how gaze behavior is influenced by such intervention, and how behavioral profile changes following the change in visual behavior.

**Table 1.** Description of the cross-sectional sample.

| Measures | ASD (n=166) Mean±SD | TD (n=51) Mean±SD | p-value |
|---|---|---|---|
| Age | 3.37±1.16 | 3.48±1.29 | 0.621[a] |
| Total Symptom Severity Score (ADOS-2 CSS) | 7.19±1.78 | 1.10±0.300 | <0.001[a] |
| Social Affect (ADOS-2 SA-CSS) | 6.08±2.06 | 1.18±0.478 | <0.001[a] |
| Repetitive Behaviors & Restricted Interests (ADOS-2 RRB CSS) | 8.63±1.85 | 2.16±1.92 | <0.001[a] |
| Social Interaction (ADI-R: A) | 14.8±5.70 | 1.04±1.39 | <0.001[a] |
| Communication (ADI-R: B) | 9.97±3.44 | 1.12±1.35 | <0.001[a] |
| Repetitive Behaviors & Restricted Interests (ADI-R: C) | 4.79±2.22 | 0.314±0.678 | <0.001[a] |
| Age of onset (ADI-R: D) | 3.60±0.997 | 0.078±0.337 | <0.001[a] |
| Best Estimate IQ | 83.6±24.0 | 119±16.5 | <0.001[a] |
| VABS-II Adaptive Behavior | 80.2±10.2 | 103±8.21 | <0.001 |
| VABS-II Communication | 80.2±13.7 | 105±8.94 | <0.001 |
| VABS-II Daily Living Skills | 83.7±11.6 | 101±8.25 | <0.001 |
| VABS-II Socialization | 79.2±9.82 | 101±8.49 | <0.001 |
| VABS-II Motor Skills | 88.4±11.5 | 102±11.2 | <0.001[a] |

Note. p-values[a] are obtained using nonparametric Mann-Whitney tests of differences between the two groups.

The method presented in the current study can easily be applied to any eye-tracking paradigm and any research question measuring the degree of similarity between any number of populations. It has the potential for application in population-wide studies for charting the developmental paths of visual exploration across the lifespan and is a promising tool for automated screening of children at risk of ASD.

## Materials and methods
### Experimental model and subject details
#### Cross-sectional sample
Hundred sixty-six males with autism (3.37 ±1.16 years) and 51 age-matched typically developing males (3.48 ±1.29 years) participated in the study. *Table 1* summarizes the clinical characteristics of our cross-sectional sample. Our study included only males due to fewer females with ASD. The clinical diagnosis of autism, based on DSM criteria, was confirmed using the standardized observational assessment of the child and interviews with caregivers(s) retracing the child's medical and developmental history. All children with ASD reached the cut-off for ASD on Autism Diagnostic Observation Schedule-Generic (ADOS-G), (*Lord et al., 2000*) or Autism Diagnostic Observation Schedule-2nd edition (ADOS-2) (*Lord et al., 2012*). For children who underwent the ADOS-G assessment, the scores were recoded according to the revised ADOS algorithm (*Gotham et al., 2007*) to ensure comparability with ADOS-2.

Before inclusion in the study, TD children were screened using a questionnaire focusing on medical history and history of pregnancy. Children were not included in our TD group if they were born prematurely or had a positive screen for the presence of any known neurological or psychiatric disorder in the child itself or known case of ASD in any first-degree relative of the child. Moreover, all TD children were also assessed using the ADOS-G or ADOS-2 evaluations to exclude the presence of ASD symptoms. The majority of TD participants had a minimal severity score of 1, except four children who had a score of 2.

The data for the current study were acquired as a part of a larger longitudinal study of early development in autism based in Geneva. Detailed information about cohort recruitment has been given elsewhere (*Franchini et al., 2017*; *Franchini et al., 2018*; *Kojovic et al., 2019*). The study protocol was approved by the Ethics Committee of the Faculty of Medicine of Geneva University, Switzerland (*Swissethics*, protocol 12–163/Psy 12–014, referral number PB_2016–01880). All families gave written informed consent to participate.

## Unstructured longitudinal sample

As participants in our study are followed longitudinally, their repeated visits were included when satisfying the inclusion criteria (later detailed in the Method details section). This yielded a total of 308 recordings for the ASD group and 105 for the TD group (all recordings were collected a year apart; 101 children with ASD contributed two recordings each, and 41 children with ASD contributed three recordings each, while 33 and 21 TD children contributed respectively 2 and 3 recordings each) (see *Figure 8* for illustration of the available recordings). This sample was employed to derive trajectories of visual exploration over the childhood years using mixed models analysis and considering both within-subject and between-subject effects (*Mutlu et al., 2013*; *Mancini et al., 2020*) and sliding windows approach (*Sandini et al., 2018*) (further detailed in the Method details subsection).

## One-year follow-up longitudinal sample

To obtain a longitudinal measure of change in visual exploration, we used a smaller subsample that included children who had recordings obtained a year apart. From the overall number of ASD children (101) that had two recordings, seven were removed as they were done two years apart. The same was done on the TD group, where four were removed. Thus, this final paired longitudinal sample included 94 males with ASD (1.66–5.43 years old) and 29 age-matched TD males (1.31–5.56 years old) who were evaluated a year later.

## Behavioral phenotype measures

As detailed above, a direct assessment of autistic symptoms was obtained using the Autism Diagnostic Observation Schedule-Generic ADOS-G, (*Lord et al., 2000*) or Autism Diagnostic Observation Schedule-2nd edition (ADOS-2) (*Lord et al., 2012*). Since its latest version (ADOS-2) the ADOS yields a measure of severity of autistic symptoms ranging from 1 to 10, conceived to be relatively independent of the participant's age or verbal functioning (*Gotham et al., 2009*; *Estes et al., 2015*). For subjects who were administered the older version of the ADOS (ADOS-G), the severity scores were obtained according to the revised ADOS algorithm (*Gotham et al., 2007*). For a more precise measure of symptoms according to their type, we included the domain severity scores, namely, social affect (SA) and restricted and repetitive behaviors (RRB) (*Hus et al., 2014*).

A detailed developmental history of symptom emergence and presentation was obtained using the Autism Diagnostic Interview-Revised (*Lord et al., 1994*). ADI-R is a standardized, semi-structured interview administered by trained clinicians to parents/caregivers. The ADI-R assesses the early developmental milestones and the present (last three months) and past behavior in the domains of reciprocal social interactions (A), communication (B), and restricted, repetitive, and stereotyped patterns of behavior (C). Being developed in the DSM-IV framework (*Association AP, 1994*) specific attention is given to the age of onset of symptoms (domain D, Demographics table).

In our large longitudinal autism cohort, the cognitive functioning of children is assessed using several assessments depending on the age of the children and their capacity to attend to the demands of cognitive tasks. Since the cohort conception in 2012, we used the Psycho-Educational Profile, third edition, PEP-3, (*Schopler, 2005*) validated for 24–83 months. In 2015 we added the Mullen Early Learning scales (*Mullen, 1995*) validated for 0–68 months. For the current study, in all analyses of the Results section, we used the scores obtained from the PEP-3 for the behavioral correlations with the PI. However, when we compared the group of ASD children with the TD children in the description of the sample at the beginning of this section, we faced a lot of missing data on the TD side, as a complete PEP-3 was frequently missing in children with TD (lack of time to complete several cognitive assessments). To be able to present a descriptive comparison between the two groups, in the Demographics table, and only there, we used the Best Estimate Intellectual Quotient, a composite measure obtained by combining available assessments as previously described in the literature (*Howlin et al.,*

*2014*; *Kojovic et al., 2019*; *Howlin et al., 2013*; *Bishop et al., 2015*; *Liu et al., 2008*). In the ASD group, the majority of children had the Psycho-Educational Profile, the third edition, Verbal/Preverbal Cognition scale (PEP-3; VPC DQ, *Schopler, 2005*) (n=154). The VPC Developmental Quotient (DQ) was obtained by dividing the age equivalent scores by the child's chronological age. For a smaller subset of children with ASD (below two years of age), as the PEP-3 could not be administered, we used Mullen Early Learning scales (*Mullen, 1995*), (n=10). Developmental quotients were obtained using the mean age equivalent scores from four cognitive scales of the MSEL (Visual Reception, Fine Motor, Receptive Language, and Expressive Language) and divided by chronological age. One child with ASD was administered only the Full-Scale IQ (FSIQ), Wechsler Preschool and Primary Scale of Intelligence, fourth edition (*David, 2014*), and one child was not testable at the initial visit (severe sensory stimulation). In the TD group, the majority of children were assessed using the MSEL (n=24), followed by PEP-3 n=23, and WPPSI-IV (n=4 children). The composite score comparison (BEIQ) is present in the Demographics table.

Adaptive functioning was assessed using the Vineland Adaptive Behavior Scales, second edition (VABS-II; *Sparrow et al., 2005*). VABS-II is a standardized parent interview measuring adaptive functioning from childhood to adulthood in communication, daily-living skills, socialization, and motor domain. The adaptive behavior composite score (ABCS), a global measure of an individual's adaptive functioning, is obtained by combining the four domain standardized scores.

## Method details
### Stimuli and apparatus
The current experiment consisted of free-viewing of one episode of the French cartoon 'Trotro' lasting 2'53" (*Lezoray, 2013*). This cartoon was the first stimulus in an experiment involving the simultaneous acquisition of High-density EEG recorded with a 129-channel Hydrocel Geodesic Sensor Net (Electrical Geodesics Inc, Eugene, OR, USA). The findings concerning the EEG data are published separately (*Jan et al., 2019*). This cartoon depicts human-like interactions between three donkey characters at a relatively slow pace. The original soundtrack was preserved during recording. Gaze data were collected using Tobii TX300 eye tracker (https://www.tobiipro.com), sampled at 300 Hz, except for five recordings acquired at a lower sampling frequency (60 Hz) using Tobii TXL60. The screen size was identical for both eye-tracking devices height: 1200 pixels (29°38') and width: 1920 pixels (45°53'), with a refresh rate of 60 Hz. Participants were seated at approximately 60 cm from the recording screen. The cartoon frames subtended a visual angle of 26°47' × 45°53' (height × width). A five-point calibration procedure consisting of child-friendly animations was performed using an inbuilt program in the Tobii system. Upon verification, the calibration procedure was repeated if the eye-tracking device failed to detect the participant's gaze position accurately. The testing room had no windows, and lighting conditions were constant for all acquisitions.

## Quantification and statistical analysis
### Eye-tracking analysis
We excluded data from participants who showed poor screen attendance, defined as binocular gaze detection on less than 65% of video frames. The screen attendance was higher in the TD sample (93.8 ±6.37 s) compared to the ASD group (87.8 ±9.33 s), $U$=2568, p<0.001. To extract fixations, we used the Tobii IV-T Fixation filter (*Olsen, 2012*) (i.e. Velocity threshold: 30° /s; Velocity window length: 20 ms. Adjacent fixations were merged Maximum time between fixations was 75 ms; Maximum angle between fixations was 0.5°). To account for differences in the screen attendance, we omitted instances of non-fixation data (saccades, blinks, off-screen moments) in all calculations.

### Determining the 'reference' of visual exploration
To define the referent gaze distribution ('reference'), against which we will compare the gaze data from the ASD group, we employed the kernel density distribution estimation function on gaze data from TD individuals on each frame of the video. The reference sample comprised 51 typically developing children (3.48 ±1.29 years). To create referent gaze distribution, we opted for a non-fixed bandwidth of a kernel as the gaze distribution characteristics vary significantly from frame to frame. Precisely, fixed bandwidth would result in over-smoothing the data in the mode and under-smoothing extreme distribution cases of gaze data at tails. We used the state-of-the-art adaptive kernel density estimation

that considers the data's local characteristics by employing adaptive kernel bandwidth (**Botev et al., 2010**). Thus a Gaussian kernel of an adaptive bandwidth was applied at each pair of gaze coordinates, and the results were summed up to obtain an estimation of the density of gaze data (see **Figure 1**). Obtained density estimation reflects a probability of gaze allocation at the given location of the visual scene for a given group. This probability is higher at the distribution's taller peaks (tightly packed kernels) and diminishes toward the edges. We used the Matlab inbuilt function *contour* to delimit isolines of the gaze density matrix.

## Quantifying the divergence in visual exploration

Upon the 'reference' definition, we calculated the distance of gaze data from this referent distribution on each frame for each child with ASD (n=166; 3.37 ±1.16 years). Comparison to this referent pattern yielded a measure of *Proximity index-PI* (see **Figure 1**). The calculation of the Proximity Index values was done for each frame separately. Proximity Index values were scaled from 0 to 1 at each frame for comparison and interpretation. We used the Matlab inbuilt function *contour* to delimit isolines of the gaze density matrix. To have a fine-grained measure, we defined 100 isolines per density matrix (i.e. each frame). Then we calculated the proximity index for each child with ASD framewise. Gaze coordinates that landed outside the polygon defined by contour(s) of the lowest level (1) obtained a PI value of 0. The gaze coordinates inside the area defined by gaze density matrix isolines obtained the PI value between 0.01 and 1. The exact value of these non-zero PI values was obtained depending on the level number of the highest isoline/contour that contained the x and y coordinates of the gaze. As we defined 100 isolines per density matrix, the levels ranged from 1 to 100. Accordingly, a gaze coordinate that landed inside the highest contour (level 100) obtained a PI value of 1, and the one that landed inside the isoline 50 obtained a PI value of 0.50. A high PI value (closer to the mode of the density distribution) indicates that the visual exploration of the individual for a given frame is less divergent from the reference (more TD-like). A summary measure of divergence in visual exploration from the TD group was obtained by averaging the PI values for the total duration of the video.

While the smoothing kernel deployed in our density estimation function is Gaussian, the final distribution of the gaze data is not assumed Gaussian. As shown in **Figure 1**, right upper panel, the final distribution was sensitive to the complexity of gaze distribution (e.g. having two or more distant gaze foci in the TD group) which allowed a flexible and ecological definition of referent gaze behavior. The coexistence of multiple foci allows for pondering the relative importance of the different scene elements from the point of view of the TD group. It further distinguishes our method from hypothesis-driven methods that measure aggregated fixation data in the scene's predefined regions. For the frames where the gaze of the TD group showed many distinct focal points, like the one in **Figure 1**, right upper panel, we calculated the PI in the same manner as for frames that had a unique focus distribution. For a given gaze coordinate from a child with ASD, we identify the level of the highest contour, ranging from 0.01 to 1, of any of the attention focus/clusters containing that coordinate. If we assume a hypothetical situation where the gaze data of the TD group are falling along two clusters identically (i.e. we obtain the density peaks of the same level/height), in this case, any two gaze coordinates that fall in the highest level of any of the peaks would obtain a PI value of 1.

## Multivariate association between gaze patterns and behavioral data

The relation between behavioral phenotype and Proximity index was tested using the multivariate approach, Partial Least squares PLS-C (**McIntosh and Lobaugh, 2004**; **Krishnan et al., 2011**), Matlab-implemented source code is publicly available on https://github.com/MIPLabCH/myPLS; **Zöller et al., 2019**. This analysis focuses on the relationship between the two matrices, A (p by b) and B (p by k), formally expressed as $R = B^T A$. Before computing the cross-correlation matrix R between A and B, both input elements are z-scored. As the correlation is not directional, the roles of A and B are symmetric, and the analyses focus on the shared information between the two. The cross-correlation matrix R was then decomposed using a singular value decomposition (SVD) according to the formula: $R = U\Delta V^T$. The two singular vectors U and V are denoted as saliences, where U represents the behavioral pattern that best characterizes the R and V corresponds to the Proximity index pattern that best characterizes R. Finally, original matrix A and B are projected on their own saliences yielding two latent variables $L_a = AV$ and $L_b = BU$. The PLS-C implements permutation testing to foster model generalization of the latent variables. Once a vector(s) of saliences is defined as generalized, its stability is

tested using the bootstrapping approach with replacement. In all the analyses in this paper, we implemented 1000 permutations and 1000 bootstrapping to test the significance of the LC and the stability of the vectors of saliences, respectively.

## Proximity Index with regards to the visual properties of the animated scene

### Pixel level salience

Previous research has put forward the enhanced sensitivity to the low-level (pixel-level) saliency properties in adults with ASD while watching static stimuli (*Wang et al., 2015*) compared to healthy controls. We were interested in whether any low-level visual properties would more significantly contribute to the gaze allocation in one of the groups.

To extract values of basic visual qualities of the scene, we used a salience model that has been extensively characterized in the literature (*Koch and Ullman, 1985*; *Itti et al., 1998*; *Itti and Koch, 2000*; *Itti et al., 2001*). We used the GBVS version of this model (*Harel et al., 2006*), (for source code see http://www.animaclock.com/harel/share/gbvs.php; *Pinoshino, 2022*; *Harel et al., 2006*; *Harel, 2022*). This model extracts features based on simulated neurons in the visual cortex: color contrast (red/green and blue/yellow), intensity contrast (light/dark), four orientation directions (0°, 45°, 90°, 135°), flicker (light offset and onset) and four motion energies (up, down, left, right) (*Itti et al., 1998*; *Itti and Koch, 2001*). The final saliency map results from the linear combination of these separate 'channels' (*Itti et al., 2001*) into a unique scalar saliency map that guides attention (see *Figure 5A* for the illustration of salience features obtained using GBVS model on a given frame). To disentangle the relative importance of the channels besides using the global conspicuity map, we also considered the channels taken separately (see Appendix 2).

Considering the heavy computational cost of these analyses, all computations were performed at the University of Geneva on the Baobab and Yggdrasil clusters.

### Movie characteristics

#### Social complexity

Furthermore, given the findings of the failure of ASD in allocating attention to social content (*Chita-Tegmark, 2016b*; *Frank et al., 2012*), we aimed to test the hypothesis that the Proximity Index values will be lower for the moments in the videos with enhanced social complexity, involving two or three characters compared to moments involving only one character (Appendix 3A). Note that, with an increasing number of characters, we recognize that the scene is inevitably richer in details, an issue we address by measuring visual and vocalization complexity.

#### Visual complexity

To measure visual complexity, we calculated the length of edges delimiting image elements (see *Figure 1*). Edge extraction was done on every image of the video using the Canny method (*Canny, 1986*) implemented in Matlab (version 2017 a; Mathworks, Natick, MA). This method finds edges by looking for the local maxima of the intensity gradient. The gradient is obtained using the derivative of a Gaussian filter and uses two thresholds to detect strong and weak edges. Weak edges are retained only if connected to strong edges, which makes this method relatively immune to noise (see Appendix 3B).

#### Vocal video aspects: Monologue and directed speech

Speech properties of the scenes were also analyzed, using the BORIS software (https://www.boris.unito.it/). We manually identified the moments where characters were vocalizing or speaking. Then we annotated the moments as a function of the social directness of the speech. In particular, we distinguished between monologue (characters thinking out loud or singing) and moments of socially directed speech (invitation to play and responses to invitations).

#### Coarse movie characteristics: Frame switching and moving background

Finally, to test how the global characteristics of video media influence gaze deployment, we focused on two movie features. The first feature, denoted as the 'Frame switch,' encompasses all instances in which the cartoon employs an abrupt frame transition using the hard-cut montage technique. To

represent this feature numerically, a feature vector was created. In this vector, the first frame following the switch is assigned a code of 1, while all other frames are coded as 0. This coding scheme effectively highlights the occurrence of these abrupt shot changes within the movie. Throughout the duration of the movie, this event type occurs 25 times (as indicated in *Figure 6*).

The feature labeled as the 'Moving background' pertains to moments when the cartoon's background moves in tandem with the characters, following their directional motion. We aimed to distinguish these segments from scenes featuring a static background, as the overall motion dynamics in these frames varied. The occurrence of a moving background is observable in 5 distinct sequences within the movie (as illustrated in *Figure 6*). Frames with a moving background were coded 1 yielding a binary feature vector.

## Maturational changes in visual exploration of complex social scene

### Sliding window approach

Besides understanding the behavioral correlates of atypical visual exploration in ASD, we wanted to characterize further the developmental pathway of visual exploration of the complex social scenes in both groups of children. We opted for a sliding window approach adapted from *Sandini et al., 2018* to delineate fine-grained changes in visual exploration on a group level. Available recordings from our unstructured longitudinal sample were first ordered according to the age in both groups separately. Then, for each group, a window encompassing 20 recordings was progressively moved, starting from the first 20 recordings in the youngest subjects until reaching the end of the recording span for both groups. The choice of window width was constrained by the sample size of our TD group. The longitudinal visits in our cohort are spaced a year from each other, and the choice of a bigger window would result in significant data loss in our group of TD children as the windows were skipped if they contained more than one recording from the same subject. The chosen window width yielded 59 sliding windows in both groups that were age-matched and spanned the period from 1.88 to 4.28 years old on average.

Upon the creation of sliding windows and to characterize each group's visual behavior and its change with age, gaze data from the TD group were pooled together to define the TD distribution in each of the 59 age windows. To characterize the group visual behavior in the ASD group, we performed the same by pooling the gaze data together from the ASD group in each of the 59 age windows (see *Figure 8A and B*). We calculated the mean pairwise distance between all gaze coordinates on every frame for the measure of gaze dispersion in each of the two groups. Then we compared the relative gaze dispersion between groups on the estimated gaze density of each group in each age window separately.

To quantify the heteroscedasticity between groups across different ages, we computed the difference in dispersion (mean pairwise distance to members of own group), denoted as (disp_t(ATD) - disp_t(ASD)), for each time window (t). Then, the permutation method was used in order to get the distribution under the null hypothesis in each window (t) (H0: disp_t(TD) - disp_t(ASD)=0). Thus, for each window (59) 100 permutations (i) were performed (i.e. individuals were mixed up randomly in each group) and then we computed our statistic (disp_ti(TD) - disp_ti(ASD)) for each permuted sample (i) and each time window (t). The hundred statistics per window thus formed a null distribution (the expected behavior of our statistic under the null hypothesis) against which we could compare the 'real' statistic estimated in the original sample. The p-value is the probability of getting a statistic at least as extreme as the one we observed in our sample if we consider H0 to be the truth. The windows where the dispersion values showed statistically significant differences between the two groups are graphically presented with color-filled circles (*Figure 8C*).

## Acknowledgements

We express our utmost gratitude to all families participating in this study. We thank the clinical team for their immense investment in the data collection. Funding Funding for this study was provided by the National Centre of Competence in Research (NCCR) Synapsy, financed by the Swiss National Science Foundation-SNF (Grant No. 51NF40–185897), by SNF grants to MS (#163859, #190084, #202235 & #212653), ERC Synergy fund BrainPlay - The Self-teaching Brain grant to DB #810580, the Fondation Privée des Hôpitaux Universitaires de Genève (https://www.fondationhug.org), and by the Fondation Pôle Autisme (https://www.pole-autisme.ch).

# Additional information

## Funding

| Funder | Grant reference number | Author |
|---|---|---|
| National Centre of Competence in Research (NCCR) SYNAPSY | 51NF40-185897 | Marie Schaer |
| Swiss National Science Foundation | 163859 | Marie Schaer |
| Swiss National Science Foundation | 190084 | Marie Schaer |
| Swiss National Science Foundation | 202235 | Marie Schaer |
| Swiss National Science Foundation | 212653 | Marie Schaer |
| ERC Synergy fund BrainPlay - The Self-teaching Brain grant | 810580 | Daphné Bavelier |
| Fondation Privée des Hôpitaux Universitaires de Genève (https://www.fondationhug.org | | Marie Schaer |
| Fondation Pôle Autisme | | Marie Schaer |

The funders had no role in study design, data collection and interpretation, or the decision to submit the work for publication.

## Author contributions

Nada Kojovic, Conceptualization, Data curation, Software, Formal analysis, Validation, Investigation, Visualization, Methodology, Writing – original draft, Writing – review and editing; Sezen Cekic, Santiago Herce Castañón, Corrado Sandini, Daniela Zöller, Methodology, Writing – review and editing; Martina Franchini, Holger Franz Sperdin, Reem Kais Jan, Investigation, Writing – review and editing; Lylia Ben Hadid, Data curation; Daphné Bavelier, Conceptualization, Supervision, Writing – review and editing; Marie Schaer, Conceptualization, Resources, Supervision, Funding acquisition, Project administration, Writing – review and editing

## Author ORCIDs

Nada Kojovic [ORCID] https://orcid.org/0000-0003-0116-2485
Holger Franz Sperdin [ORCID] http://orcid.org/0000-0002-3438-1572
Corrado Sandini [ORCID] http://orcid.org/0000-0003-2933-1607
Reem Kais Jan [ORCID] http://orcid.org/0000-0002-1685-5594
Daniela Zöller [ORCID] http://orcid.org/0000-0002-7049-0696

## Ethics

The study protocol was approved by the Ethics Committee of the Faculty of Medicine of Geneva University, Switzerland (Swissethics, protocol 12-163/Psy 12-014, referral number PB_2016-01880). All families gave written informed consent to participate.

## Decision letter and Author response

Decision letter https://doi.org/10.7554/eLife.85623.sa1
Author response https://doi.org/10.7554/eLife.85623.sa2

# Additional files

## Supplementary files
• MDAR checklist

## Data availability

The Proximity Index method code and example data are publicly available at https://github.com/nadakojovic/ProximityIndexMethod (https://doi.org/10.5281/zenodo.10409645) and the data and codes used to produce figures of the current paper can be accessed at https://github.com/nadakojovic/ProximityIndexPaper (https://doi.org/10.5281/zenodo.10409651).

The following datasets were generated:

| Author(s) | Year | Dataset title | Dataset URL | Database and Identifier |
|---|---|---|---|---|
| Kojovic N | 2023 | nadakojovic/ProximityIndexMethod: ProximityIndexMethod:data&code | https://doi.org/10.5281/zenodo.10409645 | Zenodo, 10.5281/zenodo.10409645 |
| Kojovic N | 2023 | nadakojovic/ProximityIndexPaper: ProximityIndexPaper:data&code | https://doi.org/10.5281/zenodo.10409651 | Zenodo, 10.5281/zenodo.10409651 |

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

## Appendix 1

### Stability of the normative gaze distribution using simulated samples of varying size

The sample of 51 TD children whose gaze data was used to obtain a referent gaze distribution was a convenience sample. In the present study, we only included males due to the fewer number of females with ASD. Having this unique sample of TD children, we tested the stability of the referent distribution depending on the sample size by performing bootstrap analyses. Thus, from the available sample of 51 TD children, we performed 500 bootstraps, starting with a sample size of 10 until reaching the sample size of 50. To measure the change in gaze distribution on one frame, we calculated the average pairwise distance between all gaze coordinates available on the frame. Then for each frame, we calculated the variance of the average pairwise distance over 500 resamples. Finally, the variance obtained was averaged over the 5150 frames to yield a unique value of the variance in gaze patterns per sample size (10-50). Then we calculated the 'cutoff,' as defined by a sample size increase no longer yielding significant variation in the average variance. This was done using the *kneed* package implemented in Python that estimates the point of maximal curvature (elbow in curves with positive concavity) in discrete data sets based on the mathematical definition of curvature for continuous functions (*Satopaa et al., 2011*) (see *Figure 1*). The elbow of the fitted curve on our bootstrapping data was found at 18, meaning that the distribution was estimated to be stable from a sample size of 18.

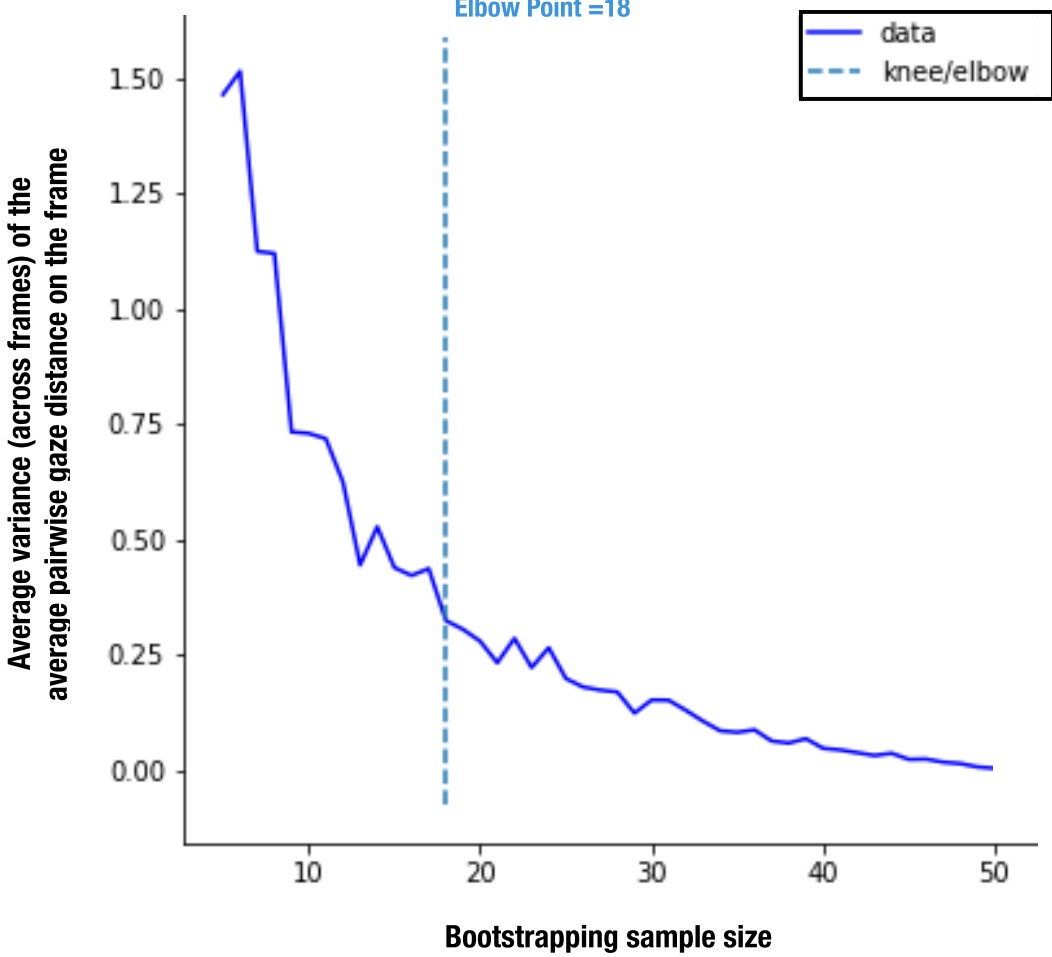

**Appendix 1—figure 1.** Stability of the normative distribution regarding the normative sample size. The continuous function was estimated using a kneed Python package using the average variance (over 5150 frames) of average (over 500 bootstrapped samples without replacement) mean pairwise distance of gaze coordinates on the frame (y-axis) for samples sizes ranging from 10 to 50 (x-axis) as the input: elbow point = 18.

## Appendix 2

## Basic visual properties of a scene: Prediction of the gaze allocation across individual salience channels

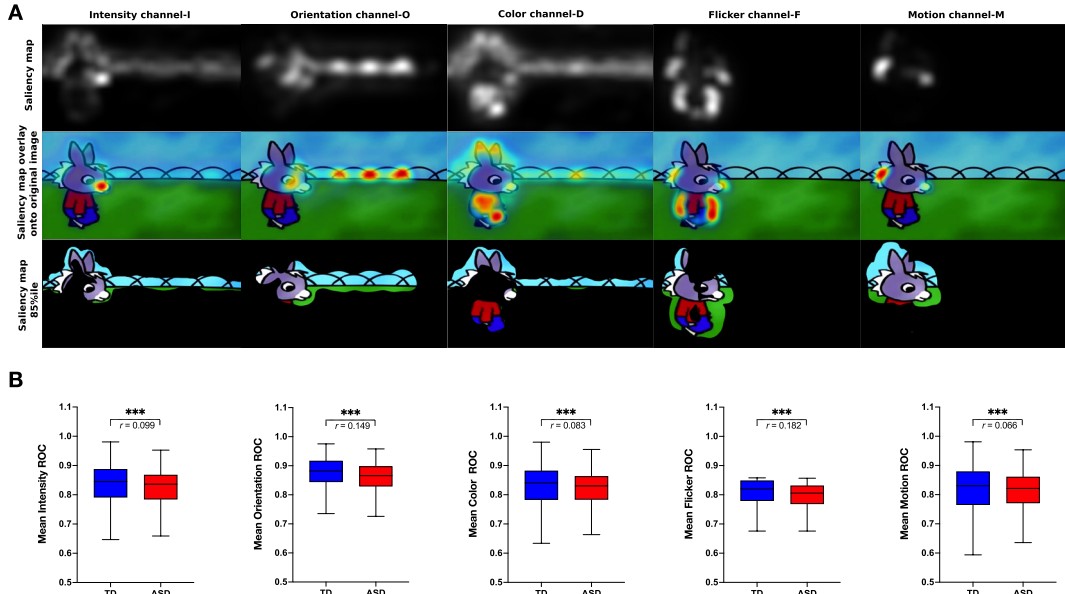

**Appendix 2—figure 1.** Visual salience group differences across channels. (**A**) From left to right: full saliency model with all five channels combined and channels taken separately: I-intensity, O-orientation, D-color, F-flicker, and M-motion channel. From top to bottom: Saliency map extracted for a given frame, Saliency map overlay on the original image, Original image with 15% most salient parts shown. **B**.

## Appendix 3

### Social and visual scene complexity

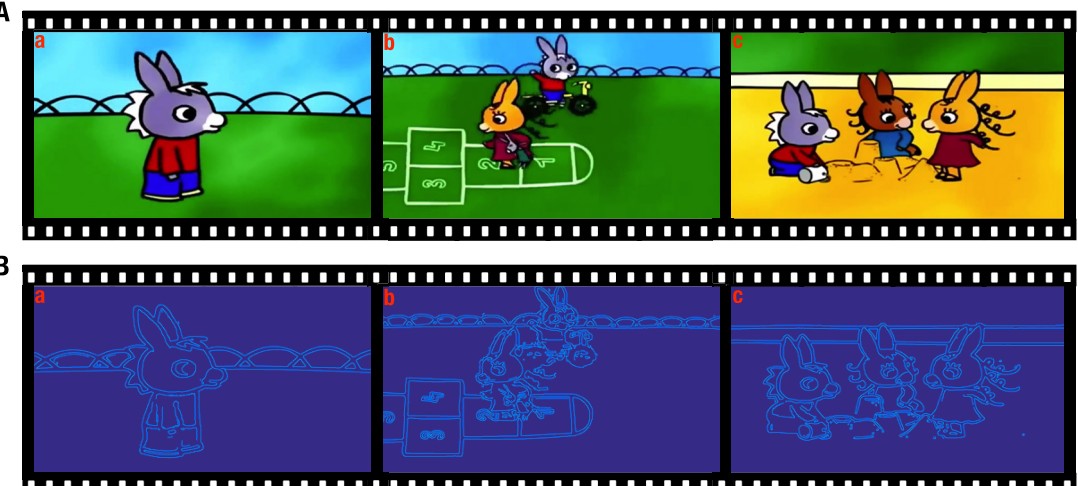

**Appendix 3—figure 1.** Illustration of the measures of social intensity and visual complexity. (**A**) Three frames (denoted as a, b, c) illustrate three levels of social intensity; (**B**) Visual complexity depicted using the edges of the images detected using the Canny method (*Canny, 1986*) for the frames a, b, and c.

# Appendix 4

## Relation between the PI and behavioral phenotype in a paired longitudinal subsample at the first time point (T1)

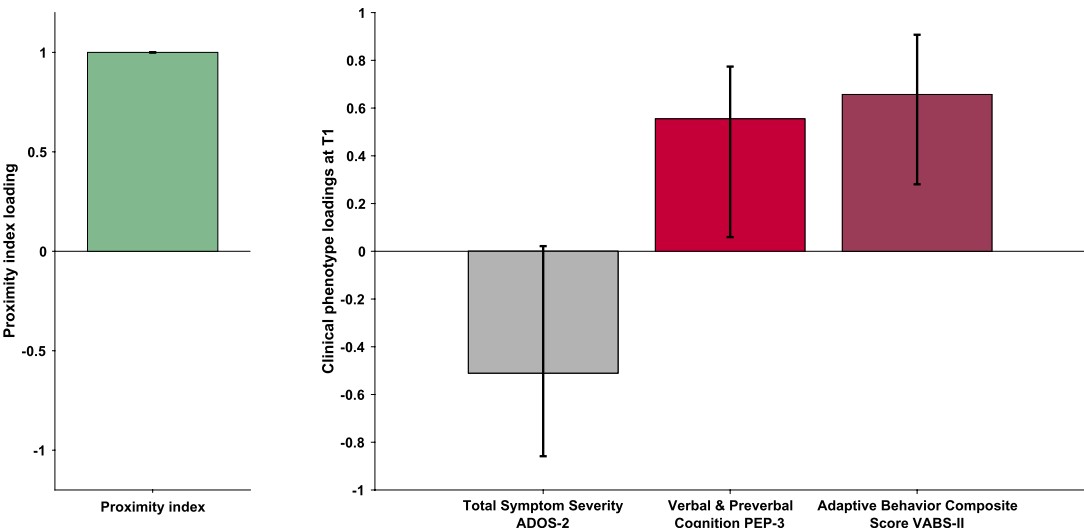

**Appendix 4—figure 1.** Proximity Index and its relation to behavioral phenotype in children with autism spectrum disorder (ASD) who were seen two times a year apart (the current figure depicts the initial (T1) visit). Sample comprised 81 children with ASD who had valid eye-tracking recording and a complete set of behavioral phenotype measures a year after the baseline (T2). The PI for this paired longitudinal cohort was established using an age-matched reference group of 29 Typically Developing (TD) children. PI was obtained at T1 and its correlation with the behavioral phenotype measures was assessed at the same time (T1). Loadings on the latent component were derived using PLS correlation analysis. The cross-correlation matrix included the Proximity Index (PI) on the imaging side A and three behavioral variables B. The behavioral matrix accounted for two domains of autistic symptoms as assessed by ADOS-2, Verbal and Preverbal Cognition (VPC) from the PEP-3, and the Adaptive Behavior Composite Score from the VABS-II. Error bars represent the bootstrapping 5th to 95th percentiles. Results that were not robust are indicated by a gray boxplot color. PI at baseline was positively correlated with developmental and adaptive functionng at baseline.

# Appendix 5

## Exploring confounding factors in the sliding window analysis of maturation in visual exploration within complex social scenes

Size

In response to the disparity in sample size between our two groups (51 TD and 166 ASD children), we implemented a methodology to mitigate the influence of this factor. We generated 100 bootstrapped ASD samples (without replacement), each with a size identical to that of the TD (51 subjects). These ASD samples were matched to the TD sample in terms of chronological age. Subsequently, for each of the bootstrapped samples, we aggregated all longitudinal data and computed the dispersion measure over time, akin to the process described in *Figure 8*, panel C. As illustrated in Panel A below, the results reveal that the bootstrapped ASD samples, characterized by both size and chronological age alignment with the TD group, exhibit higher levels of dispersion across the span of childhood years. This is in contrast to TD children, who exhibit a discernible pattern of progressive refinement in their visual exploration behavior.

It's worth noting that, while permutation testing could have been an ideal method for assessing the statistical significance of the findings in this section, we opted not to implement it due to the substantial computational cost associated with our analyses. The computational demands of our study necessitated an alternative approach to address the sample size and age-matching issue effectively. Consequently, we relied on the bootstrapping technique to provide valuable insights into the dispersion differences between the TD and ASD groups, while acknowledging the limitations imposed by the computational constraints.

Phenotypic heterogeneity. To address the considerable developmental heterogeneity inherent in the ASD group, we decided to repeat the analyses in the subsamples of a more restricted range of developmental functioning. Thus we derived 100 simulated samples of the same size as the TD group (51) firstly within the normal developmental range (DQ above 80) and then, we performed the same for the lower-functioning individuals with ASD (DQ below 80). As shown in Panels B-C below, both groups show sustained dispersion over the childhood years, in contrast to the convergence seen in the TD group. This trend is particularly pronounced in the subset of individuals with lower developmental functioning (Panel C), wherein a discernible divergence becomes increasingly evident during the preschool years.

Developmental age. As a final step, to comprehensively address the question of the difference in developmental age between our TD and ASD sample, we implemented a sliding window approach using our cross-sectional sample (51 TD and 166 ASD children). However, in this approach, we utilized developmental age for creating age-matched windows instead of chronological age as previously used. We initiated the process with the first 20 recordings from subjects with the lowest developmental age and progressively shifted a window encompassing 20 recordings. This continued until the entire range of recordings for both groups was covered. Similar to the method applied in the main part of the manuscript, we excluded windows containing duplicate recordings from the same subject. This method yielded a total of 60 windows, each matched based on age, with developmental age in the ASD group and chronological age in the TD group Panel D1. To test the stability of our findings and assess the potential influence of sample size, we replicated the sliding window procedure using 100 bootstrapped ASD samples, each comprising 51 subjects whose developmental age was matched to the chronological age of the TD subjects. For the purpose of interpretation, we plotted a linear regression line (in red) for each bootstrapped sample Panel D2. Our results reinforce our initial findings when using chronological age-matched samples *Figure 8*. Children with ASD consistently exhibit a greater degree of interindividual disparity across childhood years, in contrast to TD children. This outcome underscores our findings' robustness and strengthens our observations' validity.

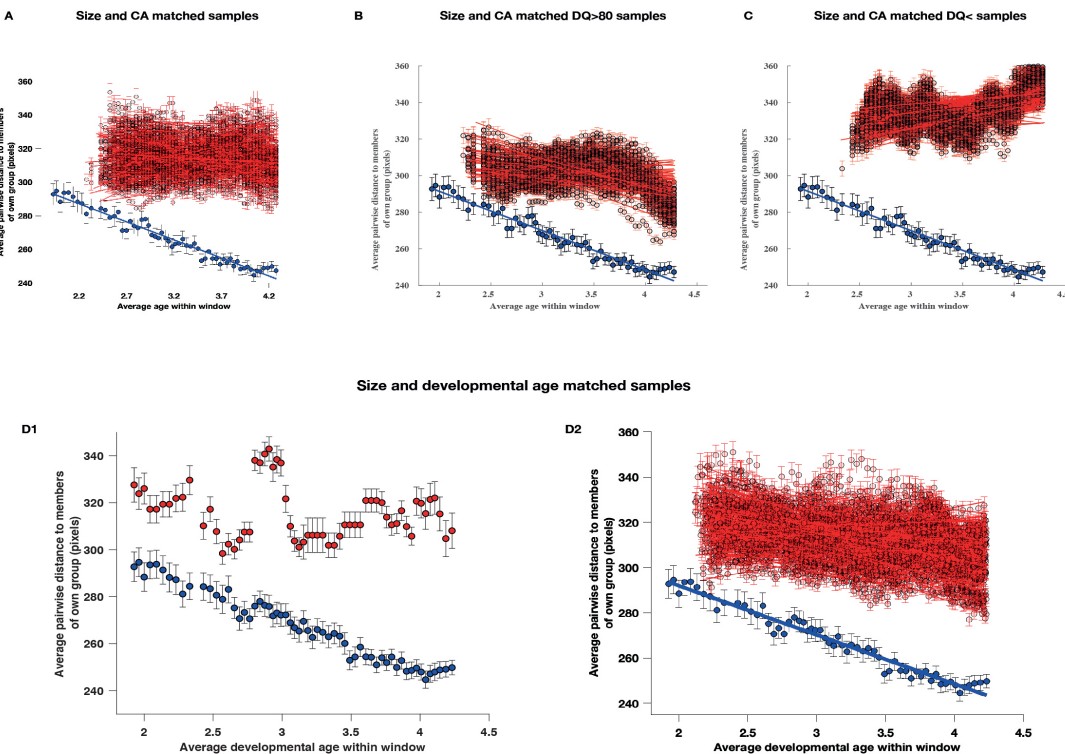

**Appendix 5—figure 1.** Evolution of visual exploration patterns in young children with autism spectrum disorder (ASD) and the typically developing (TD) group using a sliding window and bootstrapping approach. The dispersion in 100 bootstrapped samples of ASD recordings is given in red and the original group dispersion in the TD group is shown in blue. Panel A: ASD bootstrapped samples are matched to the TD group with regards to size (n=51) and chronological age; Panel B: ASD bootstrapped samples are matched to the TD group with regards to size (n=51), chronological age and have the DQ within the normal range (above 80); Panel C: ASD bootstrapped samples are matched to the TD group with regards to size (n=51), and chronological age and have the DQ below the normal range (below 80); Panel D1: Evolution of visual exploration patterns in young children with ASD whose developmental age was matched to the chronological age of the TD group using a sliding window approach. Comparison of the gaze dispersion between two groups using Mean pairwise distance of gaze coordinates on each frame. The dispersion was calculated across 60 sliding windows spanning 2.9–4.3 years of mental age on average (every circle represents a window encompassing 20 recordings); Panel D2: The sliding window approach was applied to the ASD bootstrapped samples that are matched to the TD group with regards to size (n=51) while mental age was aligned with the chronological age of the TD group.

