## [Editor Report]

This is an important study investigating a rare longitudinal dataset of eye-tracking to a cartoon video, measured in a group of children with autism and a control group that is typically developing. The core finding is a divergence in exploratory gaze onto the video stimulus in the children with ASD, compared to typically developing children, this finding is supported by convincing evidence. In addition, the effect appeared to be parametric: those autistic children with the least divergence also had the best adaptive functioning and communication skills. Additional strengths of the study are a relatively large sample size for this type of work and analyses that aim at generalizability. This study will be interesting for autism specialists, but also for a wider community interested in social cognitive, affective neuroscience, and developmental disorders.

---

## [Decision Letter]

**Decision letter after peer review:**

[Editors’ note: the authors submitted for reconsideration following the decision after peer review. What follows is the decision letter after the first round of review.]

Thank you for submitting your work entitled "Unraveling the Developmental Dynamic of Visual Exploration of Social Interactions in Autism" for consideration by *eLife*. Your article has been reviewed by 3 peer reviewers, and the evaluation has been overseen by a Reviewing Editor and a Senior Editor. The following individuals involved in review of your submission have agreed to reveal their identity: Ralph Adolphs (Reviewer #2); Julia Yurkovic (Reviewer #3).

Comments to the Authors:

We are sorry to say that, after consultation with the reviewers, we have decided that your work will not be considered further for publication by *eLife*.

The reviewers agreed that while the study is conceptually well motivated, there is some lack of clarity regarding the adequacy of the TD control group. In addition, while the longitudinal component was noted as novel, individual differences were not sufficiently considered, and mechanistic insights obtained were deemed to be moderate.

We have also prepared an Evaluation Summary and Public Reviews of your work below, which are designed to transform your manuscript into a preprint with peer reviews.

*Reviewer #1 (Recommendations for the Authors)*

In the introduction, it would be helpful if the authors mention the age range and functioning level of participants for every prior study they cite that used eye-tracking to assess social attention in autism. In addition, statements such as "these atypicalities are present early" should include what is meant by early (what age range).

35 age-matched TD children were used to obtain normative gaze patterns, but it is not clear how this number was determined. Are there either power analyses or previous data suggesting that this sample size is sufficient to determine what is normative? A related issue is that of age norms. How were age-specific norms considered, given that the sample include children between 2-7 years of age and was not very large (eg. it is not clear how many children of each age were included in the sample)?

In addition to child adaptive behavior, were any other assessments conducted and if so, were they related to visual exploration patterns in children? Were any of the visual exploration metrics related to severity of autism symptoms?

There is not much information provided regarding the TD sample. What screening tools and procedures were used to detect neurological or psychiatric diseases?

*Reviewer #2 (Recommendations for the Authors)*

The main novelty of the study lies in a longitudinal component. Some of the subjects were tested again, about a year apart. Using a sliding window approach, these data were used to derive continuous measures of gaze atypicality over age (this analysis was complemented by a more rigid analyses based only on those subjects with two datapoints one year apart). Median contour surface and convergence index were the two metrics used here. Both showed developmental differences between groups. Visual features in the video were examined using a saliency model, and social complexity was quantified as the number of characters on the screen while controlling for overall visual complexity. Overall, it was found that while spatial divergence in gaze decreased in the TD group (convergence index increased) over the developmental time window examined, this was not the case for the ASD group (Figure 3).

These are labor-intensive and valuable data that will certainly add to the literature on attention and visual processing in ASD. The longitudinal dataset is quite special, and the results are in broad strokes fairly compelling: there are different developmental trajectories in visual attention in ASD compared to TD controls. The methods are reasonably sophisticated (but see my comments below). My main hesitation with this study is that it stops short of providing a stronger mechanistic advance. In particular, the developmental eye tracking data are really the centerpiece of the work. However, they are not linked to any other developmental measures. Actually, it was unclear to me at which timepoints the other behavioral measures (Vineland, etc.) were measured. It would have been ideal to get the Vineland and other measures of functioning at each of the specific timepoints, one year apart, at which the eye tracking data were collected to examine associations. But that seems not do have been done (or at least is not reported).

A second shortcoming of the study is the lack of detail with respect to individual subjects (were there any clusters/subgroups/outliers of interest?) and with respect to stimulus features that could be driving the observed effects. There is the saliency analysis, and an analysis of how many characters are on the screen, but that is it. Surely one could quantify additional semantic features, even if ratings are obtained from adults. As it stands there are unconnected analyses in the study. For instance, I have the bottom of Figure 1, showing that proximity index in a subject with ASD varies tremendously over the frames of the movie. Does it vary like that in other subjects with ASD? Does this variation line up with specific features in the video? Most importantly, I would want to see the framewise plot like this (or better: slightly temporally smoothed) of proximity index differences between TD and ASD, as a function of the longitudinal data. What features in the video correlate with the increased convergence seen in TD that are missing in ASD? The analyses provided do not back out these specifics, and without them the story is more descriptive than mechanistic.

1. Since the TD group was always used as a reference, and single ASD participants were compared to this "norm", we do not have a good estimate of how TD subjects would look by comparison. A stronger approach would calculate the TD norm in a leave-one-out fashion and generate distributions for how each individual TD subject also compared – those are the data then to compare (single TD and single ASD) rather than just ASD individuals to one fixed TD average. The authors refer to their analyses as "data-driven", and it is perhaps that with respect to not using ROIs on the video, but they are testing specific differences between pre-defined groups (ASD and TD). It would substantially strengthen the paper if indeed a data-driven analyses were provided (clustering the subject groups on the basis of the data, rather than as predefined). This could also help reveal possible outliers (in both groups) as well as possible subgroups.

2. It is intriguing that the fixation durations between groups differed. I wonder if the authors would consider inspecting smooth pursuit, a type of oculomotor feature not mentioned but that could be relevant (and has been reported to be atypical in ASD).

3. Subjects: More information on the subjects is needed. On what criteria were ASD and TD matched, was level of functioning or intellect taken into consideration? Did ASD subjects have a DSM diagnosis? Was the ADI done? What exclusionary criteria were applied (epilepsy, comorbidity, medication, etc?). More info please.

4. Analysis: proximity index. Why was the proximity index not temporally smoothed? A frame-by-frame metric will (a) have relatively sparse normative distributions, and (b) show fairly discontinuous gaze proximity (as evident in the plot shown in Figure 1, bottom).

5. All of the analyses are within-sample and use standard parametric statistics; it would be preferable to use cross-validation together with permutation testing for a more robust approach. For correlations between gaze data and behavioral data, it seems that about 10 correlation analyses were done. It would be important to correct for the multiple tests. This is also a particularly problematic issue in the saliency analyses, where there are several that are barely at the magic "P<0.05" threshold.

6. Eyetracking exclusions. This is insufficiently described in the paper. We are only told that subjects were excluded if >45% of frames were dropped. First of all, this is an extremely lenient threshold. But we need to know what the distribution of dropped frames was between ASD and TD groups. We also need to know what other exclusions were applied to any portion of the data. Right now, only complete subject-wise exclusions are mentioned, but surely that was not the only criterion.

*Reviewer #3 (Recommendations for the Authors)*

These results are interesting and valuable to our understanding of the development of social visual attention. However, several weaknesses should be addressed that would strengthen the results. First, the authors assume Gaussian distribution of TD eye gaze, but some of their example figures show that this is not always the case. This may lead to lower proximity index scores and may inflate the significant results rather than reflecting the true proximity of gaze to the normative distribution. Second, it would be helpful to discuss more of the individual differences in normative viewing to help anchor some of the main points of the paper. Finally, causality is assumed in the lower-level visual saliency analyses when instead the social and visual saliency may be highly correlated (and inseparable).

It would be helpful if the authors could provide more details about the π scores, specifically the normalization of them. The authors should explain how the normalization of the π scores was conducted and if this normalization allows for consistency across frames, where the possible furthest distance from the mode of the Gaussian distribution may change depending on the x- and y-coordinates of the mode of the Gaussian on the screen. Additionally, a description of how the normalization of π scores may change based on the convergence of TD children (i.e., how peaked the distribution is) would be helpful. If and how these measures may be limited should be discussed. Additionally, Figure 1 shows a child's π score on a frame-by-frame level across the video. Frames where the child was looking offscreen were coded as -0.15. The authors should explain why this value was chosen, and why a value was chosen at all instead of excluding these frames from analysis. Additionally, it would be important to know that these frames are moments that the child is looking off-screen and not moments where the child is blinking.

It is unclear how the authors handled instances where there were two distinct clusters of gaze distribution and the distribution was therefore not Gaussian. This will directly impact the π score and may make children with ASD look more atypical in their viewing patterns than they truly are. For example, Figure 3b shows both groups having two distinct clusters of visual attention, but more children with ASD are attending to the second focal point than TD children. Additional information on these instances should be added, and limitations should be discussed.

It would be helpful if the authors could validate their normative gaze distribution with leave-one-out procedures or some other method to ensure that the normative distribution is not shifted by one participant. This same procedure would be necessary for the maturational sliding window to show that the gaze pattern is actually reflecting developmental change, not just change due to individual differences in two participants' gaze data (the participant newly included and the participant newly excluded in the sliding window).

It would be worthwhile to include a figure of the correlation of π and autism symptom severity in Figure 2.

In Figure 3c-d, what do the shaded regions represent?

Mean and standard errors of the Proximity Index were not reported for every comparison Figure 4. This could also be collapsed to a difference score for each participant to allow easier comparisons across figures. This particular analysis is complex and would benefit from some greater explanation of the comparisons and how to interpret them in both the results and the Discussion section. This feels like it would be the primary analysis of the paper and it was under-developed in both the results and discussion.

The authors could consider strengthening their results by including some additional analyses exploring individual differences. Children with ASD on the whole become less convergent with each other over time, but are there children who become more convergent with the TD group and those who do not?

A decent amount of space in the paper is dedicated to analyses that are only included in the supplemental analyses. The authors may consider restructuring some of the main and supplemental texts such that the relevant figures will appear near the analyses.

The authors suggest a direction of causality wherein TD children are relying more on lower-level salient features of the scene than are children with ASD. However, salient features of a scene as predicted by the Itti & Koch model are often highly correlated with the social aspects of a scene. Especially in a cartoon, backgrounds remain consistent and the only movement is vibrantly-colored characters moving across the scene. The authors should edit the methods to exclude the suggestion of causality and the above point should be discussed.

The authors should discuss why gaze behavior correlated with adaptive behavior scales but not with overall autism symptom severity.

In general, additional context could be provided to the Results section to clarify what questions the authors were trying to answer with each analysis.

[Editors’ note: further revisions were suggested prior to acceptance, as described below.]

Thank you for resubmitting your work entitled "Unraveling the Developmental Dynamic of Visual Exploration of Social Interactions in Autism" for further consideration by *eLife*. Your revised article has been evaluated by Christian Büchel (Senior Editor) and a Reviewing Editor.

The manuscript has been improved but there are some remaining issues that need to be addressed, as outlined below:

1) Clarify the "coarse movie characteristic". The description provided does not allow us to replicate this metric.

2) Details for the permutation testing on the longitudinal changes in gaze divergence using a sliding window method are missing. What exactly was done to compute the significance value?

3) Clarify the role of the "reference distribution". This distribution seems not to have been used in any of the analyses.

4) More comparisons of the TD and ASD groups at the individual subject level should be performed to provide more insightful information. This could also address the major concern of reviewer #3 that the groups are quite different with respect to homogeneity and size.

5) A control analysis taking into account mental age would be helpful as chronological and mental age seem to differ more in ASD and this analysis could rule out potential confounds that are indicative of general developmental delays rather than ASD-specific characteristics.

*Reviewer #2 (Recommendations for the authors):*

Points for clarification:

– I am a bit unclear on what the "coarse movie characteristic" is, exactly. The description provided would not allow me to replicate this metric – can a more quantitative description be provided?

– In response to Reviewer 3, RC4, the authors now provided permutation testing on the longitudinal changes in gaze divergence using a sliding window method. However, I am unclear on the details of the permutation testing. To quote from the paper: "To test for the statistical significance of the difference between the two groups, we employed random permutation testing across 59 age windows. Accordingly, in each of the 59 windows, gaze data from the TD (20) and ASD groups (20 recordings per window) were pooled together. We performed 100 randomly permuted resamples of equal size to the original distribution (20) from this pooled sample to compute the significance value. The windows where the MCS values showed statistically significant differences between the two groups are graphically presented with color-filled circles (Figure 8C)." What exactly was done to compute the significance value? As far as I understand, they calculated the group divergence from 20 resampled data to generate a 'null divergence' for each group, then calculate that difference 100 times as a null distribution of the group difference, and finally compare the actual group difference with this null distribution. However, from their response letter, the distribution seems to be the group-level dispersion, but not the group difference. In their rebuttal letter, the authors write, "We applied 100 permutations inside each of the 59 windows (containing 20TD + 20ASD gaze recordings) to derive a null distribution of the measure of dispersion -the average pairwise distance between gaze coordinates inside the group." This seems inconsistent.

– In the Method of the "Maturational changes in visual exploration of complex social scene – Sliding window approach", in the second paragraph, the authors mention a reference distribution. However, this reference distribution seems not to have been used in any of the analyses (it's not mentioned in the longitudinal results at all). To quote from the paper: "Upon the creation of sliding windows and to characterize the group's visual behavior and its change with age, gaze data from the TD group were pooled together to define the referent distribution in each of the 59 age windows. To characterize the group visual behavior in the ASD group, we performed the same by pooling the gaze data together from ASD in each of the 59 age windows (see Figure 8 A&B)." Indeed, Figure 8A is about how to decide the sliding window, and Figure 8B is about pairwise gaze dispersion, not about referent distribution.

*Reviewer #3 (Recommendations for the authors):*

The present research uses the Typical Development (TD) group as a normative reference, comparing individual participants with Autism Spectrum Disorder (ASD) against this reference. However, as pointed out by Reviewer #2, this approach doesn't allow for a comprehensive understanding of the variation within the TD group itself. The leave-one-out calculation suggests that the π is higher for TD, but the TD group is also more homogenous, so I am not sure this is truly informative. A comparison between TD and ASD groups at the individual subject level could provide more insightful information.

Additional Concerns:

i) The study matches the two groups based on chronological age rather than mental age. This introduces a potential confound as the differences reported may be indicative of general developmental delays rather than ASD-specific characteristics.

ii) The TD group is not only smaller in size but also less heterogeneous, which may be a potential explanation for the findings illustrated in Figure 8C.

---

## [Author Response]

[Editors’ note: the authors resubmitted a revised version of the paper for consideration. What follows is the authors’ response to the first round of review.]

Comments to the Authors:Reviewer #1 (Recommendations for the Authors)In the introduction, it would be helpful if the authors mention the age range and functioning level of participants for every prior study they cite that used eye-tracking to assess social attention in autism. In addition, statements such as "these atypicalities are present early" should include what is meant by early (what age range).

We thank the reviewer for attracting our attention to this point. Following the reviewer’s comment, we carefully revised the manuscript to provide age information when possible.

“[…] These atypicalities are observed as early as 2 months of age (Jones and Klin, 2013) and thus can exert tremendous impact on downstream developmental processes that critically depend on experience. […] Indeed, it has been shown that in the context of naturalistic static scenes, both children and adults with ASD tend to focus more on basic, pixel-level properties than on semantic categories, compared to their TD peers (Amso et al., 2014; Wang et al., 2015). […] For example, it has been shown that in the context of dynamic social content preschoolers with ASD tend to focus less on motion properties of the scene and more on intensity in comparison to age matched TD children (Shic et al., 2007). […] Quite strikingly, while viewing social scenes, toddler and school-age twins showed a high concordance not solely in the direction but also in the timing of their gaze movements (Constantino et al., 2017; Kennedy et al., 2017). […] Only few studies tackled the question of the moment-to-moment gaze deployment in ASD compared to TD. Indeed, while on this microstructural level TD children and adults show coherence in fixation targets, fine-grained gaze dynamic in their peers with ASD is highly idiosyncratic and heterogeneous (Avni et al., 2019; Falck-Ytter and Hofsten, 2011; Nakano et al., 2010; Wang et al., 2018).”

35 age-matched TD children were used to obtain normative gaze patterns, but it is not clear how this number was determined. Are there either power analyses or previous data suggesting that this sample size is sufficient to determine what is normative?

Indeed, a better definition of the sample size used as normative is warranted, considering its central role in our manuscript. To our knowledge, there is no consensus on the optimal size of a sample considered normative in the studies using analyses similar to the one we developed in this manuscript. Previous studies measuring the temporospatial "typicality of gaze " in ASD regarding the TD reference group also used convenience samples. The normative sample characteristics, such as size, age range, and gender distribution, varied notably. A study using multidimensional scaling had a normative group of 25 TD children (mean age: 3.1±1.11 years, 44% females)(Nakano et al., 2010); another study comparing the gaze of TD and ASD children to the mean gaze pattern of both groups included 40 TD children (mean age: 4.5±2.1, 37.5 % females) (Avni et al., 2019) and finally, a study on gaze cohesion included 163 TD children (mean age: 21.89±3.39 months, 40.5%females) (Wang et al., 2018). Similarly, our earlier work focusing on eye-tracking derived Proximity Index (PI) coupled to EEG data from preschoolers with ASD used normative gaze data of variable size. In the first paper, we defined the normative gaze distribution using the gaze recordings from 18 TD children (mean age 3.1±0.9 years, 30% females) (Sperdin et al., 2018) while for the second, we were able to increase the sample size to 26 TD males (mean age 3.4±1.2 years) (Jan et al., 2019). While preparing these studies, we noticed that the normative gaze distribution would become stable after the sample size of 15. Thus, our initial manuscript version’s sample of 35 TD children seemed adequate. Thanks to continued data collection during the revision process, we further were able to increase the sample of TD children slightly, reaching a final normative sample size of 51 in the revised analysis presented in the newly submitted manuscript.

Still, we never formally addressed the stability of the normative gaze distribution in any of our previous works, nor did we conduct power analysis before preparing the sample. To answer the reviewer’s comment, we addressed the question of interindividual gaze stability. We conducted analyses to understand how much the distribution of the TD gaze would change if done on a smaller sample. Inspired by others (Schaer et al., 2015), we performed bootstrapping to simulate smaller samples from the total available sample of 51 TD. We simulated sample sizes ranging from 10 to 50 TD children in these analyses. For each sample size level, we obtained 500 bootstrapped samples over which we measured the stability of the distribution. These analyses demonstrated that the stability of the distribution in the TD sample is, on average, reached at a sample size of 18, Appendix 1—figure 1. In the revised version of the paper, we added these results as a subsection in Appendix 1, as follows:

“The sample of 51 TD children whose gaze data was used to obtain a normative gaze distribution was a convenience sample. In the present study, we only included males due to the fewer number of females with ASD. Having this unique sample of TD children, we tested the stability of the normative distribution depending on the sample size by performing bootstrap analyses. Thus, from the available sample of 51 TD children, we performed 500 bootstraps, starting with a sample size of 10 until reaching the sample size of 50. To measure the change in gaze distribution on one frame, we calculated the average pairwise distance between all gaze coordinates available on the frame. Then for each frame, we calculated the variance of the average pairwise distance over 500 resamples. Finally, the variance obtained was averaged over the 5150 frames to yield a unique value of the variance in gaze patterns per sample size (10-50). Then we calculated the "cutoff," as defined by a sample size increase no longer yielding significant variation in the average variance. This was done using the *kneed* package implemented in Python that estimates the point of maximal curvature ("elbow in curves with positive concavity) in discrete data sets based on the mathematical definition of curvature for continuous functions (Satopaa et al., 2011) (see Figure 2). The elbow of the fitted curve on our bootstrapping data was found at 18, meaning that the distribution was estimated to be stable from a sample size of 18.”

A related issue is that of age norms. How were age-specific norms considered, given that the sample include children between 2-7 years of age and was not very large (eg. it is not clear how many children of each age were included in the sample)?

We agree with the reviewer that the age span of the normative sample in the previous version of the manuscript is relatively large. As explained in our previous answer, the normative sample increased from 35 to 51 during the revision process. We were also able to increase the sample size of the ASD sample; the exact age distribution of the revised cross-sectional sample (51 TD children and 166 children with ASD) is depicted on Author response image 1. At this initial level of analysis, we aimed for a broad norm covering a more extensive age range in childhood to allow a comparison of a bigger sample of children with ASD (having the age in the norm range) to this unique reference point. This first step of the analysis was important for delineating group differences in visual exploration during childhood. Following this first level of analysis, we adopted a more fine-grained approach to defining the trajectories of change in gaze patterns in both groups. Thus using the available longitudinal data from the children included in the initial level of analyses and using the sliding window approach we were able to define smaller age-matched groups covering more restricted age periods (Figure 8).

**Author response image 1. sa2fig1:** Age distribution in our cross-sectional sample including 51 TD children and 166 children with ASD.

In this second level of analyses, where we tackled the developmental change of visual exploration, more closely matched age groups allowed us to get a better grasp of age-specific processes. As detailed in the Method section’s subsection, "Unstructured longitudinal sample," and following Sandini et al. (2018), we used a sliding-window approach to delineate developmental changes. Participants repeated visits were included when satisfying the inclusion criteria. This yielded a total of 308 recordings for the ASD group and 105 for the TD group (all recordings were collected a year apart; 101 children with ASD contributed two recordings each, and 41 children with ASD contributed three recordings each, while 33 and 21 TD children contributed respectively 2 and 3 recordings each). Available recordings from our unstructured longitudinal sample were first ordered according to the age in both groups separately. Then, a window encompassing 20 recordings was progressively moved across two groups, starting from the first 20 recordings in the youngest subjects until reaching the end of the recording span for both groups. The choice of window width was constrained by the sample size of our TD group. The longitudinal visits in our cohort are spaced a year from each other. The choice of a bigger window would result in important data loss in our group of TD children, as the windows were skipped if they contained more than one recording from the same subject. The chosen window width of 20 yielded 59 sliding windows in both groups that were age-matched and spanned the period from 1.9 – 4.2 years old on average (see Figure 6, Figure 8). As discussed in response to comment nº2, according to our stability analysis, the window width of 20 seemed a good compromise given that stability is reached at the sample size of 18. Moreover, it allows inferences on the gaze behavior between the two groups with enough temporal resolution between the age of 1.9 years to 4.2 years of age.

**Author response table 1. sa2table1:** Comparison of mean age across 59 age-matched sliding windows for the TD and the ASD group. pvalues are obtained using t-test of differences between the ASD and TD group across 59 windows.

	TD	ASD				TD	ASD				TD	ASD					
	**Mean**	**SD**	**Mean**	**SD *p* value**	**Mean**		**Mean**	**SD**	**Mean**	**SD *p* value**	**Mean**		**Mean**	**SD**	**Mean**	**SD *p* value**	**Mean**
**SW 1**	1.88	0.242	1.94	0.118	0.813	**SW 21**	2.78	0.226	2.86	0.065	0.975	**SW 41**	3.53	0.239	3.61	0.082	0.931
**SW 2**	1.93	0.224	1.97	0.111	0.931	**SW 22**	2.82	0.215	2.90	0.060	0.911	**SW 42**	3.57	0.236	3.64	0.088	0.919
**SW 3**	1.96	0.238	2.01	0.108	0.872	**SW 23**	2.86	0.202	2.93	0.055	0.973	**SW 43**	3.61	0.233	3.68	0.092	0.998
**SW 4**	2.00	0.251	2.05	0.111	0.880	**SW 24**	2.90	0.193	2.96	0.059	1.000	**SW 44**	3.65	0.230	3.73	0.091	0.951
**SW 5**	2.04	0.268	2.09	0.113	0.986	**SW 25**	2.93	0.187	3.00	0.062	0.916	**SW 45**	3.68	0.220	3.75	0.085	0.952
**SW 6**	2.09	0.286	2.14	0.114	0.912	**SW 26**	2.96	0.189	3.04	0.062	0.980	**SW 46**	3.72	0.212	3.80	0.072	0.981
**SW 7**	2.14	0.300	2.18	0.112	0.936	**SW 27**	3.00	0.197	3.07	0.060	0.935	**SW 47**	3.76	0.211	3.83	0.061	0.996
**SW 8**	2.18	0.307	2.22	0.100	0.933	**SW 28**	3.04	0.201	3.11	0.057	0.958	**SW 48**	3.79	0.209	3.86	0.054	0.950
**SW 9**	2.23	0.309	2.28	0.089	0.951	**SW 29**	3.07	0.201	3.14	0.057	0.926	**SW 49**	3.83	0.203	3.90	0.054	0.972
**SW 10**	2.28	0.308	2.33	0.066	0.965	**SW 30**	3.11	0.204	3.18	0.063	0.952	**SW 50**	3.87	0.199	3.94	0.062	0.920
**SW 11**	2.33	0.308	2.43	0.080	0.948	**SW 31**	3.14	0.209	3.21	0.062	0.957	**SW 51**	3.90	0.197	3.97	0.061	0.961
**SW 12**	2.38	0.301	2.48	0.073	0.928	**SW 32**	3.18	0.210	3.25	0.059	0.971	**SW 52**	3.94	0.190	4.01	0.055	0.977
**SW 13**	2.43	0.296	2.52	0.070	0.947	**SW 33**	3.21	0.213	3.29	0.054	0.982	**SW 53**	3.97	0.184	4.04	0.049	0.939
**SW 14**	2.48	0.285	2.57	0.065	0.992	**SW 34**	3.25	0.217	3.33	0.054	0.962	**SW 54**	4.01	0.179	4.07	0.047	0.998
**SW 15**	2.53	0.278	2.62	0.067	0.963	**SW 35**	3.29	0.230	3.37	0.068	0.959	**SW 55**	4.04	0.188	4.10	0.054	0.933
**SW 16**	2.57	0.271	2.66	0.071	0.936	**SW 36**	3.33	0.240	3.42	0.070	0.977	**SW 56**	4.07	0.199	4.14	0.068	0.998
**SW 17**	2.61	0.260	2.70	0.072	0.998	**SW 37**	3.38	0.243	3.45	0.067	0.929	**SW 57**	4.11	0.205	4.18	0.069	0.976
**SW 18**	2.66	0.242	2.74	0.077	0.985	**SW 38**	3.42	0.242	3.49	0.062	0.966	**SW 58**	4.14	0.213	4.23	0.063	0,941
**SW 19**	2.70	0.238	2.78	0.076	0.950	**SW 39**	3.45	0.244	3.53	0.057	0.992	**SW 59**	4.19	0.229	4.28	0.050	0.978
**SW 20**	2.74	0.233	2.83	0.069	0.937	**SW 40**	3.49	0.242	3.56	0.066						

In addition to child adaptive behavior, were any other assessments conducted and if so, were they related to visual exploration patterns in children? Were any of the visual exploration metrics related to severity of autism symptoms?

Besides adaptive behavior, in all children included in our sample, we measured autistic symptoms and developmental functioning levels. As stated in the initial version of the manuscript, we assessed the symptoms of autism using the Autism Diagnostic Observation Scale (ADOS), the developmental profile was assessed using the Psychoeducational Profile – 3rd version (PEP-3), and adaptive levels were measured using the Vineland Adaptive Behavior Scale – 2nd version (VABS-II). To understand how the Proximity index (PI) was related to these three broad phenotype domains, we conducted several correlation analyses in the previous version of the manuscript. Visual exploration in children with ASD (expressed in PI) was positively related to developmental and adaptive functioning and negatively to the measure of symptom severity as measured by ADOS (although this latter association was not statistically significant). In the initial manuscript version, we included the details of the association of the π and uniquely global scores across three assessed domains to limit the number of comparisons in the main manuscript. The relation between π and more specific skill domains (i.e., sub-scales of the used tests, such as the Communication or Socialization domain of the VABS-II) was presented only in the section of the Supplementary material in the previous version of the manuscript.

In the current version of the paper, we changed analysis strategy to obtain a more holistic appreciation of the relationship between the π and the phenotype measures. Here we conducted a multivariate analysis allowing, in our opinion, a better grasp of the relation between the visual exploration of the given social scene and the more nuanced clinical behavioral characteristics of the children with ASD. The advantage of the current analysis is that we can appreciate the relation between the π and several behavioral characteristics simultaneously in relation to one another. This analysis confirmed that the children with better developmental and adaptive functioning levels across all assessed domains also had higher values of the PI. We found no significant relationship between the π and symptom severity as measured by the ADOS at the initial time point (cross-sectional sample of 166 children) (see Figure 3). Additionally, in our longitudinal sample, we found that the PI, while still not related to the simultaneous measure of symptoms (ADOS scores at baseline – T1) was related to the symptoms a year later (T2). This finding is discussed in the current version of the paper. The description of the relations between visual exploration (as measured by PI) and the phenotype characteristics in the current version of the manuscript is as follows:

*“*Less divergence in visual exploration is associated with better overall functioning in children with ASD

To explore how the gaze patterns, specifically divergence in the way children with ASD attended to the social content, related to the child’s functioning, we conducted a multivariate analysis. We opted for this approach to obtain a holistic vision of the relationship between visual exploration, as measured by PI, and different features of the complex behavioral phenotype in ASD. Behavioral phenotype included the measure of autistic symptoms and the developmental and functional status of the children with ASD. Individuals with ASD often present lower levels of adaptive functioning (Franchini et al., 2018; Hus Bal et al., 2015) and this despite cognitive potential (Klin et al., 2007). Understanding factors that contribute to better adaptive functioning in very young children is of utmost importance (Franchini et al., 2018) given the important predictive value of adaptive functioning on later quality of life. The association between behavioral phenotype and π was examined using the PLS-C analysis (Krishnan et al., 2011; McIntosh and Lobaugh, 2004). This method extracts commonalities between two data sets by deriving latent variables representing the optimal linear combinations of the variables of the compared data sets. We built the cross-correlation matrix using the π on the left (A) and 12 behavioral phenotype variables on the right (B) side (see Methods section for more details on the analysis).

[…]

The PLS-C conducted on simultaneous π and phenotype measures at the first time point (T1-PI – T1 symptoms) essentially replicated the pattern we observed on a bigger cross-sectional sample. One significant LC (*r*=0.306 and *p*=0.011) showed higher π co-occurring with higher cognitive and adaptive measures (see Figure 12). The cross-covariance matrix using a π at T1 to relate to the phenotype at the T2 also yielded one significant latent component (*r*=0.287 and *p*=0.033). Interestingly, the pattern reflected by this LC showed higher loading on the π co-occurring with lower loading on autistic symptoms. Children who presented lower π values at T1 were the ones with higher symptom severity at T2. The gaze pattern at T1 was not related to cognition nor adaptation at T2 (see Figure 13, panel A). Finally, the simultaneous PLS-C done at T2 yielded one significant LC where higher loading of the π coexisted with negative loading on autistic symptoms and higher positive loading on the adaptation score (*r*=0.322 and *p*=0.014) Figure 13, panel B. The level of typicality of gaze related to the symptoms of autism at T2 (mean age of 4.05±0.929) but not at a younger age (mean age of 3.01±0.885). This finding warrants further investigation. Indeed, on the one hand, the way children with TD comprehend the world changes tremendously during the preschool years, and this directly influences how the typicality of gaze is estimated. Also, on the other hand, the symptoms of autism naturally change over the preschool years, and all these elements can be responsible for the effect we observe.”

There is not much information provided regarding the TD sample. What screening tools and procedures were used to detect neurological or psychiatric diseases?

We agree with the reviewer that the description of the TD group was insufficient in the initial manuscript version. Before inclusion in our cohort, we conduct screening interviews with the child’s parent(s). The children are not eligible for our TD group if they were treated for any psychological or neurological problem, if the parents have concerns about their development, or if they have a first-degree relative with a known ASD. Following this initial screening, all eligible TD children in our sample underwent the same assessment as the ASD children. The presence of autistic symptoms is excluded using direct observation (ADOS) and parent-reported measure of symptoms (Autism Diagnosis Interview Revised, ADI-R). In addition, we obtain detailed developmental and cognitive profiles and the profile of adaptive functioning in all TD children. We collect information on children’s medical history, including pregnancy history. Children are excluded from the TD group if they have a developmental delay (-1SD) in any assessed developmental areas of functioning.

Following the reviewer’s comment, we have modified the Method section of the manuscript. We now include a table containing the detailed clinical characteristics of the two groups with regards to the autistic symptoms (ADOS-2 & ADI-R), developmental (PEP-3), and adaptive functioning (VABS-II), Table 1. The table was added to the methods section of the revised manuscript version, and the manuscript was modified as

*“*Experimental Model and Subject Details

Hundred sixty-six males with autism (3.37 ± 1.16 years) and 51 age-matched typically developing males (3.48 ± 1.29 years) participated in the study. Table 1 summarizes the clinical characteristics of our crosssectional sample. Our study included only males due to fewer females with ASD. The clinical diagnosis of autism was confirmed using the standardized observational assessment of the child and interviews with caregivers(s) retracing the child’s medical and developmental history. All children with ASD reached the cut-off for ASD on Autism Diagnostic Observation Schedule-Generic (ADOS-G), (Lord et al., 2000) or Autism Diagnostic Observation Schedule-2^nd^ edition (ADOS-2) (Lord et al., 2012). For children who underwent the ADOS-G assessment, the scores were recoded according to the revised ADOS algorithm (Gotham et al., 2007) to ensure comparability with ADOS-2.

Before inclusion in the study, typically developing (TD) children were screened using a questionnaire focusing on medical history and history of pregnancy. Children were not included in our TD group if they were born prematurely or had a positive screen for the presence of any known neurological or psychiatric disorder in the child itself or the known case of ASD in any first-degree relative of the child. Moreover, all TD children were also assessed using the ADOS-G or ADOS-2 evaluations to exclude the presence of ASD symptoms. The majority of TD participants had a minimal severity score of 1, except four children who had a score of 2.

The data for the current study were acquired as a part of a larger longitudinal study of early development in autism based in Geneva. Detailed information about cohort recruitment has been given elsewhere (Franchini et al., 2018; Franchini et al., 2017; Kojovic, 2019). This study protocol was approved by the Ethics Committee of the Faculty of Medicine of Geneva University, Switzerland. All families gave written informed consent to participate.”

Reviewer #2 (Recommendations for the Authors)The main novelty of the study lies in a longitudinal component. Some of the subjects were tested again, about a year apart. Using a sliding window approach, these data were used to derive continuous measures of gaze atypicality over age (this analysis was complemented by a more rigid analyses based only on those subjects with two datapoints one year apart). Median contour surface and convergence index were the two metrics used here. Both showed developmental differences between groups. Visual features in the video were examined using a saliency model, and social complexity was quantified as the number of characters on the screen while controlling for overall visual complexity. Overall, it was found that while spatial divergence in gaze decreased in the TD group (convergence index increased) over the developmental time window examined, this was not the case for the ASD group (Figure 3).These are labor-intensive and valuable data that will certainly add to the literature on attention and visual processing in ASD. The longitudinal dataset is quite special, and the results are in broad strokes fairly compelling: there are different developmental trajectories in visual attention in ASD compared to TD controls. The methods are reasonably sophisticated (but see my comments below). My main hesitation with this study is that it stops short of providing a stronger mechanistic advance. In particular, the developmental eye tracking data are really the centerpiece of the work. However, they are not linked to any other developmental measures. Actually, it was unclear to me at which timepoints the other behavioral measures (Vineland, etc.) were measured. It would have been ideal to get the Vineland and other measures of functioning at each of the specific timepoints, one year apart, at which the eye tracking data were collected to examine associations. But that seems not do have been done (or at least is not reported).

We thank the reviewer for his time invested in reviewing our work and for the very insightful and inspiring comments. We are glad that the reviewer recognizes the contribution of our study to understanding the early development of visual exploration in autism.

We are thankful that the reviewer pointed out to the lack of clarity in the previous version of the manuscript regarding the behavioral measures available at different measurement points. In all children included in our sample, we measured autistic symptoms and developmental and adaptive functioning levels. Thus each time we obtained the eye-tracking measures, we also obtained the measures in these three phenotype domains. As stated in the initial version of the manuscript, we assessed the symptoms of autism using the ADOS (Lord et al., 2000; Lord et al., 2012), the developmental profile was assessed using the PEP-3 (Schopler, 2005), and adaptive levels were measured using the VABS-II (Sparrow, Balla, and Cicchetti, 2005).

In the present manuscript version, we significantly increased our sample size, the cross-sectional sample now includes 166 children with ASD compared to 59 in our initial manuscript version, and the longitudinal sample consists of 81 children with ASD (previously 34). Another important change concerns the analysis strategy. In the present manuscript version, we used the multivariate approach to test the relation between π and behavioral phenotype (autistic symptoms, developmental and adaptive levels). As the sample size increased, we could deploy the multivariate analyses not solely on the cross-sectional sample but also on the longitudinal sample. By exploring the link between the π and clinical phenotype measures in the longitudinal sample, we added an element that was critically lacking in the previous version of the manuscript. Here we tested both the simultaneous relation (at baseline: T1 and one-year-follow-up: T2) and also how the π at baseline was related to the behavioral phenotype at one-year-follow-up.

A second shortcoming of the study is the lack of detail with respect to individual subjects (were there any clusters/subgroups/outliers of interest?) and with respect to stimulus features that could be driving the observed effects. There is the saliency analysis, and an analysis of how many characters are on the screen, but that is it. Surely one could quantify additional semantic features, even if ratings are obtained from adults. As it stands there are unconnected analyses in the study. For instance, I have the bottom of Figure 1, showing that proximity index in a subject with ASD varies tremendously over the frames of the movie. Does it vary like that in other subjects with ASD? Does this variation line up with specific features in the video? Most importantly, I would want to see the framewise plot like this (or better: slightly temporally smoothed) of proximity index differences between TD and ASD, as a function of the longitudinal data. What features in the video correlate with the increased convergence seen in TD that are missing in ASD? The analyses provided do not back out these specifics, and without them the story is more descriptive than mechanistic.

We thank the reviewer for the detailed suggestions. We did not identify outliers driving the observed effects (no π value was more than three scaled median absolute deviations (MAD) away from the group median). With the current sample of 166 children with ASD, we confirm the finding from the initial version that focused on 59 children where more divergence in gaze patterns in the ASD group at baseline was related to poorer cognitive and adaptive functioning. The fact that the findings are confirmed on an almost three times bigger sample speaks against the fact that the outliers drive the initial effects.

Considering the movie content, in the initial manuscript version, we focused only on the social and visual complexity and saliency analyses that were considered separately. Following the reviewer’s suggestions, we now extracted additional movie features in the current version of the work. Thus besides the social and visual complexity, we now have included the vocal characteristics of the movie, namely whether vocalizations/verbalizations were socially addressed (directed speech) or not (monologue). Finally, we also marked the coarse characteristics of the movie sequence that might influence the gaze deployment, namely the rapid change in the movie (frame switch) but also slower change where the whole background of the scene was moving to follow the motion of the characters. To appreciate the relation of all these movie features with the PI, especially considering the lack of orthogonality among them, we conducted a multivariate analysis. In addition to six regressors obtained from the annotation of the movie content, we added the salience information to obtain a more global view of what elements contributed more to the gaze deployment in the ASD group. As described in the example we show below, the π was most influenced by social complexity, followed by visual complexity, vocal aspects of the movie, and finally, coarse characteristics of movie sequence (rapid switch and slide) and salience.

Methods:

“Movie characteristics

Social complexity

Furthermore, given the findings of the important impact of intensity of social content on social attention in ASD (Chita-Tegmark, 2016; Frank, Vul, and Saxe, 2012), we aimed to test the hypothesis that the Proximity Index values will be lower for the moments in the videos with enhanced social complexity, involving two or three characters compared to moments involving only one character Figure 8A. However, with an increasing number of characters, the scene is inevitably richer in detail, an issue we address through measuring visual and vocalization complexity.

Visual complexity

To measure visual complexity, we calculated the length of edges delimiting image elements (see Figure 8B). Edge extraction was done on every image of the video using the Canny method (Canny, 1986) implemented in Matlab (version 2017a; Mathworks, Natick, MA). This method finds edges by looking for local maxima of the intensity gradient. The gradient is obtained using the derivative of a Gaussian filter and uses two thresholds to detect strong and weak edges. Weak edges are retained only if connected to strong edges, which makes this method relatively immune to noise.

Vocal video aspects: Monologue and Directed speech

Speech properties of the scenes were also analyzed, using the BORIS software (https://www.boris.unito. it/). We manually identified the moments when characters were vocalizing or speaking. Then we annotated the moments as a function of the social directness of the speech. In particular, we distinguished between monologue (characters thinking out loud or singing) and moments of socially directed speech (invitation to play and responses to invitations).

Coarse movie characteristics: Frame switching and moving background

Finally, to test how global characteristics of the video media, scene changes, or type of the scenes would influence gaze deployment, we extracted moments of the frame switch and moments where the background would move (slide) to follow the movement of the characters along movie scenery.”

Results:

“The association of movie content with divergence in visual exploration in ASD group

Taking into account previous findings of enhanced difficulties in processing more complex social information (Chita-Tegmark, 2016; Frank, Vul, and Saxe, 2012; Parish-Morris et al., 2019) in individuals with ASD, we tested how the intensity of social content influenced visual exploration of the given social scene. As detailed in the Methods section, social complexity was defined as the total number of characters for a given frame and ranged from 1 to 3. Frames with no characters represented a substantial minority (0.02% of total video duration) and were excluded from the analysis. We also analyzed the influence of the overall visual complexity of the scene on this divergent visual exploration in the ASD group. The total length of edges defining details on the images was employed as a proxy for visual complexity (see Methods section for more details). Additionally, we identified the moments of vocalization (monologues versus directed speech) and more global characteristics of the scene (frame cuts and sliding background) to understand better how these elements might have influenced gaze allocation. Finally, as an additional measure, we considered how well the gaze of ASD children was predicted by the GBVS salience model or the average ROC scores we derived in the previous section Figure 9, panel A.

To explore the relationship between the π and different measures of the movie content as previously, we used a PLS-C analysis that is more suitable than the GLM in case of strong collinearity of the regressors this is particularly the case of the visual and social complexity *r* = 0.763, *p <* 0.001, as well as social complexity and vocalization (*r* = 0.223, *p <* 0.001), as can be appreciated on the Figure 9, panel B. The PLS-C produced one significant latent component (*r* = 0.331, *p <* 0.001). The latent component pattern was such that lower π was related to higher social complexity, followed by higher visual complexity and the presence of directed speech. In addition, moments including characters engaged in monologue, moments of frame change, and background sliding increased the π in the group of ASD children. The monologue scenes also coincide with the moments of lowest social complexity that produce higher π values. For the frame switch and the sliding background, the TD reference appears more dispersed in these moments as children may recalibrate their attention onto the new or changing scene, making the referent gaze distribution more variable in these moments and thus giving ASD more chance to fall into the reference space as it is larger. Finally, visual salience also positively contributed to the π loading, which is in line with our previous finding of the salience model being more successful in predicting TD gaze than ASD gaze.”

1. Since the TD group was always used as a reference, and single ASD participants were compared to this "norm", we do not have a good estimate of how TD subjects would look by comparison. A stronger approach would calculate the TD norm in a leave-one-out fashion and generate distributions for how each individual TD subject also compared – those are the data then to compare (single TD and single ASD) rather than just ASD individuals to one fixed TD average. The authors refer to their analyses as "data-driven", and it is perhaps that with respect to not using ROIs on the video, but they are testing specific differences between pre-defined groups (ASD and TD). It would substantially strengthen the paper if indeed a data-driven analyses were provided (clustering the subject groups on the basis of the data, rather than as predefined). This could also help reveal possible outliers (in both groups) as well as possible subgroups.

We are extremely grateful to the reviewer for these precious comments. We agree that the absence of an additional control group was a true limiting factor of our manuscript. Following the suggestion of the reviewers and using the leave-one-out method, we obtained the values of the π for our TD group (Figure 4). The following paragraph has been added to the Results section of the manuscript:

“As the gaze data of the TD group were used as a reference, we wanted to understand how their individual gazing patterns would behave compared to a fixed average. Due to the absence of an additional control group, we employed the leave-one-out method to obtain the values of the π for the 51 TD children. In this manner, the gazing pattern of each TD child was compared to the norm comprising the gaze data of 50 other TD children. The difference in average π values between the two groups was found significant, *t*_(215)_ = 5.51, *p <* 0.001, with a considerable heterogeneity, especially in the ASD group (Figure 2).

Regarding the reviewer’s comment on clustering the subjects using a data-driven approach, we completly agree with the reviewer that such a fully data-driven method would be more interesting. However, data driven clustering is beyond the scope of the current study and would substantially expand the breadth and number of levels of an already long study. At this stage, our aim was to present a novel method for measuring deviation in gaze patterns that can be used to measure the developmental dynamics of gaze deployment in children with ASD and establish its relation with the clinical phenotype, both at baseline and in one-year follow-up. We felt that this already represented a comprehensive endeavor. That being said, we fully agree with the reviewer about the potential of a data-driven approach. Our next goal will be to explore the potential of the new measure to enable the classification of the two groups without an a priori definition of the class. For this, we would need to consider the dynamic properties of the π for the duration of the entire video to predict the class (ASD or TD) of our participants as the aggregated (averaged π values for the video) show considerable overlap between the two groups, as shown in Figure 2. An unsupervised class attribution using the eye-tracking measure may be a potentially promising avenue. We hope, however, that the reviewer agrees with us that this would be better suited for a separate paper as its goals would, in our opinion, be a better fit in a paper discussing this new eye-tracking measure as a potentially useful screening tool for the detection of the ASD, which is out of the scope of the current paper.

2. It is intriguing that the fixation durations between groups differed. I wonder if the authors would consider inspecting smooth pursuit, a type of oculomotor feature not mentioned but that could be relevant (and has been reported to be atypical in ASD).

Indeed, this is a precious suggestion. A more fine-grained gaze behavior analysis is always appealing. Up to this point, we explored mostly coarse characteristics of the visual exploration style for the current manuscript, using the fixations and saccades properties. However, considering that the manuscript is already rather rich encompassing two separate types of the analysis (cross-sectional and longitudinal), we were afraid to broaden the scope by adding a new set of analyses. We are thankful to the reviewer for this valuable suggestion and will explore the potential of this visual behavior in our future work.

3. Subjects: More information on the subjects is needed. On what criteria were ASD and TD matched, was level of functioning or intellect taken into consideration? Did ASD subjects have a DSM diagnosis? Was the ADI done? What exclusionary criteria were applied (epilepsy, comorbidity, medication, etc?). More info please.

We apologize for the lack of precise characterization of the two groups in our initial sample. Our two groups were age-matched in both previous and current versions of the manuscript. In both versions our sample only included males, as the number of females in the ASD group was smaller compared to males. Having a sex-mixed sample would mean that we would need to substantially lower the sample size, which would be not optimal considering our analysis strategy. In addition, we were very cautious to include females as previous research (Harrop et al., 2018, 2019) has highlighted sex-related differences in visual exploration between males and females, and this difference would warrant more detailed characterization using our method. We, of course, recognize the importance of the more in-depth characterization of sex differences in visual exploration in young children using our method.

We did not match our two groups based on intellectual or functional levels. In our opinion, matching the children on these two criteria would drive us away from our intention to understand better the visual exploration in a representative group of children with ASD and not uniquely the ones on the higher functioning end of the autistic spectrum. All children with ASD had the DSM-5-informed ASD diagnosis. We obtained a detailed medical and developmental history of all children in our sample. In addition, children were assessed on three broad domains, namely autistic symptoms (using ADOS-G/ADOS-2 and ADI-R), developmental (PEP-3), and adaptive functioning (VABS-II). For the exclusion criteria, potential participants would be excluded if they presented a known genetic condition with autism-like traits (such as Fragile X, and Tuberous Sclerosis). To our knowledge, we had no children with such genetic conditions in the sample that satisfied specific eye-tracking inclusion criteria. One child has a known diagnosis of epilepsy but no medication at the time of our study. In the revised manuscript, we added a table that compares the two groups with regard to their age, the severity of autism symptoms, and developmental and adaptive functioning across domains (see Table 1). Also, we modified the participant description paragraph in the Method section as follows:

4. Analysis: proximity index. Why was the proximity index not temporally smoothed? A frame-by-frame metric will (a) have relatively sparse normative distributions, and (b) show fairly discontinuous gaze proximity (as evident in the plot shown in Figure 1, bottom).

We are aware that the frame-by-frame metric such as the Proximity Index is prone to scarceness and we understand the reviewer’s concern that temporal smoothing could provide useful for analyses of the time course. Nevertheless, we did not perform smoothing or any other intervention on the frame-by-frame derived π values, because we only use average π values (average over all frames) in all analyses presented in the manuscript. As such, using the raw signal or the smoothed value would not change anything in any of the presented results.

Following the reviewer’s highly pertinent comments on the relation between the content of the video and the Proximity index, we performed smoothing for visual illustration in the section dealing with the movie content and its effects on the gaze deployment, Figure 6, first panel(in red).

5. All of the analyses are within-sample and use standard parametric statistics; it would be preferable to use cross-validation together with permutation testing for a more robust approach. For correlations between gaze data and behavioral data, it seems that about 10 correlation analyses were done. It would be important to correct for the multiple tests. This is also a particularly problematic issue in the saliency analyses, where there are several that are barely at the magic "P<0.05" threshold.

We completely agree with the reviewer that the statistical design used in our initial manuscript version, with several correlations presented simultaneously, was not optimal. Following the reviewers’ comments, the revised manuscript uses a multivariate approach better suited for dealing with highly colinear variables. The advantage of the current analysis is that we can appreciate the relation between the π and several behavioral characteristics simultaneously in relation to one another. The results of the multivariate approach confirm that the higher values of the Proximity Index are found in children with better developmental and functional levels across all assessed domains. We found no stable association between the π and symptom severity as measured by the ADOS at baseline. However, the π at baseline was associated with the phenotype measures one year later in our longitudinal sample. The description of the relation between visual exploration (PI) and the phenotype characteristics (cross-sectional and longitudinal analyses) is now included in the current version of the manuscript as follows:

*“*Less divergence in visual exploration is associated with better overall functioning in children with ASD

To explore how the gaze patterns, specifically divergence in the way children with ASD attended to the social content, related to the child’s functioning, we conducted a multivariate analysis. We opted for this approach to obtain a holistic vision of the relationship between visual exploration, as measured by PI, and different features of the complex behavioral phenotype in ASD. Behavioral phenotype included the measure of autistic symptoms and the developmental and functional status of the children with ASD. Individuals with ASD often present lower levels of adaptive functioning (Franchini et al., 2018; Hus Bal et al., 2015) and this despite cognitive potential (Klin et al., 2007). Understanding factors that contribute to better adaptive functioning in very young children is of utmost importance (Franchini et al., 2018) given the important predictive value of adaptive functioning on later quality of life. The association between behavioral phenotype and π was examined using the PLS-C analysis (Krishnan et al., 2011; McIntosh and Lobaugh, 2004). This method extracts commonalities between two data sets by deriving latent variables representing the optimal linear combinations of the variables of the compared data sets. We built the cross-correlation matrix using the π on the left (A) and 12 behavioral phenotype variables on the right (B) side (see Methods section for more details on the analysis).

In our cohort, child autistic symptoms were assessed using the ADOS (Lord et al., 2000; Lord et al., 2012), child developmental functioning using the PEP-3 scale (Schopler, 2005) and child adaptive behavior using the Vineland Adaptive Behavior Scales, Second Edition, (Sparrow, Balla, and Cicchetti, 2005). Thus the final behavior matrix included two domains of autistic symptoms from the ADOS: social affect (SA) and repetitive and restricted behaviors (RRB); six subscales of the PEP-3: verbal and preverbal cognition (VPC), expressive language (EL), receptive language (RL), fine motor skills (FM), gross motor skills (GM), oculomotor imitation (OMI) and four domains from VABS-II: communication (COM), daily living skills (DAI), socialization (SOC) and motor skills (MOT). Age was regressed from both sets of the imputed data.

The PLS-C yielded one significant latent component (r = 0.331, p = 0.001), best explaining the crosscorrelation pattern between the π and the behavioral phenotype in the ASD group. The significance of the latent component was tested using 1000 permutations, and the stability of the obtained loadings was tested using 1000 bootstrap resamples. Behavioral characteristics that showed stable contributions to the pattern reflected in the latent component are shown in red Figure 7. Higher values of the π were found in children with better developmental functioning across all six assessed domains and better adaptive functioning across all four assessed domains. Autistic symptoms did not produce a stable enough contribution to the pattern (loadings showed in gray bars on the Figure 7). Still, numerically, a more TD-like gazing pattern (high PI) was seen in the presence of fewer ASD symptoms (negative loading of both SA and RRB scales of the ADOS-2). Despite the lack of stability of this pattern, the loading directionality of ASD symptoms is in line with the previous literature (Avni et al., 2019; Wen et al., 2022), showing a negative relationship between visual behavior and social impairment. Among the developmental scales, the biggest loading was found on verbal and preverbal cognition, followed by fine motor skills. While the involvement of verbal and nonverbal cognition in the PI, an index of visual exploration of these complex social scenes is no surprise, the role of fine motor skills might be harder to grasp. Interestingly, in addition to measuring the control of hand and wrist small muscle groups, the fine motor scale also reflects the capacity of the child to stay focused on the activity while performing controlled actions. Thus, besides the measure of movement control, relevant as scene viewing implies control of eye movement, the attentional component measured by this scale might explain the high involvement of the fine motor scale in the latent construct pattern we obtain.”

“More divergence in visual exploration is associated with unfolding autistic symptomatology a year later

To capture the developmental change in the π and its relation to clinical phenotype we conducted the multivariate analysis considering only the subjects that had valid eye-tracking recordings at two time points one year apart. Out of 94 eligible children (having two valid eye-tracking recordings a year apart), 81 had a complete set of phenotype measures. All 94 children had an ADOS, but ten children were missing PEP-3 (9 were assessed using Mullen Scales of Early Learning (Mullen, 1995), one child was not testable at the initial visit), and three children were missing VABS-II as the parents were not available for the interview at a given visit. The proximity index in this smaller paired longitudinal sample was defined using the age-matched reference composed of 29 TD children spanning the age (1.66-5.56) who also had a valid eye-tracking recording a year later. As the current subsample was smaller than the initial one, we limited our analyses to more global measures, such as domain scales (not the test subscales as in our bigger cross-sectional sample). Thus, for the measure of autistic symptoms, we used the total severity score of ADOS. Cognition was measured using the Verbal and preverbal cognition scale of PEP-3 (as the PEP-3 does not provide a more global measure of development (Schopler, 2005)) and adaptive functioning using the Adaptive behavior Composite score of Vineland (Sparrow, Balla, and Cicchetti, 2005). To test how the π relates within and across time points, we built three cross-covariance matrices (T1-PI to T1-symptoms; T1-PI to T2-symptoms; T2-PI to T2-symptoms) with the π on one side (A) and the measure of autistic symptoms, cognition, and adaptation on the other side (B). As previously, the significance of the patterns was tested using 1000 permutations, and the stability of the significant latent components using 1000 bootstrap samples.

The PLS-C conducted on simultaneous π and phenotype measues at the first time point (T1-PI – T1 symptoms) essentially replicated the pattern we observed on a bigger cross-sectional sample. One significant LC (*r*=0.306 and *p*=0.011) showed higher π co-occurring with higher cognitive and adaptive measures (see Figure 12). The cross-covariance matrix using a π at T1 to relate to the phenotype at the T2 also yielded one significant latent component (*r*=0.287 and *p*=0.033). Interestingly, the pattern reflected by this LC showed higher loading on the π co-occurring with lower loading on autistic symptoms. Children who presented lower π values at T1 were the ones with higher symptom severity at T2. The gaze pattern at T1 was not related to cognition nor adaptation at T2 (see Figure 13, panel A). Finally, the simultaneous PLS-C done at T2 yielded one significant LC where higher loading of the π coexisted with negative loading on autistic symptoms and higher positive loading on the adaptation score (*r*=0.322 and *p*=0.014) Figure 13, panel B. The level of typicality of gaze related to the symptoms of autism at T2(mean age of 4.05±0.929) but not at a younger age (mean age of 3.01±0.885). This finding warrants further investigation. Indeed, on the one hand, the way children with TD comprehend the world changes tremendously during the preschool years, and this directly influences how the typicality of gaze is estimated. Also, on the other hand, the symptoms of autism naturally change over the preschool years, and all these elements can be responsible for the effect we observe.”

For the developmental patterns of gaze, we now conducted permutation testing analysis over the 59 sliding windows to establish the statistical significance of our finding that the gaze patterns in ASD become progressively dispersed with age while the gaze patterns become more coherent over time in TD. In the current analyses, inside each of the 59 windows, gaze data of ASD and TD children were permuted 100 times to derive the null distribution of the measure of dispersion -average pairwise distance. The statistically significant difference in gaze dispersion between groups is evident between the age of the 2.5 and 3 years old (Appendix 2fFigure 3 Panel C). The corresponding manuscript parts are modified as follows:

“Divergent developmental trajectories of visual exploration in children with ASD

After exploring the π association with various aspects of the behavioral phenotype in ASD children, we were also interested in the developmental pathway of visual exploration in this complex social scene for both groups of children. Previous studies using cross-sectional designs have demonstrated important changes in how children attend to social stimuli depending on their age (Frank, Vul, and Saxe, 2012; Helo et al., 2014). As our initial sample spanned a relatively large age range (1.7 – 6.9 years), we wanted to obtain a more fine-grained insight into the developmental dynamic of visual exploration during the given period. To that end, when study-specific inclusion criteria were satisfied, we included longitudinal data from our participants who had a one-year and/or a two years follow-up visit (see Methods section). With the available 306 recordings for the ASD group and 105 for the TD group, we applied a sliding window approach (Sandini et al., 2018) (see Methods section). Our goal was to discern critical periods of change in the visual exploration of complex social scenes in ASD compared to the TD group. We opted for a sliding window approach considering its flexibility to derive a continuous trajectory of visual exploration and thereby capture such non-linear periods. The sliding window approach yielded a total of 59 age-matched partially overlapping windows for both groups covering the age range between 1.18 – 4.28 years (mean age of the window) (Figure 8 panel A illustrates the sliding window method).

We then estimated gaze dispersion on a group level across all 59 windows. Dispersion on a single frame was conceptualized as the mean pairwise distance between all gaze coordinates present on a given frame (Figure 8, panel B). Gaze dispersion was computed separately for ASD and TD. The measure of dispersion indicated an increasingly discordant pattern of visual exploration between groups during early childhood years. The significance of the difference in the gaze dispersion between two groups across age windows was tested using the permutation testing Methods section. The statistically significant difference (at the level of 0.05) in a window was indicated using color-filled circles and as can be appreciated from the Figure 8, panel C was observed in 46 consecutive windows out of 59 starting the age of 2.5 to 4.3 (average age of the window). While the TD children showed more convergent visual exploration patterns as they got older, as revealed by progressively smaller values of dispersion (narrowing of focus), the opposite pattern characterized gaze deployment in children with ASD. From the age of 2 years up to the age of 4.3 years, this group showed a progressively discordant pattern of visual exploration (see Figure 8, panel C).

And finally, for the saliency analyses, Wilcoxon t-test of group differences was significant for the full model (Figure 5) and all the 5 channels taken individually (Appendix 2-figure 1). We report the effect sizes according to formula r=Z/N, (Rosenthal, 1991).

“The relative contribution of the basic visual properties of the animated scene to gaze allocation in ASD and TD children

… for all channels taken individually as well as for the full model, the salience model better predicted gaze allocation in the TD group compared to the ASD group (Wilcoxon t-test returned with the value of <0.001, Figure 10). The effect sizes (r=Z/N, (Rosenthal, 1991)). of this difference were most pronounced for the flicker channel *r* = 0.182, followed by the orientation channel *r* = 0.149, full model *r* = 0.132, intensity *r* = 0.099, color *r* = 0.083 and lastly motion *r* = 0.066, Appendix 2. The finding that the salient model predicted better gaze location in TD groups compared to the ASD was not expected based on the previous literature.Still, most studies used static stimuli and gaze control when viewing dynamic content is very different. As the salience model used in this work has been validated on adults, our findings suggest that the gaze behavior in TD approximates that of TD adults better than the ASD gaze behavior.

6. Eyetracking exclusions. This is insufficiently described in the paper. We are only told that subjects were excluded if >45% of frames were dropped. First of all, this is an extremely lenient threshold. But we need to know what the distribution of dropped frames was between ASD and TD groups. We also need to know what other exclusions were applied to any portion of the data.

As in our first version of the manuscript, subjects were excluded if they showed poor screen attendance, defined as binocular gaze detection on less than 65% of video duration. Our exclusion criterion is, thus, more than 35% of dropped frames. No other exclusion criteria were applied to any portion of the data. While we understand that the threshold of 35% dropped frames might appear lenient, choosing it looked up to the thresholds of other studies in preschool children. Pierce and collaborators, in the much shorter Geometric preference task (1 minute) administered in toddlers (TD, ASD, DD) aged 12-43 months, defined a threshold of 50% of stimulus duration (Pierce et al., 2011, 2016; Wen et al., 2022). The authors also reported that the viewing time was significantly different between the groups in all studies. Then, in their 2013prospective study with children aged two months at the study onset, Jones and collaborators note "Trials in which a child failed to fixate on the presentation screen for a minimum of 20% total trial duration were excluded from analyses.". The apparent lack of a consensus regarding the screen attendance threshold might be due to differences in populations, stimuli, and goals of the studies. Our eye-tracking stimulus was coupled with simultaneous high-density EEG recording, and the data acquisition is particularly challenging in this context. More stringent inclusion criteria would undoubtedly result in a much smaller but more biased group, as the children with ASD who struggle to tolerate the EEG cap on their head while watching a cartoon are usually the ones who have more pronounced symptoms of ASD. To control for the missing data, we omitted the instances of non-fixation data (saccades, blinks, off-screen moments) in all the proximity index calculations. Thus the average value of the proximity index is based uniquely on the moments where eyes were detected under the condition that children looked at the screen for more than 65% of the time. Following the reviewer’s comment in the current version of the manuscript methods section, we added the following information:

“Eye-tracking analysis

We excluded data from participants who showed poor screen attendance, defined as binocular gaze detection on less than 65% of video frames. The screen attendance was somewhat higher in the TD sample (93.8 ±} 6.37 seconds) compared to the ASD group (87.8 ±} 9.33 seconds), U=2568, p < 0.001. To extract fixations, we used the Tobii IV-T Fixation filter (Olsen, 2012) (i.e., Velocity threshold: 30◦/s; Velocity window length: 20ms. Adjacent fixations were merged Maximum time between fixations was 75ms; Maximum angle between fixations was 0.5°). To account for differences in screen attendance, we omitted instances of non-fixation data (saccades, blinks, off-screen moments) in all calculations.

Reviewer #3 (Recommendations for the Authors)These results are interesting and valuable to our understanding of the development of social visual attention. However, several weaknesses should be addressed that would strengthen the results. First, the authors assume Gaussian distribution of TD eye gaze, but some of their example figures show that this is not always the case. This may lead to lower proximity index scores and may inflate the significant results rather than reflecting the true proximity of gaze to the normative distribution. Second, it would be helpful to discuss more of the individual differences in normative viewing to help anchor some of the main points of the paper. Finally, causality is assumed in the lower-level visual saliency analyses when instead the social and visual saliency may be highly correlated (and inseparable).

We are very grateful to the reviewer for a very thorough analysis of our paper and for highly inspiring comments.

It would be helpful if the authors could provide more details about the π scores, specifically the normalization of them. The authors should explain how the normalization of the π scores was conducted and if this normalization allows for consistency across frames, where the possible furthest distance from the mode of the Gaussian distribution may change depending on the x- and y-coordinates of the mode of the Gaussian on the screen. Additionally, a description of how the normalization of π scores may change based on the convergence of TD children (i.e., how peaked the distribution is) would be helpful. If and how these measures may be limited should be discussed.

We thank the reviewer for this pertinent question. Following the reviewer’s comment, we realized that the term "normalization " was inadequate to reflect the analyses we did. In our approach, we did not post hoc normalize values of the PI. The correct definition would be that while calculating the PI, we defined a range for the values of the π (from 0 to 1) to allow comparability between frames. As we detail more in the revised manuscript, the values of the π are obtained as a function of the isolines projected onto the density matrix. In the current manuscript version, we project 100 isolines on every density matrix to facilitate the interpretation of their relation with the PI. Thus the gaze coordinates captured only by the isoline of the lowest level (nº1) will have a π value of 0.01. Accordingly, the gaze coordinates captured by the isoline at level 50 will obtain the π value of 0.50, and level 100 will yield the π value of 1.

Moreover, the calculation of the π depends directly on the density of the distribution of the TD gaze coordinates, so when the distribution is peaked, this would be represented by tightly packed kernels and thus the physical area concerned with the distribution is smaller than in the cases of wider attention distribution. In this case of extremely peaked distribution, attaining the maximum level of the π (PI=1) is more challenging than in the moments of more widespread distribution. This feature of the normative distribution (that we now refer to as "referent") could further be used to ponder the gaze differences even more, where the π can be weighted by the relative difficulty of the frame (peaked referent distribution = frame more difficult). For the current paper, we explored the global properties of the π (averaged over all frames) but in our future work, we intend to explore more in detail the fine-tuning of the π with regards to the frame content and temporal dynamics. The revised version of the manuscripts comprises the following paragraph in the Methods section:

“Upon the "reference" definition, we calculated the distance of gaze data from this referent distribution on each frame for each child with ASD (*n* = 166; 3.37 ± 1.16 years). Comparison to this referent pattern yielded a measure of *Proximity Index-PI* (see Figure 1). The calculation of the Proximity Index values was done for each frame separately. Proximity Index values were scaled from 0 to 1 at each frame for comparison and interpretation. We used the Matlab inbuilt function *contour* to delimit isolines of the gaze density matrix. To have a fine-grained measure, we defined 100 isolines per density matrix (i.e., each frame). Then we calculated the proximity index for each child with ASD framewise. Gaze coordinates that landed outside the polygon defined by contour(s) of the lowest level (1) obtained a π value of 0. The gaze coordinates inside the area defined by gaze density matrix isolines obtained the π value between 0.01 and 1. The exact value of these non-zero π values was obtained depending on the level number of the highest isoline/contour that contained the x and y coordinates of the gaze. As we defined 100 isolines per density matrix, the levels ranged from 1 to 100. Accordingly, a gaze coordinate that landed inside the highest contour (level 100) obtained a π value of 1, and the one that landed inside the isoline 50 obtained a π value of 0.50. A high π value (closer to the mode of the density distribution) indicates that the visual exploration of the individual for a given frame is less divergent from the reference (more TD-like). A summary measure of divergence in visual exploration from the TD group was obtained by averaging the π values for the total duration of the video.”

Additionally, Figure 1 shows a child's π score on a frame-by-frame level across the video. Frames where the child was looking offscreen were coded as -0.15. The authors should explain why this value was chosen, and why a value was chosen at all instead of excluding these frames from analysis. Additionally, it would be important to know that these frames are moments that the child is looking off-screen and not moments where the child is blinking.

We agree that the details in Figure 1 of the initial manuscript was somewhat misleading. Indeed, the value "-0.15" was used to visually illustrate the off-screen moments for the recording used in the example in Figure 1. We used it to indicate the periods where the child was not looking at the screen to distinguish them from moments where the child obtained a π value of 0. We have now modified Figure 1 (in the current version of the manuscript) so that it does not contain misleading information in the present document, the figure is also denoted as Figure 1.

For all the analyses, the missing gaze data were considered "NaNs" and were not included in the π calculation. A part of the data loss is also due to the blinking, and we did not interpolate these moments after applying the Tobii-IVT fixation filter.

It is unclear how the authors handled instances where there were two distinct clusters of gaze distribution and the distribution was therefore not Gaussian. This will directly impact the π score and may make children with ASD look more atypical in their viewing patterns than they truly are. For example, Figure 3b shows both groups having two distinct clusters of visual attention, but more children with ASD are attending to the second focal point than TD children. Additional information on these instances should be added, and limitations should be discussed.

We thank the reviewer for this highly pertinent question. Indeed, one of the study’s main goals was to develop a method sensitive to the complexity of gaze distribution in a rich in details scene. Preserving the complexity of attention distribution (e.g., having two or more distant foci of attention) was essential to us while creating our method, as this would correspond to a more flexible and ecological definition of gaze behavior. The coexistence of multiple foci allows for pondering the relative importance of the different scene elements from the point of view of the TD group. It further distinguishes our method from hypothesis-driven methods that measure aggregated fixation data in the scene’s predefined regions.

As detailed in our initial version of the manuscript (please see below), our approach uses a density estimation function that is flexibly adapting to the data without a predefined smoothing parameter. For this reason, we used *kde2d* (Botev, Grotowski, and Kroese, 2010) MATLAB function as it achieves this flexibility by applying a Gaussian kernel of an adaptive bandwidth on the data. Figure 3b in the initial version of the manuscript, in our view, was a good illustration that in this manner the complexity of gaze deployment is preserved (we observe two distinct clusters and not a unique cluster – potentially Gaussian). Moreover, while the smoothing kernel deployed in our density estimation function is Gaussian, we would like to state that the final distribution of the gaze data is not assumed Gaussian. We carefully revised all elements of the previous version of the manuscript that might have been misleading regarding this point. For the frames where the attention of the TD group showed many distinct focal points, like the one in Figure 3b in the initial manuscript version, we calculated the π in the same manner as for frames with a unique focus distribution. For a given gaze coordinate from a child with ASD, we identify the level (ranging from 0.01 to 1) of the highest contour (of any of the attention focus/clusters) containing that coordinate. Suppose we assume a hypothetical situation where the gaze data of the TD group are falling along two clusters identically (i.e., we obtain the density peaks of the same level/height), in this. In that case, the π value will obtain a value of 1 if the gaze coordinates are captured by any of the two contours of the highest level. The following text was added in the Methods section:

*“*While the smoothing kernel deployed in our density estimation function is Gaussian, the final distribution of the gaze data is not assumed Gaussian. As shown in Figure 1, right upper panel, the final distribution was sensitive to the complexity of gaze distribution (e.g., having two or more distant gaze foci in the TD group) which allowed a flexible and ecological definition of referent gaze behavior. The coexistence of multiple foci allows for pondering the relative importance of the different scene elements from the point of view of the TD group. It further distinguishes our method from hypothesis-driven methods that measure aggregated fixation data in the scene’s predefined regions. For the frames where the gaze of the TD group showed many distinct focal points, like the one in Figure 1, right upper panel, we calculated the π in the same manner as for frames that had a unique focus distribution. For a given gaze coordinate from a child with ASD, we identify the level of the highest contour, ranging from 0.01 to 1, of any of the attention focus/clusters containing that coordinate. If we assume a hypothetical situation where the gaze data of the TD group are falling along two clusters identically (i.e., we obtain the density peaks of the same level/height), in this case, any two gaze coordinates that fall in the highest level of any of the peaks would obtain a π value of 1.”

It would be helpful if the authors could validate their normative gaze distribution with leave-one-out procedures or some other method to ensure that the normative distribution is not shifted by one participant. This same procedure would be necessary for the maturational sliding window to show that the gaze pattern is actually reflecting developmental change, not just change due to individual differences in two participants' gaze data (the participant newly included and the participant newly excluded in the sliding window).

We are very thankful to the reviewer for these valuable suggestions. We indeed already touched upon these topics in our responses to other reviewers, but we repeat the argumentation here as well.

For the stability of the gaze distribution, we conducted bootstrapping analysis. Inspired by others (Schaer et al., 2015) we performed bootstrapping procedure to simulate smaller samples originating from the available, total sample of 51 TD. For each sample size level (ranging from 10 to 50 TD children), we obtained 500 bootstrapped samples over which we measure the stability of the distribution. We found that the stability of the distribution in the TD sample is on average stable from the sample size of 18, Figure 2. In the present version of the paper, we added a subsection in the supplementary materials that details these analyses as follows:

*“*The sample of 51 TD children whose gaze data was used to obtain a normative gaze distribution was a convenience sample. In the present study, we only included males due to the fewer number of females with ASD. Having this unique sample of TD children, we tested the stability of the normative distribution depending on the sample size by performing bootstrap analyses. Thus, from the available sample of 51 TD children, we performed 500 bootstraps, starting with a sample size of 10 until reaching the sample size of 50. To measure the change in gaze distribution on one frame, we calculated the average pairwise distance between all gaze coordinates available on the frame. Then for each frame, we calculated the variance of the average pairwise distance over 500 resamples. Finally, the variance obtained was averaged over the 5150 frames to yield a unique value of the variance in gaze patterns per sample size (10-50). Then we calculated the "cutoff," as defined by a sample size increase no longer yielding significant variation in the average variance. This was done using the *kneed* package implemented in Python that estimates the point of maximal curvature ("elbow in curves with positive concavity) in discrete data sets based on the mathematical definition of curvature for continuous functions (Satopaa et al., 2011) (see Figure 2). The elbow of the fitted curve on our bootstrapping data was found at 18, meaning that the distribution was estimated to be stable from a sample size of 18.”

For the maturational sliding window, we used a bigger sample than in the initial version of the manuscript using the window size of 20. Based on the findings above on the stability of distribution on smaller sample size, we can conclude that the sample size of 20 allows capturing the developmental effect with sufficient strength. To test the statistical significance of the differences, we conducted permutation testing. We applied 100 permutations inside each of the 59 windows (containing 20TD + 20ASD gaze recordings) to derive a null distribution of the measure of dispersion -the average pairwise distance between gaze coordinates inside the group. The statistically significant difference in gaze dispersion between groups emerges after the age of 2.5 years (Figure 8 Panel C). The corresponding manuscript parts are modified as follows:

Methods:

“Sliding window approach

[…] We opted for a sliding window approach adapted from Sandini et al. (2018) to delineate fine-grained changes in visual exploration on a group level. Available recordings from our unstructured longitudinal sample were first ordered according to the age in both groups separately. Then, for each group, a window encompassing 20 recordings was progressively moved, starting from the first 20 recordings in the youngest subjects until reaching the end of the recording span for both groups. The choice of window width was constrained by the sample size of our TD group. The longitudinal visits in our cohort are spaced a year from each other, and the choice of a bigger window would result in significant data loss in our group of TD children as the windows were skipped if they contained more than one recording from the same subject. The chosen window width yielded 59 sliding windows in both groups that were age-matched and spanned the period from 1.88 – 4.28 years old on average.

Upon the creation of sliding windows and to characterize the group’s visual behavior and its change with age, gaze data from the TD group were pooled together to define the referent distribution in each of the 59 age windows. To characterize the group visual behavior in the ASD group, we performed the same by pooling the gaze data together from ASD in each of the 59 age windows (see Figure 5 A&B). We calculated the mean pairwise distance between all gaze coordinates on every frame for the measure of gaze dispersion in each of the two groups. Then we compared the relative gaze dispersion between groups on the estimated gaze density of each group in each age window separately.

To test for the statistical significance of the difference between the two groups, we employed random permutation testing across 59 age windows. Accordingly, in each of the 59 windows, gaze data from the TD (20) and ASD groups (20 recordings per window) were pooled together. We performed 100 randomly permuted resamples of equal size to the original distribution (20) from this pooled sample to compute the significance value. The windows where the MCS values showed statistically significant differences between the two groups are graphically presented with color-filled circles (Figure 8C).”

Results:

“Divergent developmental trajectories of visual exploration in children with ASD

[…] We then estimated gaze dispersion on a group level across all 59 windows. Dispersion on a single frame was conceptualized as the mean pairwise distance between all gaze coordinates present on a given frame (Figure 8, panel B). Gaze dispersion was computed separately for ASD and TD. The measure of dispersion indicated an increasingly discordant pattern of visual exploration between groups during early childhood years. The significance of the difference in the gaze dispersion between two groups across age windows was tested using the permutation testing Methods section. The statistically significant difference (at the level of 0.05) in a window was indicated using color-filled circles and as can be appreciated from the Figure 8, panel C was observed in 46 consecutive windows out of 59 starting the age of 2.5 to 4.3 (average age of the window). While the TD children showed more convergent visual exploration patterns as they got older, as revealed by progressively smaller values of dispersion (narrowing of focus), the opposite pattern characterized gaze deployment in children with ASD. From the age of 2 years up to the age of 4.3 years, this group showed a progressively discordant pattern of visual exploration (see Figure 8, panel C).”

It would be worthwhile to include a figure of the correlation of π and autism symptom severity in Figure 2.

We agree with the reviewer that a figure of correlation of the π and autism symptoms would have been useful in our initial manuscript version. Our initial decision to limit the number of figures to only statistically significant findings was a potentially limiting factor for better transparency of our work. In the current version of the manuscript, we changed the analysis strategy. Instead of uni-variate measures of association between the π and the phenotype (several correlations analyses), we now conducted the multivariate PLS-C analysis. PLS-C allowed a more comprehensive view of the relationship between visual exploration and the clinical characteristics of the children.

In Figure 3c-d, what do the shaded regions represent?

In our previous analyses we used a GLM to establish the significance of the difference in the intercept and slope between the two groups over 40 windows. The shaded regions represented the 95% confidence interval of the fitted model (quadratic). In the review process, the complete set of these analyses has been changed along with the corresponding figure. We now used permutation testing to establish the significance of the group difference in each of the sliding windows. Thus the shaded regions are no longer present (see Figure 8).

Mean and standard errors of the Proximity Index were not reported for every comparison Figure 4. This could also be collapsed to a difference score for each participant to allow easier comparisons across figures. This particular analysis is complex and would benefit from some greater explanation of the comparisons and how to interpret them in both the results and the Discussion section. This feels like it would be the primary analysis of the paper and it was under-developed in both the results and discussion.

Following the major changes in the paper, using now the bigger sample, we completely changed our strategy also for the developmental analyses. Instead of the approach where we simply compared the Proximity indexes at the first time point with the Proximity index values to the next one, we currently adopt the approach allowing us to observe the proximity index in the context of autistic symptomatology, developmental and adaptive functioning. This was done using the same multivariate approach as in our current strategy for the cross-sectional approach (PLS-C). The Result section now contains the following description of the developmental analyses:

*“*More divergence in visual exploration is associated with unfolding autistic symptomatology a year later

To capture the developmental change in the π and its relation to clinical phenotype we conducted the multivariate analysis considering only the subjects that had valid eye-tracking recordings at two time points one year apart. Out of 94 eligible children (having two valid eye-tracking recordings a year apart), 81 had a complete set of phenotype measures. All 94 children had an ADOS, but ten children were missing PEP-3 (9 were assessed using Mullen Scales of Early Learning (Mullen, 1995), one child was not testable at the initial visit), and three children were missing VABS-II as the parents were not available for the interview at a given visit. The proximity index in this smaller paired longitudinal sample was defined using the age-matched reference composed of 29 TD children spanning the age (1.66-5.56) who also had a valid eye-tracking recording a year later. As the current subsample was smaller than the initial one, we limited our analyses to more global measures, such as domain scales (not the test subscales as in our bigger cross-sectional sample). Thus, for the measure of autistic symptoms, we used the total severity score of ADOS. Cognition was measured using the Verbal and preverbal cognition scale of PEP-3 (as the PEP-3 does not provide a more global measure of development (Schopler, 2005)) and adaptive functioning using the Adaptive behavior Composite score of Vineland (Sparrow, Balla, and Cicchetti, 2005). To test how the π relates within and across time points, we built three cross-covariance matrices (T1-PI to T1-symptoms; T1-PI to T2-symptoms; T2-PI to T2-symptoms) with the π on one side (A) and the measure of autistic symptoms, cognition, and adaptation on the other side (B). As previously, the significance of the patterns was tested using 1000 permutations, and the stability of the significant latent components using 1000 bootstrap samples.

The PLS-C conducted on simultaneous π and phenotype measures at the first time point (T1-PI – T1 symptoms) essentially replicated the pattern we observed on a bigger cross-sectional sample. One significant LC (*r*=0.306 and *p*=0.011) showed higher π co-occurring with higher cognitive and adaptive measures (see Appendix ??). The cross-covariance matrix using a π at T1 to relate to the phenotype at the T2 also yielded one significant latent component (*r*=0.287 and *p*=0.033). Interestingly, the pattern reflected by this LC showed higher loading on the π co-occurring with lower loading on autistic symptoms. Children who presented lower π values at T1 were the ones with higher symptom severity at T2. The gaze pattern at T1 was not related to cognition nor adaptation at T2 (see Figure 13, panel A). Finally, the simultaneous PLS-C done at T2 yielded one significant LC where higher loading of the π coexisted with negative loading on autistic symptoms and higher positive loading on the adaptation score (*r*=0.322 and *p*=0.014) Figure 13, panel B. The level of typicality of gaze related to the symptoms of autism at T2(mean age of 4.05±0.929) but not at a younger age (mean age of 3.01±0.885). This finding warrants further investigation. Indeed, on the one hand, the way children with TD comprehend the world changes tremendously during the preschool years, and this directly influences how the typicality of gaze is estimated. Also, on the other hand, the symptoms of autism naturally change over the preschool years, and all these elements can be responsible for the effect we observe.”

The authors could consider strengthening their results by including some additional analyses exploring individual differences. Children with ASD on the whole become less convergent with each other over time, but are there children who become more convergent with the TD group and those who do not?

This is a fascinating question. Indeed, we showed previously in the analyses using the sliding windows that children with ASD become more divergent from their group over the childhood years, unlike TD children, who show more group cohesion with age. While the sliding window design allows us to obtain a fine-grained inference on the developmental processes over childhood, a better approach would include more densely sampled purely longitudinal data, which at the time of writing this revised manuscript, we did not have. We can conclude from the present data that the more pronounced effect of change with age seems to happen on the side of the TD, not ASD, meaning that in one year, the TD changed more impressively than the ASD. The trajectories of development in ASD are also more heterogeneous, so the average effect is inevitably blunted. Still, to be able to address properly the heterogeneity of the visual exploration pathway in ASD an important topic we would need to include a precise measure of the interventions the children receive following the diagnosis. After diagnosis, almost all children start some intervention, speech therapy, occupational therapy, or more structured intensive naturalistic behavioral interventions. Previous research from our group showed that the type of intervention children receive impacts the way children attend to social information (Latrèche et al., 2021). Given the complexity of the question, we decided this would be a better fit for a separate study. We would also like to have a bigger sample of children with ASD at the follow-up visit two years after the initial one in order to be able to have a more thorough insight into the trajectories in this very dynamic developmental period.

A decent amount of space in the paper is dedicated to analyses that are only included in the supplemental analyses. The authors may consider restructuring some of the main and supplemental texts such that the relevant figures will appear near the analyses.

We thank the reviewer for this suggestion. In the review process, we changed many of the analyses, so some of the figures from the supplementary materials were omitted. The current supplementary material contains only complementary figures to the analyses we present in the Results section.

The authors suggest a direction of causality wherein TD children are relying more on lower-level salient features of the scene than are children with ASD. However, salient features of a scene as predicted by the Itti & Koch model are often highly correlated with the social aspects of a scene. Especially in a cartoon, backgrounds remain consistent and the only movement is vibrantly-colored characters moving across the scene. The authors should edit the methods to exclude the suggestion of causality and the above point should be discussed.

We thank the reviewer for attracting our attention to the misleading phrasing. Indeed, our intention was not to convey any causality in interpreting the salience findings. We were interested in group differences regarding how gaze may be influenced by the low-level salience features of the scene. All between-group comparisons are made frame by frame. While we fully agree that we cannot disentangle these basic aspects of the scene from the purely social ones, we aimed to highlight the potential group differences in how the salience model predicted gaze allocation. We modified the corresponding parts of the manuscript as follows:

Methods:

“Previous research has put forward the enhanced sensitivity to the low-level (pixel-level) saliency properties in adults with ASD while watching static stimuli (Wang et al., 2015) compared to healthy controls. We were interested in whether any low-level visual properties would more significantly contribute to the gaze allocation in one of the groups.”

Results:

“Contrarily to our hypothesis, for all channels taken individually as well as for the full model, the salience model better predicted gaze allocation in the TD group compared to the ASD group (Wilcoxon t-test returned with the value of p < 0.001, Figure 10). The effect sizes r=Z/N, (Rosenthal, 1991) of this difference were most pronounced for the flicker channel r = 0.182, followed by the orientation channel r = 0.149, full model r = 0.132, intensity r = 0.099, color r = 0.083 and lastly motion r = 0.066, Figure 11. The finding that the salient model predicted better gaze location in TD groups compared to the ASD was not expected based on the previous literature.

Still, most studies used static stimuli and the processes implicated in the process of the dynamic content are very different. The salience model itself was validated on the adult vision system. It might be that the gaze in TD better approximates the adult, mature gaze behavior than the gaze behavior in the ASD group.

The authors should discuss why gaze behavior correlated with adaptive behavior scales but not with overall autism symptom severity.

We agree with the reviewer, in our previous manuscript we did not devote enough attention to the relation between the π and autistic symptomatology. Still, despite the absence of a significant association between π and autism severity, these negative findings warrant discussion. In the current version of the manuscript, as previously, we do not find a significant relationship between the severity of autistic symptomatology (as measured by ADOS) and the π in the cross-sectional sample. One possible reason can be the lack of granularity of the ADOS scale (ranging from 3-10 in our sample of ASD children) compared to the Vineland scale. Interestingly, in our current analyses, while we do not find the relationship between the ADOS and the π initially (at the initial time point) we find that the π at T1 is related to the symptoms of ADOS a year later, at T2. Additionally, ADOS scores at T2 are also related to π at T2. This might be due to the better stability of autistic symptomatology at the visit a year later. In the new version of the manuscript we added the following paragraph to the Discussion section:

“We showed that the level of divergence in gaze exploration of this 3-minute video was correlated with ASD children’s developmental level in children with ASD and their overall level of autonomy in various domains of everyday life. This finding stresses the importance of studying the subtlety of gaze deployment with respect to its downstream contribution to more divergent global behavioral patterns later in development (Jones and Klin, 2013; Klin, Shultz, and Jones, 2015; Schultz, 2005; Young et al., 2009). Gaze movements in a rich environment, as the cartoon used here, inform not only immediate perception but also future behavior as experience-dependent perception now is likely to alter the ongoing developmental trajectory. In accordance with this view, the level of typicality of visual exploration in ASD children at T1 was related to the level of autistic symptoms at T2 but not at T1. One possible interpretation of the lack of stable association at T1 might be due to the lower stability of symptoms early on. Indeed, while diagnoses of ASD show stability with age, still a certain percentage of children might show fluctuation. The study by Lord and collaborators (Lord et al., 2006) following 172 2-year-olds up to the age of 9 years old showed that diagnosis fluctuations are more likely in children with lesser symptoms compared to children with more severe symptoms. Still, as our study included all ASD severities, it is subject to such fluctuations. Another possible interpretation comes from the maturation of the gaze patterns in the TD group, against which we define the typicality of gaze in the ASD group. As can be seen in our results, children with TD show a tremendous synchronization of their gaze during the age range considered, resulting in a tighter gaze distribution at T2 and thus, a more sensitive evaluation of ASD gaze at that time point. The possibility that TD show more similar gaze allocation with age, while ASD’s gaze becomes increasingly idio-syncretic with age, highlights the value of addressing the mechanisms underlying the developmental trajectories of gaze allocation in future studies.”

In general, additional context could be provided to the Results section to clarify what questions the authors were trying to answer with each analysis.

We thank the reviewer for this valuable suggestion. In the revised manuscript we added more descriptions at the beginning of subsections in the Results section. Also, analyses in the Results section are mirrored by the corresponding description in the Methods section that provides complementary details with regard to the goals and theoretical motivation for the analyses.

[Editors’ note: further revisions were suggested prior to acceptance, as described below.]

The manuscript has been improved but there are some remaining issues that need to be addressed, as outlined below:1) Clarify the "coarse movie characteristic". The description provided does not allow us to replicate this metric.

We thank both the Editor and Reviewer 2 for attracting our attention to the lack of clarity in this aspect of our manuscript. As explained in detail in our response to Reviewer 2, the term "coarse movie characteristics" refers to two global, high-level aspects of the movie: abrupt changes in the movie frames from one scene to another, and a distinction between all frames where the background is static vs all frames where the backgrounds moves along with the moving characters. Our aim was to probe how these global properties of the movie exert an influence on how the gaze is directed. We think that our clarification eases the concern of lack of replicability about this aspect of our study.

2) Details for the permutation testing on the longitudinal changes in gaze divergence using a sliding window method are missing. What exactly was done to compute the significance value?

We observed in the longitudinal data that TD children become increasingly similar to each other in their gaze deployment compared to ASD children. For each longitudinal window, we have an estimate of each group’s dispersion and we computed a difference between the dispersion across the two groups (i.e., TD vs. ASD). The permutation analysis allows us to make a statistical inference about the observed difference between groups by first generating a null distribution of differences in the gaze dispersion between random group assignments which preserves the general statistics of the data but corrupts the true grouping assignments, and then comparing the original estimate against the estimates of the null. Thus, for each longitudinal window (each of the 59 longitudinal windows) and for each permutation iteration (i.e., each of the one hundred random iterations) we randomly assigned each child to one of two groups (e.g., g1 and g2) preserving the original number of group members (i.e., 20 children per group), we then computed the dispersion quantity within each newly formed group (by computing the average distance in gazes across the entire movie) and computed the difference, across the two groups, in the average dispersion values. The one hundred iterations thus formed a null distribution against which we could compare the original estimation in a two-tailed fashion with an α value of 5%, the p-value is the proportion of the random samples that resulted in estimated differences between the groups that were higher than or equal to the original difference estimation.

3) Clarify the role of the "reference distribution". This distribution seems not to have been used in any of the analyses.

We acknowledge and agree that the use of the term "reference distribution" in the Method section of "Maturational changes in visual exploration of complex social scene – Sliding window approach" is inaccurate. In the section where we measure developmental changes, we do not compare each individual child with ASD to a "reference distribution." Instead, we are focused on evaluating how the group behavior, as defined by group gaze dispersion, evolves across defined sliding windows of age.

As detailed in our response to Reviewer 2, we have made revisions to the manuscript to rectify this terminology and avoid any further misunderstanding.

4) More comparisons of the TD and ASD groups at the individual subject level should be performed to provide more insightful information. This could also address the major concern of reviewer #3 that the groups are quite different with respect to homogeneity and size.

We sincerely appreciate the suggestion to delve deeper into individual-level processes. In the first part of our paper, we indeed focus on individual-level information and examine how each individual with ASD behaves in comparison to normative visual exploration patterns. We demonstrated that individual deviations from normative behavior exhibit distinctive phenotypic characteristics and are associated with less optimal developmental functioning, as well as an increase in ASD symptoms in subsequent years. Our next objective in this paper was to document the developmental dynamics in the visual exploration of social scenes during early childhood. Here, we aimed to outline this developmental dynamic at the population level rather than the individual level; we accordingly employed a sliding window approach to highlight a notable trend of progressive convergence among typically developing (TD) children over the preschool years, wherein their manner of viewing social scenes became increasingly consistent across participants. Conversely, we observed a contrasting process among children with ASD, where, at the group level, we noted an aberrant maturation process and a growing degree of heterogeneity.

In response to the valuable feedback from the Editor and Reviewers, we have incorporated additional individual-level approaches into our developmental analyses. By utilizing a mixed model approach, we were able to assess developmental changes in children with ASD compared to their typically developing counterparts throughout the preschool years. Our findings highlight that in TD children, within-subject effects align closely with the group effect, of increasing convergence with the own group, whereas in children with ASD, we observe more heterogeneous patterns of maturation with development. We hope that these additional analyses offer a more nuanced perspective on the complex interplay of factors involved. In the revised manuscript, we have included solely group-level analysis, as we intend to perform a more in-depth study of the question of individual-level analysis in the context of a separate paper, as suggested by Reviewer 2. However, if the Editor and the Reviewers deem it necessary, we are willing to include this level of analysis in the manuscript.

5) A control analysis taking into account mental age would be helpful as chronological and mental age seem to differ more in ASD and this analysis could rule out potential confounds that are indicative of general developmental delays rather than ASD-specific characteristics.

We appreciate the concerns raised by both the Editor and Reviewer 3 regarding the potential impact of developmental delay. In our manuscript, we initially did not match our two groups based on intellectual or functional levels. Our rationale for this decision was to maintain our focus on gaining a comprehensive understanding of visual exploration in a diverse group of children with ASD, rather than exclusively studying those at the (relatively) higher functioning end of the autistic spectrum. However, in direct response to the feedback provided by Reviewer 3, we detail the steps undertaken in a more fine-grained evaluation of developmental trajectories in visual exploration. This evaluation involves considering the developmental age of children with ASD and aligning it with their typically developing peers. We have reanalyzed our data using these developmental age-matched samples. Remarkably, our findings remain consistent with our initial results. Specifically, we observe atypical developmental patterns in visual exploration among children with ASD when compared to their typically developing counterparts, despite these being matched for developmental age. We have provided a more detailed exposition of these novel findings in response to Reviewer 3’s second comment. We thank the reviewer for suggesting this additional analysis which underscores the robustness and reliability of our findings and allows us to explore the developmental nuances within our study population more comprehensively.

Reviewer #2 (Recommendations for the authors):Points for clarification:– I am a bit unclear on what the "coarse movie characteristic" is, exactly. The description provided would not allow me to replicate this metric – can a more quantitative description be provided?

We appreciate the reviewer bringing to our attention the lack of precision in our reporting of this particular aspect of our analysis. What we have labeled "coarse movie characteristics" represents a foundational attribute of this cartoon design. Our intention was to explore how these properties of the cartoon design influenced the deployment of gaze. We defined two coarse movie characteristics; these were then extracted manually.

The first characteristic, denoted "frame switch," encompasses all instances in which the cartoon employs an abrupt frame transition using the hard-cut montage technique (like the example illustrated in Figure 1). Throughout the duration of the movie, this event type occurs 25 times (as indicated in Figure 2). During these moments, the viewer’s gaze necessitates recalibration to synchronize with the new scene. Accordingly, the ability to disengage from the previous scene and adapt to the novel social context at a pace similar to the normative group might have an impact on the Proximity index. The second characteristic labeled as the "Moving background" pertains to moments when the cartoon’s background moves in tandem with the characters, following their directional motion. We aimed to distinguish these segments from scenes featuring a static background, as the overall motion dynamics in these frames varied. The occurrence of a moving background is observable in 5 distinct sequences within the movie (as illustrated in Appendix 1-figure 1). We have accordingly revised the method section to provide a more accurate description.

“Coarse movie characteristics: Frame switching and moving background Finally, to test how the global characteristics of video media influence gaze deployment, we focused on two movie features.

The first feature, denoted as the "Frame switch," encompasses all instances in which the cartoon employs an abrupt frame transition using the hard-cut montage technique. To represent this feature numerically, a feature vector was created. In this vector, the first frame following the switch is assigned a code of 1, while all other frames are coded as 0. This coding scheme effectively highlights the occurrence of these abrupt shot changes within the movie. Throughout the duration of the movie, this event type occurs 25 times (as indicated in Figure 6).

The feature labeled as the "Moving background" pertains to moments when the cartoon’s background moves in tandem with the characters, following their directional motion. We aimed to distinguish these segments from scenes featuring a static background, as the overall motion dynamics in these frames varied. The occurrence of a moving background is observable in 5 distinct sequences within the movie (as illustrated in Figure 6). Frames with a moving background were coded 1 yielding a binary feature vector.

ARI

– In response to Reviewer 3, RC4, the authors now provided permutation testing on the longitudinal changes in gaze divergence using a sliding window method. However, I am unclear on the details of the permutation testing. To quote from the paper: "To test for the statistical significance of the difference between the two groups, we employed random permutation testing across 59 age windows. Accordingly, in each of the 59 windows, gaze data from the TD (20) and ASD groups (20 recordings per window) were pooled together. We performed 100 randomly permuted resamples of equal size to the original distribution (20) from this pooled sample to compute the significance value. The windows where the MCS values showed statistically significant differences between the two groups are graphically presented with color-filled circles (Figure 8C)." What exactly was done to compute the significance value? As far as I understand, they calculated the group divergence from 20 resampled data to generate a 'null divergence' for each group, then calculate that difference 100 times as a null distribution of the group difference, and finally compare the actual group difference with this null distribution. However, from their response letter, the distribution seems to be the group-level dispersion, but not the group difference. In their rebuttal letter, the authors write, "We applied 100 permutations inside each of the 59 windows (containing 20TD + 20ASD gaze recordings) to derive a null distribution of the measure of dispersion -the average pairwise distance between gaze coordinates inside the group." This seems inconsistent.

Indeed, for each age window (comprising 20 recordings from ASD children and 20 recordings from TD children, totaling 59 windows), we calculated the average dispersion value across all frames in parallel for both groups. Available recordings from our unstructured longitudinal sample were first ordered according to the age in both groups separately. Then, for each group, a window encompassing 20 recordings was progressively moved, starting from the first 20 recordings in the youngest subjects until reaching the end of the recording span for both groups. Dispersion on an individual frame is defined as the average pairwise distance across all pairs of members of the same group.

To assess whether group differences in dispersion were statistically significant within windows, we conducted 100 permutations inside each window. In each permutation, we created two new samples (g1 and g2), by randomly selecting 20 recordings from the combined dataset of that specific window (40 recordings in total). Subsequently, we computed the dispersion values for each group within the newly generated samples (g1 and g2), averaging them across all frames. This iterative process enabled us to construct the null distribution of the dispersion measure for each window. Following this, we examined the proportion of sampled permutations in which the disparity in dispersion between g1 and g2 deviated from the original estimate with the true groupings. A significance level (α) of 0.05 was set for the analysis. If the observed disparity in dispersion fell within the lower or upper 2.5% of the null distribution, we rejected the null hypothesis and concluded that the observed difference held statistical significance. This was consistently the outcome for all windows beginning from the average age of 2.5 years. We modified the specific paragraph in the methods section of the manuscript as follows:

*“*To quantify the heteroscedasticity between groups across different ages, we computed the difference in dispersion (mean pairwise distance to members of own group), denoted as (disp_t(ATD) – disp_t(ASD)), for each time window (t). Then, the permutation method was used in order to get the distribution under the null hypothesis in each window (t) (H0: disp_t(TD) – disp_t(ASD)=0). Thus, for each window (59) 100 permutations (i) were performed (i.e. individuals were mixed up randomly in each group) and then we computed our statistic (disp_ti(TD) – disp_ti(ASD)) for each permuted sample (i) and each time window (t). The hundred statistics per window thus formed a null distribution (the expected behavior of our statistic under the null hypothesis) against which we could compare the"real" statistic estimated in the original sample. The p-value is the probability of getting a statistic at least as extreme as the one we observed in our sample if we consider H0 to be the truth. The windows where the dispersion values showed statistically significant differences between the two groups are graphically presented with color-filled circles (Figure 8C).”

– In the Method of the "Maturational changes in visual exploration of complex social scene – Sliding window approach", in the second paragraph, the authors mention a reference distribution. However, this reference distribution seems not to have been used in any of the analyses (it's not mentioned in the longitudinal results at all). To quote from the paper: "Upon the creation of sliding windows and to characterize the group's visual behavior and its change with age, gaze data from the TD group were pooled together to define the referent distribution in each of the 59 age windows. To characterize the group visual behavior in the ASD group, we performed the same by pooling the gaze data together from ASD in each of the 59 age windows (see Figure 8 A&B)." Indeed, Figure 8A is about how to decide the sliding window, and Figure 8B is about pairwise gaze dispersion, not about referent distribution.

We agree with the reviewer that the use of the term "reference distribution” here might appear misleading. Indeed, in this set of analyses we were interested in the group gaze behavior thus the idea of a unique reference is not relevant here. What we meant by the reference distribution here is that we extracted a distribution of the TD gaze in each of the 59 windows. As we did not calculate the π index in each window, the proper terminology should have been “TD distribution”. We changed the Mthods section of the manuscript as follows:

“Upon the creation of sliding windows and to characterize each group’s visual behavior and its change with age, gaze data from the TD group were pooled together to define the TD distribution in each of the 59 age windows. To characterize the group visual behavior in the ASD group, we performed the same by pooling the gaze data together from the ASD group in each of the 59 age windows (see Figure 3 A&B).”

Reviewer #3 (Recommendations for the authors):The present research uses the Typical Development (TD) group as a normative reference, comparing individual participants with Autism Spectrum Disorder (ASD) against this reference. However, as pointed out by Reviewer #2, this approach doesn't allow for a comprehensive understanding of the variation within the TD group itself. The leave-one-out calculation suggests that the π is higher for TD, but the TD group is also more homogenous, so I am not sure this is truly informative. A comparison between TD and ASD groups at the individual subject level could provide more insightful information.

In the initial manuscript version we compared the ASD children to the referent group falling short of showing how would the TD group behave if put in relation to another control group. Following the suggestion put forth by Reviewer 2 during the first round of review, we have successfully rectified this important limitation of our study. Specifically, we adopted the leave-one-out method, which allowed us to address the absence of an additional control group, and we are pleased to report that this development has met Reviewer 2’s approval. This method yielded the Proximity Index (PI) values for our TD group, demonstrating that, on average, TD children exhibit higher π values (see Figure 4). Importantly, it is noteworthy that these TD children also display interindividual variations similar to those observed in the ASD group. Furthermore, statistical analysis revealed that the variance between the two groups was not statistically significant. In the first part of our paper, we then focused on individual-level information and examined how each individual with ASD behaves in comparison to normative visual exploration patterns. We have demonstrated that individual deviations from normative behavior exhibit distinctive phenotypic characteristics and are associated with less optimal developmental functioning, as well as an increase in ASD symptoms in subsequent years.

While the behavioral characterization of the TD group implies a higher degree of homogeneity, attributed to their normal IQ range and absence of ASD symptoms, our investigation into the visual exploration patterns among TD individuals also captured an inherent interindividual diversity (as shown in Figure 4). This finding inspired our next steps where we wanted to address more specifically the topic of intragroup variation, notably its temporal evolution, in the latter portion of our manuscript, specifically within the "Developmental Patterns of Visual Exploration" section of our results. We provide a broad-strokes delineation of the developmental dynamics in the visual exploration of social scenes during early childhood. To achieve this, we employed a sliding window approach to highlight a notable trend of progressive convergence among typically developing (TD) children over the preschool years, wherein their manner of viewing social scenes became increasingly consistent. Conversely, we observed a contrasting process among children with ASD, where, at the group level, we noted an aberrant maturation process and a growing degree of heterogeneity. Regrettably, delving into individual-level developmental information was initially beyond the scope of our manuscript. In response to your feedback, we have incorporated additional individual-level approaches into our developmental analyses, which we will elaborate on in more detail as part of our response to your following comment.

Additional Concerns:i) The study matches the two groups based on chronological age rather than mental age. This introduces a potential confound as the differences reported may be indicative of general developmental delays rather than ASD-specific characteristics.ii) The TD group is not only smaller in size but also less heterogeneous, which may be a potential explanation for the findings illustrated in Figure 8C.

We agree with the reviewer that addressing the matters of age matching, size, and homogeneity holds significant relevance. To effectively tackle these concerns, we employed a sequential approach in which we first addressed the reviewer’s second concern, the question of inequality of size, followed by the combined consideration of size and homogeneity. Subsequently, our analysis incorporated a sliding window methodology that utilized developmental age, as opposed to chronological age, as the focal parameter. This sequential approach was undertaken to ensure a comprehensive and nuanced exploration of the concerns raised by the reviewer.

Size. In response to the disparity in sample size between our two groups (51 TD and 167 ASD children), we implemented a methodology to mitigate the influence of this factor. We generated 100 bootstrapped ASD samples (without replacement), each with a size identical to that of the TD (51 subjects). These ASD samples were matched to the TD sample in terms of chronological age. Subsequently, for each of the bootstrapped samples, we aggregated all longitudinal data and computed the dispersion measure over time, akin to the process described in Figure 8C of the manuscript. As illustrated in Figure 6, Panel A, the results reveal that the bootstrapped ASD samples, characterized by both size and chronological age alignment with the TD group, exhibit higher levels of dispersion across the span of childhood years. This is in contrast to TD children, who exhibit a discernible pattern of progressive refinement in their visual exploration behavior.

.

It’s worth noting that, while permutation testing could have been an ideal method for assessing the statistical significance of the findings in this section, we opted not to implement it due to the substantial computational cost associated with our analyses. The computational demands of our study necessitated an alternative approach to address the sample size and age-matching issue effectively. Consequently, we relied on the bootstrapping technique to provide valuable insights into the dispersion differences between the TD and ASD groups, while acknowledging the limitations imposed by the computational constraints.

Homogeneity. To address the question of intragroup homogeneity, taking into account the considerable developmental heterogeneity inherent in the ASD group, we restricted the range of developmental functioning. Thus we derived 100 simulated samples of the same size as the TD group (51) firstly within the normal developmental range (DQ above 80) and then, we performed the same for the lower-functioning individuals with ASD (DQ below 80). As shown in Figure 6, Panels B-C, both groups show sustained dispersion over the childhood years, in contrast to the convergence seen in the TD group. This trend is particularly pronounced in the subset of individuals with lower developmental functioning (Panel C), wherein a discernible divergence becomes increasingly evident during the preschool years.

Developmental age. To comprehensively address the question of developmental age-matched samples, we implemented a sliding window approach using the same dataset as in our manuscript (51 TD and 167 ASD children). However, in this approach, we utilized developmental age for creating age-matched windows instead of chronological age as previously used. We initiated the process with the first 20 recordings from subjects with the lowest developmental age and progressively shifted a window encompassing 20 recordings. This continued until the entire range of recordings for both groups was covered. Similar to the method applied in the manuscript, we excluded windows containing duplicate recordings from the same subject. This method yielded a total of 60 windows, each matched based on age, with developmental age in the ASD group and chronological age in the TD group (developmental age and chronological age are highly aligned, r = 0.93, p = 6.82E-23). To test the stability of our findings and assess the potential influence of sample size, we replicated the sliding window procedure using 100 bootstrapped ASD samples, each comprising 51 subjects whose developmental age was matched to the chronological age of the TD subjects. For the purpose of interpretation, we plotted a linear regression line (in red) for each bootstrapped sample. Remarkably, our results reinforce our initial findings when using chronological age-matched samples. Children with ASD consistently exhibit a greater degree of interindividual disparity across childhood years, in contrast to TD children. This outcome underscores the robustness of our findings and strengthens the validity of our observations.

Individual level developmental change. In response to the valuable feedback from the Reviewer, we have integrated additional individual-level approaches into our developmental analyses, recognizing their crucial role in understanding the dynamics of visual exploration. In the previous step, we employed a method that utilized the mental age of children with ASD to match them with typically developing (TD) subjects of the same chronological age. However, instead of aggregating the information regarding an individual’s deviation from their respective group at the level of a sliding window, we retained individual-level data. This approach enabled us to delve deeper into the intricacies of individual developmental trajectories using the mixed model approach. The mixed model approach is well-suited for designs like the present case, where we have multiple time points for each subject, with varying numbers of observations. For our analysis, we utilized a publicly available toolbox (Mancini et al., 2019; Mutlu et al., 2013) in MATLAB R2021a (MathWorks). We modelled age and diagnosis as fixed effects, while within-subject factors were treated as random effects, utilizing the nlmefit function. To estimate developmental trajectories, we employed random-slope models, including constant, linear, quadratic, and cubic terms, each representing a different relationship between age and the averaged pairwise distance to members of the same group. These models accounted for both within-subject and between-subject effects. We determined the most appropriate model order using the Bayesian information criterion.

Strikingly, in typically developing children, we observed a strong alignment between within-subject effects and the group effect, signifying a trend of increasing convergence within the TD group. Conversely, in children with ASD, our observations unveiled again more heterogeneity in patterns of maturational processes. These nuanced insights are graphically represented in Figure 7, providing a visual representation of the developmental trajectories that underlie our findings.

The following paragraph has been added to the Results section:

“To ensure the robustness and validity of our findings, we addressed several potential confounding factors. These included differences in sample size TD (TD sample included 51 and ASD sample 166 children), the heterogeneity of ASD behavioral phenotypes, and the use of developmental age rather than chronological age in our sliding window approach. We adopted a sequential approach, first examining the impact of unequal sample sizes and then considering both sample size and phenotypic heterogeneity together. Additionally, we implemented a sliding window methodology using developmental age as the primary matching parameter for a detailed description, see Appendix 5. Our results consistently reaffirmed our initial findings obtained when using chronologically age-matched samples. Specifically, when matched for both sample size and developmental age, children with ASD consistently demonstrated a greater degree of interindividual disparity across childhood years compared to TD children (Appendix 5, Panels D1-D2).”

The Appendix 5 has been added as a supplementary material.